# ONLINE INFORMATION ACQUISITION: HIRING MULTIPLE AGENTS

**Federico Cacciamani**
Politecnico di Milano
`federico.cacciamani@polimi.it`

**Matteo Castiglioni**
Politecnico di Milano
`matteo.castiglioni@polimi.it`

**Nicola Gatti**
Politecnico di Milano
`nicola.gatti@polimi.it`

## ABSTRACT

We investigate the mechanism design problem faced by a principal who hires *multiple* agents to gather and report costly information. Then, the principal exploits the information to make an informed decision. We model this problem as a game, where the principal announces a mechanism consisting in action recommendations and a payment function, a.k.a. scoring rule. Then, each agent chooses an effort level and receives partial information about an underlying state of nature based on the effort. Finally, the agents report the information (possibly non-truthfully), the principal takes a decision based on this information, and the agents are paid according to the scoring rule. While previous work focuses on single-agent problems, we consider multi-agents settings. This poses the challenge of coordinating the agents' efforts and aggregating correlated information. Indeed, we show that optimal mechanisms must correlate agents' efforts, which introduces externalities among the agents, and hence complex incentive compatibility constraints and equilibrium selection problems. First, we design a polynomial-time algorithm to find an optimal incentive compatible mechanism. Then, we study an online problem, where the principal repeatedly interacts with a group of unknown agents. We design a no-regret algorithm that provides $\widetilde{\mathcal{O}}(T^{2/3})$ regret with respect to an optimal mechanism, matching the state-of-the-art bound for single-agent settings.

## 1 INTRODUCTION

Acquiring reliable information is crucial in any decision making problem. Often, decision makers delegate the task of gathering information to other parties. In the classical information acquisition scenario (Savage, 1971), a principal delegates a *single* agent to acquire information (Chen & Yu, 2021; Papireddygari & Waggoner, 2022; Li et al., 2022; Chen et al., 2023). However, in many real-world scenarios, the principal may have multiple sources of information (Cacciamani et al., 2023). Consider a portfolio manager that wants to learn the potential of a company to make an informed investment. The manager could hire multiple analysts to conduct separate researches on the same company, where each analyst spends effort to produce a report. The manager gains information from the reports and decides whether or not to make the investment. To incentivize the analysts to produce accurate reports, the manager designs a payment scheme that pays the analysts based on the reports accuracy. Formally, this problems can be modeled as a game between a principal that wants to acquire information about a stochastic state $\theta$ and a group of agents that receive information about this state through signals whose accuracy depends on undertaken effort (effort levels are modeled through different *actions*). The game goes as follows. First, the principal commits to a distribution $\boldsymbol{\mu}$ over action recommendations and payment function, *a.k.a.*, scoring rule (Oesterheld & Conitzer, 2020; Neyman et al., 2021). Then, each agent $i$ observes an action recommendation sampled from $\boldsymbol{\mu}$ and performs a costly action $b_i$. Each agent receives a signal $s_i$ from a joint probability distribution $\mathbb{P}^{(i)}(\boldsymbol{s}|\boldsymbol{b}, \theta)$ and reports the signal (possibly non-truthfully) to the principal. Based on the reported information, the principal makes a decision $a \in \mathcal{A}$. Finally, the agents are paid according to the

scoring rule. Our goal is twofold. First, we want to analyze the optimization problem of computing optimal mechanisms, characterizing their properties and computational complexity. Moreover, we study an online problem in which the principal does not have prior knowledge about the agents. Our goal is to design online no-regret learning algorithms that repeatedly interacts with an unknown group of agents maximizing the cumulative principal's utility.[1]

**Original contributions.** We study the design of efficient algorithms for the multi-agent information acquisition problem, with a focus on both the optimization problem and the online learning problem arising when a principal interacts repeatedly with unknown agents. First, we assume that the principal knows all the game parameters. We show that an optimal mechanism can be computed efficiently. Our algorithm solves a quadratic optimization problem by providing a linear relaxation and then recovers a solution to the original problem in polynomial time. Moreover, we characterize the settings in which the optimal mechanism is correlated, *i.e.*, each agent's payment depends also on the signals reported by other agents, and the settings in which the optimal mechanism is uncorrelated, *i.e.*, each agent's payment depends only from her reported signal and the state of nature. Next, we consider the online problem in which the principal does not know the game parameters and needs to learn them by repeatedly interacting with the agents. We present an online algorithm that attains $\widetilde{\mathcal{O}}(T^{2/3})$ regret with respect to an optimal mechanism, which matches the state-of-the-art bound for single-agent settings (Chen et al., 2023). Our algorithm comprises three phases. In the first one, we estimate the probability distribution over states of nature induced by the agents' signals. Ensuring that the estimations are sufficiently accurate is the main challenge in this phase. The required level of accuracy depends on the specific instance and it is crucial for designing strictly-*incentive compatible* (IC) scoring rules, which maintain truthfulness under uncertainty. In the second phase, the algorithm estimates the differences among costs of the agents' actions. To do so, we employ non-truthful mechanisms, which require a non-trivial analysis of the agents' behavior. Finally, the algorithm commits to an approximately optimal strategy while ensuring truthfulness under uncertainty. We achieve this goal by leveraging the estimations obtained in the previous phases. Specifically, we find an approximately optimal and approximately IC mechanism by exploiting the estimations. Then, we take a convex combination between this scoring rule and an ad-hoc strictly IC scoring rule to obtain an IC mechanism.

## 2 PRELIMINARIES

**Game model.** We investigate games between a *principal* and a set $\mathcal{N} = [n]$ of $n$ *agents*[2]. We assume that $n$ is constant.[3] Each agent $i \in \mathcal{N}$ can choose an action $b$ from a set $B_i$ of $k$ actions, each one with a cost $c_i(b)$ specified by a cost function $c_i : B_i \to [0, 1]$. For $i \in \mathcal{N}$ and $b_i, b_i' \in B_i$, we let $C_i(b_i, b_i') = C_i(b_i) - c_i(b_i')$. We denote the set of all possible action profiles as $\mathcal{B} := \times_{i \in \mathcal{N}} B_i$ and the set of all possible action profiles excluding agent $i$'s action as $\mathcal{B}_{-i} = \times_{j \neq i} B_j$. Given $\boldsymbol{b} \in \mathcal{B}$, we denote as $\boldsymbol{b}_{-i} \in \mathcal{B}_{-i}$ the tuple obtained from $\boldsymbol{b}$ by removing element $b_i$ relative to agent $i$. The interaction model we consider is enhanced with a *pre-play* round of communication between the principal and the agents in which the principal can privately recommend to each agent which action to take. [4] After each agent takes an action $b_i \in B_i$ (not necessarily equal to the action recommended by the principal), a state of nature $\theta$ is sampled from a finite set $\Theta = \{\theta_1, \ldots, \theta_m\}$ according to a *prior* $\boldsymbol{p} \in \Delta(\Theta)$.[5] The state of nature is observed neither by the principal nor by the agents. Instead, depending on the action profile $\boldsymbol{b} \in \mathcal{B}$ chosen by all the agents, each agent $i$ observes a signal $s_i$ drawn from a finite $l$-dimensional set $S_i$. The signal profile received by the principal is denoted as $\boldsymbol{s} = (s_1, ..., s_n)$. The set of all possible signal profiles is $\mathcal{S} = \times_i S_i$. Moreover, the set of all possible signal profiles excluding agent $i$'s signal is denoted as $\mathcal{S}_{-i} = \times_{j \neq i} S_j$. Given a signal profile $\boldsymbol{s} \in \mathcal{S}$, we let $\boldsymbol{s}_{-i} \in \mathcal{S}_{-i}$ the signal profile obtained from $s_i$ by removing signal $s_i$. The signal profile and the state of nature are sampled according to a joint probability distribution that depends

---

[1] For space constraints, we defer a discussion on the related works to Appendix A.

[2] In this work, for any $n \in \mathbb{N}_{>0}$, we use $[n] = \{1, .., n\}$ to denote the set of the first $n$ natural numbers.

[3] This avoids computational issues related to the exponential-size representation of the problem instance. Most of our results continue to hold for arbitrary $n$.

[4] In Section 4, we show that optimal mechanisms correlate the agents' efforts and hence this step is fundamental to maximize the principal's utility.

[5] In this work, given a finite set $\mathcal{Z}$, we denote as $\Delta(\mathcal{Z})$ the $|\mathcal{Z}|$-dimensional probability simplex.

on the agent's action profile $\mathbb{P}\left(\boldsymbol{s}, \theta | \boldsymbol{b}\right) := p[\theta]\mathbb{P}\left(\boldsymbol{s}|\theta, \boldsymbol{b}\right)$. The marginal signal probability of agent $i$ is defined as $\mathbb{P}^{(i)}\left(s_i|\boldsymbol{b}, \theta\right) := \sum_{\boldsymbol{s}' \in \mathcal{S}: s_i' = s_i} \mathbb{P}\left(\boldsymbol{s}|\boldsymbol{b}, \theta\right)$, where we use the superscript $(i)$ to explicit that we are considering the probability with which agent $i$ receives signal $s_i \in S_i$. We assume that the information received by each agent $i$ is independent from the actions taken by other agents[6], *i.e.,*

$$\mathbb{P}^{(i)}\left(s_i|\boldsymbol{b}, \theta\right) := \mathbb{P}^{(i)}\left(s_i|b_i, \theta\right) \quad \forall i \in \mathcal{N}, \, \forall s_i \in S_i, \, \forall \boldsymbol{b} \in \mathcal{B}, \forall \theta \in \Theta.$$

Notice that this does not exclude that the signals received by the agents are correlated. Moreover, we denote as $\mathbb{P}^{(i)}\left(s_i|b_i\right) := \sum_\theta p(\theta)\mathbb{P}^{(i)}\left(s_i|b_i, \theta\right)$ the probability with which agent $i$ observes signal $s_i$ after the agent played action $b_i$. Finally, we denote with $\mathbb{P}^{(i)}\left(\theta|b_i, s_i\right)$ the probability of state $\theta$ in the posterior induced to agent $i \in \mathcal{N}$ by signal $s_i \in S_i$ and action $b_i \in B_i$. In our model, the agents can communicate a signal observation (possibly lying and communicating a signal $s_i'$ different than the signal $s_i$ actually received) to the principal, which can pay the agents to reward their communication. After receiving a signal profile $\boldsymbol{s}'$ from the agents, the principal chooses an action $a$ from a finite set $\mathcal{A} = \{a_1, ..., a_d\}$ and receives utility $u(a, \theta) \in [0, 1]$.

**Mechanisms.** A mechanism for the principal must specify three different components: (i) a *recommendation policy* $\boldsymbol{\mu}$ that determines which actions are recommended to the agents, (ii) a *payment scheme* $\boldsymbol{\gamma} = (\gamma_1, ..., \gamma_n)$ – also called *scoring rule* – specifying how each agent will be paid, and (iii) an *action policy* $\boldsymbol{\pi}$ encoding which action the principal chooses as a function of the received signal profile and the recommended actions. In this work we are interested in the class of *correlated mechanisms*, in which the payment received by agent $i$ depends on the whole action profile $\boldsymbol{b}$ recommended to the agents and on the whole signal profile $\boldsymbol{s}$ received by the principal, as well as on the state of nature $\theta$. The set of correlated mechanisms is denoted as $\mathcal{C}$. Formally,

$$C := \{(\boldsymbol{\mu}, \boldsymbol{\gamma}, \boldsymbol{\pi}) \mid \boldsymbol{\mu} \in \Delta(\mathcal{B}), \, \boldsymbol{\gamma}_i : \mathcal{B} \times \mathcal{S} \times \Theta \to [0, M] \, \forall i \in \mathcal{N}, \, \boldsymbol{\pi} : \mathcal{B} \times \mathcal{S} \to \Delta(\mathcal{A})\},$$

where $M$ is a parameter that limits the principal's budget. [7] When the principal uses mechanism $(\boldsymbol{\mu}, \boldsymbol{\gamma}, \boldsymbol{\pi})$, the action profile recommended is $\boldsymbol{b} \in \mathcal{B}$, the signal profile reported is $\boldsymbol{s} \in \mathcal{S}$ and the state of nature is $\theta$, the payment received by agent $i$ is $\gamma_i[\boldsymbol{b}, \boldsymbol{s}, \theta]$, while $\pi[\boldsymbol{b}, \boldsymbol{s}, a]$ denotes the probability with which the principal plays action $a \in \mathcal{A}$.

**Optimal mechanisms and incentive compatibility.** The objective of the principal is to find an optimal mechanism, *i.e.,* a mechanism guaranteeing the maximum possible difference between utility of the principal and total payments. By the revelation principle, it is possible to restrict our attention to mechanism that are *truthful*, *i.e.,* such that the agents are incentivized to follow the principal's recommendations and to report truthfully the signals they observe. Hence, we denote the expected payment received by player $i$ when she behaves truthfully and the mechanism is $(\boldsymbol{\mu}, \boldsymbol{\gamma}, \boldsymbol{\pi})$ as:

$$F_i(\boldsymbol{\mu}, \boldsymbol{\gamma}) = \sum_{\boldsymbol{b} \in \mathcal{B}} \sum_{\boldsymbol{s} \in \mathcal{S}} \sum_{\theta \in \Theta} \mu[\boldsymbol{b}]\mathbb{P}\left(\boldsymbol{s}, \theta|\boldsymbol{b}\right)\gamma_i[\boldsymbol{b}, \boldsymbol{s}, \theta].$$

Then, the expected utility of the principal is obtained as the difference between the expected utility received by actions in $\mathcal{A}$ and the total expected payments (assuming truthful behavior of the agents):

$$U(\boldsymbol{\mu}, \boldsymbol{\gamma}, \boldsymbol{\pi}) = \sum_{\boldsymbol{b} \in \mathcal{B}} \sum_{\boldsymbol{s} \in \mathcal{S}} \sum_{\theta \in \Theta} \left[ \mu[\boldsymbol{b}]\mathbb{P}\left(\boldsymbol{s}, \theta|\boldsymbol{b}\right) \left[ \sum_{a \in \mathcal{A}} \pi[\boldsymbol{b}, \boldsymbol{s}, a]u(a, \theta) \right] \right] - \sum_{i \in \mathcal{N}} F_i(\boldsymbol{\mu}, \boldsymbol{\gamma}).$$

To ensure that truthful behavior is optimal (*i.e.,* the mechanism is IC), we introduce the concept of *deviation functions*, which model the possible deviations from truthful behavior. Formally, the set $\Phi_i$ of agent $i$'s deviation functions is $\Phi_i = \{(\phi, \varphi) \mid \phi : B_i \to B_i, \, \varphi : B_i \times S_i \to S_i\}$. Given any couple $(\phi, \varphi) \in \Phi_i$, the function $\phi$ models the deviation from the recommended action, while $\varphi$ models untruthful reporting of the received signal. Agent $i$'s expected payment when she deviates according to $(\phi, \varphi)$, all the other agents behave truthfully, and the mechanism is $(\boldsymbol{\mu}, \boldsymbol{\gamma}, \boldsymbol{\pi})$ is:

$$F_i^{\phi, \varphi}(\boldsymbol{\mu}, \boldsymbol{\gamma}) = \sum_{\boldsymbol{b} \in \mathcal{B}} \sum_{\boldsymbol{s} \in \mathcal{S}} \sum_{\theta \in \Theta} \mu[\boldsymbol{b}]\mathbb{P}\left(\boldsymbol{s}, \theta|(\phi(b_i), \boldsymbol{b}_{-i})\right)\gamma_i[\boldsymbol{b}, (\varphi(b_i, s_i), \boldsymbol{s}_{-i}), \theta].$$

---

[6]Intuitively, this assumption models those cases in which the information received by an agent depends exclusively on her level of effort.

[7]Bounding the payments is a classical assumption in online problems related to information acquisition and in principal-agent problems Chen et al. (2023); Zhu et al. (2023). Without this assumption the learner decision space is unbounded.

An optimal correlated mechanism can be found as a solution of the following optimization problem:

$$\max_{(\boldsymbol{\mu}, \boldsymbol{\gamma}, \boldsymbol{\pi}) \in \mathcal{C}} \quad U(\boldsymbol{\mu}, \boldsymbol{\gamma}, \boldsymbol{\pi}) \tag{1a}$$

$$\text{s.t.} \quad F_i(\boldsymbol{\mu}, \boldsymbol{\gamma}) - F_i^{\phi,\varphi}(\boldsymbol{\mu}, \boldsymbol{\gamma}) \geq \sum_{\boldsymbol{b} \in \mathcal{B}} \mu[\boldsymbol{b}] C_i(b_i, \phi(b_i)) \quad \forall i \in \mathcal{N}, \, \forall (\phi, \varphi) \in \Phi_i \tag{1b}$$

The objective function of Equation (1) is the maximization of principal's expected utility assuming honest behavior of the agents, and Equation (1b) guarantees that the mechanism is incentive compatible (IC), *i.e.,* it guarantees that, for all agents, truthful behavior is an equilibrium.

# 3 A LINEAR PROGRAMMING RELAXATION FOR COMPUTING OPTIMAL MECHANISMS

In this section, we provide a polynomial-time algorithm to solve Problem (1), *i.e.*, to find an optimal mechanism. As a first step, we provide a Linear Program (LP) relaxation of (1). Notice that (1) presents two main issues: (i) the objective function and the constraints are non-linear in the variables $(\boldsymbol{\mu}, \boldsymbol{\gamma}, \boldsymbol{\pi})$, and (ii) it has an exponential number of constraints, since $|\Phi_i| = k^k l^{lk}$. To address (i), we introduce variables $\boldsymbol{x} = (\boldsymbol{x}_1, ..., \boldsymbol{x}_n)$ and $\boldsymbol{y}$, where $\boldsymbol{x}_i \in \mathbb{R}_{\geq 0}^{|\mathcal{B}| \times |\mathcal{S}| \times |\Theta|}$ for each $i \in \mathcal{N}$, and $\boldsymbol{y} \in \mathbb{R}_{\geq 0}^{|\mathcal{B}| \times |\mathcal{S}| \times |\mathcal{A}|}$. Intuitively, $x_i[\boldsymbol{b}, \boldsymbol{s}, \theta]$ represents the product $\mu[\boldsymbol{b}]\gamma_i[\boldsymbol{b}, \boldsymbol{s}, \theta]$, while $y[\boldsymbol{b}, \boldsymbol{s}, a]$ represents the product $\mu[\boldsymbol{b}]\pi[\boldsymbol{b}, \boldsymbol{s}, a]$, thus making the objective function and the constraints linear. This yields a relaxation of the original non-linear optimization problem. Then, in order to be able to recover valid mechanisms from variables $\boldsymbol{x}_i, \boldsymbol{y}$, we introduce additional constraints. For what concerns (ii), in order to reduce the number of constraints, we observe that it is possible to safely consider a restricted set of deviations for each agent, while still guaranteeing incentive compatibility w.r.t. deviations in $\Phi_i$. In the following, we will denote the expected payment received by agent $i$ when she is recommended to play $b_i \in B_i$, she plays $b_i'$, observes $s_i \in S_i$, and reports $s_i' \in S_i$ as:

$$f_i(\boldsymbol{x}_i | b_i, b_i', s_i, s_i', \mathbb{P}) = \sum_{\boldsymbol{b}_{-i} \in \mathcal{B}_{-i}} \sum_{\boldsymbol{s}_{-i} \in \mathcal{S}_{-i}} \sum_{\theta \in \Theta} x_i[(b_i, \boldsymbol{b}_{-i}), (s_i', \boldsymbol{s}_{-i}), \theta] \mathbb{P}\left((s_i, \boldsymbol{s}_{-i}), \theta | (b_i', \boldsymbol{b}_{-i})\right),$$

where we made explicit $f_i$'s dependency from the probability distribution $\mathbb{P}(\cdot, \cdot | \boldsymbol{b}) \in \Delta(\mathcal{S} \times \Theta)$. Moreover, we write $f_i(\boldsymbol{x}_i | b_i, s_i, \mathbb{P}) := f_i(\boldsymbol{x}_i | b_i, b_i, s_i, s_i, \mathbb{P})$ to denote the expected payment received by agent $i$ when she is recommended action $b_i \in B_i$, she observes signal $s_i \in S_i$ and she behaves honestly. Then, consider the following LP, which we denote as LP $(\boldsymbol{\zeta}, \Lambda, \varepsilon)$. It is parameterized by $\boldsymbol{\zeta}$, $\Lambda$ and $\varepsilon$, where $\boldsymbol{\zeta} = (\zeta_{\boldsymbol{b}})_{\boldsymbol{b} \in \mathcal{B}}$ is a collection of probability distributions over $\mathcal{S} \times \Theta$, $\Lambda = (\Lambda_1, ..., \Lambda_n)$ with $\Lambda_i : B_i \times B_i \to [-1, 1]$ represent pairwise cost differences, and $\varepsilon > 0$:

$$\max_{\substack{\boldsymbol{x} \succeq 0, \boldsymbol{y} \succeq 0, \\ \boldsymbol{z} \succeq 0, \boldsymbol{\mu} \in \Delta(\mathcal{B})}} \sum_{\boldsymbol{b} \in \mathcal{B}} \sum_{\boldsymbol{s} \in \mathcal{S}} \sum_{\theta \in \Theta} \left[ \sum_{a \in \mathcal{A}} [y[\boldsymbol{b}, \boldsymbol{s}, a] \zeta_{\boldsymbol{b}}[\boldsymbol{s}, \theta] u(a, \theta)] - \sum_{i \in \mathcal{N}} x_i[\boldsymbol{b}, \boldsymbol{s}, \theta] \zeta_{\boldsymbol{b}}[\boldsymbol{s}, \theta] \right] \quad \text{s.t.} \tag{2a}$$

$$\sum_{s_i \in S_i} [f_i(\boldsymbol{x}_i | b_i, s_i, \boldsymbol{\zeta}') - z_i[b_i, b_i', s]] \geq \sum_{\boldsymbol{b}_{-i} \in \mathcal{B}_{-i}} \mu[(b_i, \boldsymbol{b}_{-i})] \Lambda_i(b_i, b_i') - \varepsilon \quad \forall i \in \mathcal{N} \, \forall b_i, b_i' \in B_i \tag{2b}$$

$$z_i[b_i, b_i', s_i] \geq f_i(\boldsymbol{x}_i | b_i, b_i', s_i, s_i', \boldsymbol{\zeta}') \qquad \forall i \in \mathcal{N}, \, \forall b_i, b_i' \in B_i, \, \forall s_i, s_i' \in S_i \tag{2c}$$

$$\sum_{a \in \mathcal{A}} y[\boldsymbol{b}, \boldsymbol{s}, a] = \mu[\boldsymbol{b}] \qquad \forall \boldsymbol{b} \in \mathcal{B}, \, \forall \boldsymbol{s} \in \mathcal{S} \tag{2d}$$

$$x_i[\boldsymbol{b}, \boldsymbol{s}, \theta] \leq M \mu[\boldsymbol{b}] \qquad \forall i \in \mathcal{N}, \, \forall \boldsymbol{b} \in \mathcal{B}, \, \forall \boldsymbol{s} \in \mathcal{S}, \, \forall \theta \in \Theta \tag{2e}$$

Intuitively, constraint (2c) ensures that the auxiliary variable $z_i[b_i, b_i', s_i]$ provides an upper bound on the expected payment that agent $i$ could get through any untruthful signal reporting when she was recommended to play $b_i$, she played $b_i'$ and observes $s_i$. Constraint (2b) exploits the auxiliary variables $\boldsymbol{z} = (\boldsymbol{z}_1, ..., \boldsymbol{z}_n)$ with $\boldsymbol{z}_i \in \mathbb{R}_{\geq 0}^{|B_i| \times |B_i| \times |S_i|}$ to guarantee that no deviation is profitable for agent $i$. The following theorem shows how to recover an optimal correlated mechanism from LP (2)[8].

---

[8]All the proofs omitted from the main paper can be found in the Appendix.

**Theorem 3.1.** *Let $(\boldsymbol{x}^\star, \boldsymbol{y}^\star, \boldsymbol{\mu}^\star, \boldsymbol{z}^\star)$ be an optimal solution to LP $(\mathbb{P}, C, 0)$, where $C = (C_1, ..., C_n)$. Then, let $\boldsymbol{\gamma}^\star = (\boldsymbol{\gamma}_1^\star, ..., \boldsymbol{\gamma}_n^\star)$ and $\boldsymbol{\pi}^\star$ be such that*

$$\gamma_i^\star[\boldsymbol{b}, \boldsymbol{s}, \theta] = \begin{cases} \frac{x_i^\star[\boldsymbol{b}, \boldsymbol{s}, \theta]}{\mu^\star[\boldsymbol{b}]} & \text{if } \mu[\boldsymbol{b}] \neq 0 \\ 0 & \text{otherwise} \end{cases} \qquad \pi^\star[\boldsymbol{b}, \boldsymbol{s}, a] = \begin{cases} \frac{y^\star[\boldsymbol{b}, \boldsymbol{s}, a]}{\mu^\star[\boldsymbol{b}]} & \text{if } \mu[\boldsymbol{b}] \neq 0 \\ \frac{1}{d} & \text{otherwise.} \end{cases}$$

*Then $(\boldsymbol{\mu}^\star, \boldsymbol{\gamma}^\star, \boldsymbol{\pi}^\star)$ is an optimal solution to Problem (1).*

As a consequence of Theorem 3.1, noticing that the linear program has a number of constraints and variables polynomial in $k$, $l$ and $m$, we obtain the following corollary.

**Corollary 3.2.** *An optimal mechanism can be found in polynomial time.*

## 4 CORRELATED VS UNCORRELATED MECHANISMS

Before introducing our online learning problem, we discuss one of the main issues arising from the adoption of correlated mechanism. Indeed, consider the case in which the principal is not able to commit to an IC mechanism, for instance because she is uncertain about the game parameters and thus she is not able to characterize the set of IC mechanisms. When the principal uses non-IC mechanisms it becomes complex to characterize the behavior of the agents. This is because correlated mechanisms introduce *externalities* among the agents. More precisely, since the payments received by agent $i$ depend also on the deviation policies $(\phi_j, \varphi_j) \in \Phi_j$ adopted by agents $j \neq i$, the agents should play an *equilibrium* of the $n$-players game induced by the correlated mechanism. This introduces well-known issues related to both computational complexity and equilibrium selection Daskalakis et al. (2009). Thus, committing to a correlated mechanism which is not IC induces an unpredictable behavior of the agents, which in online settings makes it impossible for the learner to even estimate the game parameters.

To address such drawbacks of correlated mechanisms, we introduce the class of *uncorrelated* mechanisms. An uncorrelated mechanism is composed by an uncorrelated scoring rule $\boldsymbol{\gamma} = (\boldsymbol{\gamma}_1, ..., \boldsymbol{\gamma}_n)$ and an action policy $\boldsymbol{\pi}$. Formally, the set of uncorrelated mechanisms is defined as:

$$\mathcal{U} = \{(\boldsymbol{\gamma}, \boldsymbol{\pi}) \mid \gamma_i : S_i \times \Theta \in [0, M] \ \forall i \in \mathcal{N}, \ \boldsymbol{\pi} : \mathcal{S} \to \Delta(\mathcal{A})\}.$$

Notice that any uncorrelated mechanism can be represented as a correlated mechanism and hence $\mathcal{U} \subset \mathcal{C}$. Differently from correlated mechanisms, any $(\boldsymbol{\gamma}, \boldsymbol{\pi}) \in \mathcal{U}$ induces a well-defined best response for each agent. In particular, the best-response problem can be framed as a single-follower Stackelberg game. Given any $(\boldsymbol{\gamma}, \boldsymbol{\pi}) \in \mathcal{U}$, we define the optimal action $b_i^\circ(\boldsymbol{\gamma}_i) \in B_i$ and the optimal signal reporting policy $\varphi_i^\circ(\cdot|\boldsymbol{\gamma}_i) : S_i \to S_i$ when the principal commits to mechanism $(\boldsymbol{\gamma}, \boldsymbol{\pi})$ as:

$$(b_i^\circ(\boldsymbol{\gamma}_i), \varphi_i^\circ(\cdot|\boldsymbol{\gamma}_i)) \in \arg \max_{\substack{b_i \in B_i \\ \varphi : S_i \to S_i}} \left\{ \sum_{s_i \in S_i} \sum_{\theta \in \Theta} \mathbb{P}^{(i)}(s_i, \theta|b_i) \gamma_i[\varphi(s_i), \theta] - c_i(b_i) \right\},$$

where, as common in the literature, ties are broken in favor of the principal. Therefore, the expected utility of the principal when she commits to mechanism $(\boldsymbol{\gamma}, \boldsymbol{\pi}) \in \mathcal{U}$ is:

$$U^\circ(\boldsymbol{\gamma}, \boldsymbol{\pi}) = \sum_{\boldsymbol{s} \in \mathcal{S}} \sum_{\theta \in \Theta} \mathbb{P}(\boldsymbol{s}, \theta|\boldsymbol{b}^\circ(\boldsymbol{\gamma})) \left[ \left( \sum_{a \in \mathcal{A}} \pi[\varphi^\circ(\boldsymbol{s}|\boldsymbol{\gamma}), a] u(a, \theta) \right) - \sum_{i \in \mathcal{N}} \gamma_i[\varphi_i^\circ(s_i|\boldsymbol{\gamma}_i), \theta] \right],$$

where $\boldsymbol{b}^\circ(\boldsymbol{\gamma}) = (b_1^\circ(\boldsymbol{\gamma}_1), ..., b_n^\circ(\boldsymbol{\gamma}_n))$ and $\varphi^\circ(\boldsymbol{s}|\boldsymbol{\gamma}) = (\varphi_1^\circ(s_1|\boldsymbol{\gamma}_1), ..., \varphi_n^\circ(s_n|\boldsymbol{\gamma}_n))$. Uncorrelated mechanisms eliminate all externalities among the agents, inducing predictable agents' responses. Hence, they are appealing in online settings in which the principal must learn from agents' behavior. We conclude the section showing that despite their advantages that make uncorrelated mechanisms a very useful tool, they can be suboptimal with respect to correlated mechanisms.

**Theorem 4.1.** *There exists a game in which no uncorrelated mechanism is optimal.*

However, while uncorrelated mechanisms are suboptimal in general, there exists realistic classes of games in which optimal mechanisms are uncorrelated as shown by the following theorem.

**Theorem 4.2.** *Assume there for each $i \in \mathcal{N}$, $\theta \in \Theta$ and $b_i \in B_i$, there exists a probability distribution $\psi_i(\cdot|b_i) \in \Delta(S_i)$ such that $\forall \boldsymbol{b} \in \mathcal{B}$, $\forall \boldsymbol{s} \in \mathcal{S}$ and $\forall \theta \in \Theta$, $\mathbb{P}(\boldsymbol{s}, \theta|\boldsymbol{b}) = p[\theta] \prod_{i \in \mathcal{N}} \psi_i(s_i|b_i, \theta)$. Then, there exists a mechanism $(\boldsymbol{\gamma}, \boldsymbol{\pi}) \in \mathcal{U}$ that is optimal among correlated mechanisms.*

## 5 LEARNING THE OPTIMAL MECHANISM

We study an online learning scenario in which the principal interacts for $T$ rounds with $n$ agents without knowing neither the joint probability distribution $\mathbb{P}\left(\boldsymbol{s}, \theta | \boldsymbol{b}\right)$ nor the cost functions $c_i$. At each round $t \in [T]$, the principal publicly announces her mechanism. If the mechanism is correlated, then the agents receive recommendations $\boldsymbol{b}^t \sim \boldsymbol{\mu}$. Then, each $i \in \mathcal{N}$ chooses action $\tilde{b}_i^t$ (possibly different that $b_i^t$) incurring the cost $c_i(\tilde{b}_i^t)$. If the mechanism is uncorrelated the agents play according to the tuple of best responses $\tilde{\boldsymbol{b}}_t = \boldsymbol{b}^\circ(\boldsymbol{\gamma}^t)$. To avoid the issues highlighted in Section 4, we will never employ correlated mechanisms that are *not* IC. Then, a state of nature $\theta^t$ and a signal profile $\boldsymbol{s}^t$ are sampled according to $\mathbb{P}(\boldsymbol{s}, \theta | \tilde{\boldsymbol{b}}^t)$. Each agent $i$ observes signal $s_i$ and reports signal $\tilde{s}_i^t$ to the principal. If the mechanism is correlated, then $\tilde{s}_i^t = s_i$. If the mechanism is uncorrelated, then $\tilde{s}_i^t = \varphi_i^\circ(s_i^t | \boldsymbol{\gamma}_i^t)$. Finally, the principal takes action $a^t \sim \pi[\boldsymbol{b}^t, \tilde{\boldsymbol{s}}^t, \cdot]$ and gets utility $u(a^t, \theta^t)$ while each agent is paid according to the scoring rule. At the end of the round, the feedback received by the learner includes the actions $\tilde{\boldsymbol{b}}^t$ taken by the agents [9], the signals $\tilde{\boldsymbol{s}}^t$ reported by the agents, and the state of nature $\theta^t$, while she is not able to observe the signals $\boldsymbol{s}^t$ that were actually observed by the agents.

The performances of the algorithm are measured in terms of *cumulative regret* $R^T$, which represents the expected loss of utility for the principal due to not having selected the optimal mechanism at each $t \in [T]$. Formally, let $(\boldsymbol{\mu}^\star, \boldsymbol{\gamma}^\star, \boldsymbol{\pi}^\star)$ be an optimal mechanism (*i.e.,* an optimal solution to (1)), and let $T_c, T_u, T_c' \subseteq [T]$ be the sets of rounds in which the principal committed to a correlated and IC mechanism, to an uncorrelated mechanism, and to a correlated and non-IC mechanism, respectively (it holds $T_c \cup T_u \cup T_c' = [T]$). Then, the cumulative regret is defined as:

$$R^T = \sum_{t \in [T]} U(\boldsymbol{\mu}^\star, \boldsymbol{\gamma}^\star, \boldsymbol{\pi}^\star) - \sum_{t \in T_c} U(\boldsymbol{\mu}^t, \boldsymbol{\gamma}^t, \boldsymbol{\pi}^t) - \sum_{t \in T_u} U^\circ(\boldsymbol{\gamma}^t, \boldsymbol{\pi}^t),$$

where, as discussed in Section 4, we used the fact that when the principal commits to a correlated mechanism which is not IC, then she can incur in a constant per-round regret in the worst case, since the behavior of the agents is unpredictable. Our goal is to design an algorithm that achieves $R^T = o(T)$. In the following, we let $\ell > 0$ be the minimum distance between the posteriors induced by two signals, *i.e.,* $\ell = \min_{i \in \mathcal{N}, b_i \in B_i, s_i, s_i' \in S_i} \sum_{\theta \in \Theta} \left(\mathbb{P}^{(i)}\left(\theta | b_i, s_i\right) - \mathbb{P}^{(i)}\left(\theta | b_i, s_i'\right)\right)^2$, and $\iota > 0$ be the minimum probability with which each signal is received by an agent, *i.e.,* $\iota = \min_{i \in \mathcal{N}, b_i \in B_i, s_i \in S_i} \mathbb{P}^{(i)}\left(s_i | b_i\right)$.

### 5.1 ALGORITHM OVERVIEW AND ASSUMPTIONS

For each agent $i \in \mathcal{N}$ and action $b_i \in B_i$, we assume to know an uncorrelated scoring rule strictly incentivizing $i$ to play action $b_i$ while also incentivizing her to report truthfully the observed signal.

**Assumption 1.** For each $i \in \mathcal{N}$, the learner knows a set of scoring rules $\Gamma_i = \{\boldsymbol{\gamma}_i^{b_i} : S_i \times \Theta \in [0, M] \mid b_i \in B_i\}$ and $\rho > 0$ such that

$$\sum_{s_i \in S_i} \sum_{\theta \in \Theta} \left[\mathbb{P}^{(i)}\left(s_i, \theta | b_i\right) \gamma_i^{b_i}[s_i, \theta] - \mathbb{P}^{(i)}\left(s_i, \theta | b_i'\right) \gamma_i^{b_i}[\varphi_i(s_i), \theta]\right] \geq C(b_i, b_i') + \rho \qquad (3)$$

for all $b_i' \in B_i \setminus \{b_i\}$ and $\varphi_i \in S_i \to S_i$, and such that

$$\sum_{\theta \in \Theta} \mathbb{P}^{(i)}\left(\theta | s_i, b_i\right) \left[\gamma_i^{b_i}[s_i, \theta] - \gamma_i^{b_i}[s_i', \theta]\right] \geq 0 \quad \forall s_i, s_i' \in S_i. \qquad (4)$$

Intuitively, Eq. (3) guarantees that for each agent $i$, following the action recommendation is strictly better than deviating to a different action, while Eq. (4) guarantees that reporting the observed signal is never worse in expectation than reporting a different one. This assumption is common in the literature (see, *e.g.*, Chen et al. (2023)), and it is necessary to achieve incentive compatibility under uncertainly. Throughout the remaining of the paper, we let $\Gamma = \{\boldsymbol{\gamma}^{\boldsymbol{b}} := (\gamma_1^{b_1}, ..., \gamma_n^{b_n}) \mid \boldsymbol{b} \in \mathcal{B}\}$.

---

[9]It is common in the literature to assume that agents' actions can be observed (see, *e.g.,* Chen et al. (2023)). Intuitively, this is because the truthful reporting of signals can be used to discriminate between different actions.

Algorithm 1 provides an high-level overview of our algorithm. The procedure is divided in two phases, an *exploration phase* and a *commit phase*. The exploration phase is devoted to finding the estimators $\boldsymbol{\zeta_b} \in \Delta(\mathcal{S} \times \Theta)$ of the joint probabilities $\mathbb{P}(\cdot, \cdot | \boldsymbol{b})$ for each $\boldsymbol{b}$, the estimators $\boldsymbol{\xi}_{b_i,s_i}^{(i)} \in \Delta(\Theta)$ of the posteriors $\mathbb{P}^{(i)}(\cdot | b_i, s_i)$ for each $i \in \mathcal{N}$, $b_i \in B_i$ and

---

**Algorithm 1** Online Information Acquisition

---

**Require:** $T, N_1, N_2, N_3, \rho, \Gamma, \delta$
  ▷ *Exploration phase*
  $(\boldsymbol{\zeta}, \nu, \boldsymbol{\xi}, \varrho) \leftarrow \text{ESTIMATEPROB}(N_1, \Gamma, \delta)$ ▷ *Sec. 6*
  $(\Lambda, \chi) \leftarrow \text{ESTIMATECOSTS}(N_2, N_3, \Gamma, \delta)$ ▷ *Sec. 7*
  ▷ *Commit Phase*
  $\text{COMMIT}(\boldsymbol{\zeta}, \nu, \boldsymbol{\xi}, \varrho, \Lambda, \chi, \Gamma, \rho)$ ▷ *Sec. 8*

---

$s_i \in S_i$, and the estimators $\Lambda_i(b_i, b_i')$ of the cost differences $C_i(b_i, b_i')$ for $i \in \mathcal{N}$, $b_i, b_i' \in B_i$, together with the respective confidence bounds $\nu, \varrho, \chi \in \mathbb{R}_{\geq 0}$. During the commit phase, instead, we leverage the estimates obtained in the previous rounds to output a sequence of IC mechanisms that guarantee sublinear cumulative regret. As inputs to the algorithm we provide the total number of rounds $T$, the minimum number of rounds $N_1, N_2, N_3$ that regulate the length of the exploration phase, the scoring rules $\Gamma$, the scalar $\rho$ described in Assumption 1, and the desired confidence level $\delta \in (0,1)$ on the regret bound. We provide a description of the three algorithms ESTIMATEPROB, ESTIMATECOSTS and COMMIT in Sections 6, 7 and 8, respectively. The guarantees of Algorithm 1 are stated in the following theorem.

**Theorem 5.1.** *Let $\kappa = \frac{289}{2} m^2 \ln(12|\mathcal{B}||\mathcal{S}|Tmn/\delta) \frac{1}{\iota^2 \ell^2}$. For any $\delta \in (0,1)$, with probability at least $1 - \delta$, running Algorithm 1 with $N_1 = N_3 = T^{2/3}$ and $N_2 = \log(T)$ guarantees*

$$R^T \leq \widetilde{\mathcal{O}}\left(\frac{M^3}{\rho\ell}|\mathcal{B}||\mathcal{S}|mnk^3l^2\sqrt{\ln(1/\delta)}\max\{T^{2/3}, \kappa\}\right)$$

The upper bound on $R^T$ presents a term $\max\{T^{2/3}, \kappa\}$. For $T$ sufficiently large, $\kappa$ –that depends on the instance and *logarithmically* on $T$– is dominated by $T^{2/3}$ and we recover the $\widetilde{\mathcal{O}}(T^{2/3})$ bound[10].

## 6 ESTIMATION OF THE PROBABILITY DISTRIBUTIONS

The estimation phase is devoted to the estimation of the joint probabilities $\mathbb{P}(\cdot, \cdot | \boldsymbol{b})$ and of the posteriors $\mathbb{P}^{(i)}(\cdot | s_i, b_i)$ induced by action-signal couples. Let $\mathcal{T}_p \subseteq T$ be the set of rounds devoted to ESTIMATEPROB. Furthermore, for $\boldsymbol{b} \in \mathcal{B}$, $b_i \in B_i$ and $s_i \in S_i$, let $\mathcal{T}_p(\boldsymbol{b}) = \{t \in \mathcal{T}_p \mid \boldsymbol{b}^t = \boldsymbol{b}\}$ and $\mathcal{T}_p^{(i)}(b_i, s_i) = \{t \in \mathcal{T}_p \mid \tilde{s}_i^t = s_i, b_i^t = b_i\}$. For $K = 6|\mathcal{B}|T|\mathcal{S}|nm$, we introduce estimators $\boldsymbol{\zeta_b} \in \Delta(\mathcal{S} \times \Theta)$ and $\boldsymbol{\xi}_{b_i,s_i}^{(i)} \in \Delta(\Theta)$ with their confidence bounds $\nu_{\boldsymbol{b}}$ and $\varrho_{b_i,s_i}^{(i)}$, defined as[11]:

$$\zeta_{\boldsymbol{b}}[\boldsymbol{s}, \theta] = \frac{1}{|\mathcal{T}_p(\boldsymbol{b})|} \sum_{t \in \mathcal{T}_p(\boldsymbol{b})} \mathbb{1}\left[\tilde{\boldsymbol{s}}^t = \boldsymbol{s}, \theta^t = \theta\right], \quad \nu_{\boldsymbol{b}} = \sqrt{\frac{\ln(2K/\delta)}{2|\mathcal{T}_p(\boldsymbol{b})|}}, \quad \forall \boldsymbol{b}, \boldsymbol{s}, \theta \tag{5}$$

and

$$\xi_{b_i,s_i}^{(i)}[\theta] = \frac{1}{|\mathcal{T}_p^{(i)}(b_i, s_i)|} \sum_{t \in \mathcal{T}_p^{(i)}(b_i,s_i)} \mathbb{1}\left[\theta^t = \theta\right] \quad \varrho_{b_i,s_i}^{(i)} = \sqrt{\frac{\ln(2K/\delta)}{2|\mathcal{T}_p^{(i)}(b_i, s_i)|}} \quad \forall i, b_i, s_i, \theta. \tag{6}$$

The procedure for obtaining such estimators is described in Algorithm 2. It leverages the knowledge of the scoring rules in Assumption 1 to guarantee a sufficient number of samples for each probability distribution that we want to estimate. In particular, the algorithm iterates over all $\boldsymbol{b} \in \mathcal{B}$ and commits to scoring rule $\boldsymbol{\gamma^b} \in \Gamma$ for at least $N_1$ rounds and until a specific condition is met. Committing to $\boldsymbol{\gamma^b}$ guarantees that the agents are incentivized to play $\boldsymbol{b}$ (Eq. 3) and that they report the received signal (Eq. 4). This ensures that the feedback received is reliable for estimating both probability distributions. The condition $\bar{\varrho} \leq \underline{d}/13m$ on the confidence bounds guarantees that we have collected enough samples to estimate each posterior distribution. The required precision depends on the instance parameters $\iota$ and $\ell$ and will be fundamental to design approximately optimal mechanism that are IC (see Section 8). To formalize the guarantees of Algorithm 2, we introduce the *clean event* $\mathcal{E}_p$.

---

[10]We remark that our algorithm does *not* need to know $\ell$ and $\iota$ in advance, but implicitly estimates them during the execution.

[11]In this work, we denote as $\mathbb{1}[\cdot]$ the indicator function.

**Definition 6.1** (Clean event for probability estimation). Let $\kappa := 289 \ln(2K/\delta) m^2 / (2\iota^2 \ell^2)$. Let $\nu := \max_{\boldsymbol{b} \in \mathcal{B}} \nu_{\boldsymbol{b}}$ and $\varrho := \max_{i \in \mathcal{N}, b_i \in B_i, s_i \in S_i} \varrho^{(i)}_{b_i, s_i}$ The clean event $\mathcal{E}_p$ holds if, for all $t \in \mathcal{T}_p$ it holds that:

$$|\zeta_{\boldsymbol{b}}[\boldsymbol{s}, \theta] - \mathbb{P}(\boldsymbol{s}, \theta | \boldsymbol{b})| \leq \nu \quad \forall \boldsymbol{b}, \boldsymbol{s}, \theta, \qquad |\xi^{(i)}_{b_i, s_i}[\theta] - \mathbb{P}^{(i)}(\theta | b_i, s_i)| \leq \varrho \quad \forall i, b_i, s_i, \theta$$

and if, whenever $|\mathcal{T}_p(\boldsymbol{b})| \geq \kappa$, it holds that

$$|\mathcal{T}^{(i)}_p(b_i, s_i)| \geq \frac{1}{2} \iota |\mathcal{T}_p(\boldsymbol{b})| \quad \forall i \in \mathcal{N}, \forall \boldsymbol{b} \in \mathcal{B}, \forall s_i \in S_i,$$

Using standard concentration arguments, it is possible to show the following.

**Lemma 6.1.** *The clean event $\mathcal{E}_p$ holds with probability at least $1 - \frac{\delta}{2}$.*

Before concluding this section, we point out that the number of samples $|\mathcal{T}_p(\boldsymbol{b})|$ needed for each $\boldsymbol{b} \in \mathcal{B}$ to have $\bar{\varrho} \leq \underline{d}/13m$ depends on the value of the parameters $\ell$ and $\iota$. Indeed, the smaller are the minimum signal probability $\iota$ and the minimum posteriors distance $\ell$, the higher is the number of samples needed to have an accurate estimation and to satisfy the above condition. However, setting $N_1 = T^{2/3}$, the number of samples needed to satisfy the two terminating conditions becomes dominated by $N_1$ in non-degenerate instances in which $T$ is sufficiently large. Formally,

**Lemma 6.2.** *Assume the clean event $\mathcal{E}_p$ holds. Then, at the end of the execution of* ESTIMATEPROB, *for each $\boldsymbol{b} \in \mathcal{B}$ it holds that $|\mathcal{T}_p(\boldsymbol{b})| \leq \max\{N_1, \kappa\}$.*

---

**Algorithm 2** ESTIMATEPROB

**Require:** $N_1, \Gamma, \delta$
  $\mathcal{T}_p(\boldsymbol{b}) \leftarrow \{\varnothing\} \quad \forall \boldsymbol{b}$
  $\mathcal{T}^{(i)}_p(b_i, s_i) \leftarrow \{\varnothing\} \quad \forall i, b_i, s_i$
  $\boldsymbol{\pi}[\boldsymbol{s}, a] = 1/d \quad \forall \boldsymbol{s}, a$
  **for** $b \in \mathcal{B}$ **do**
    **while** $|\mathcal{T}(\boldsymbol{b})| < N_1 \vee \bar{\varrho} > \frac{d}{13m}$ **do**
      Select $(\boldsymbol{\gamma}^{\boldsymbol{b}}, \boldsymbol{\pi})$ and observe $\tilde{\boldsymbol{s}}^t, \theta^t$
      $\mathcal{T}_p(\boldsymbol{b}) \leftarrow \mathcal{T}_p(\boldsymbol{b}) \cup \{t\}$
      $\mathcal{T}^{(i)}_p(b_i, \tilde{s}^t_i) \leftarrow T^{(i)}_p(b_i, \tilde{s}^t_i) \cup \{t\} \; \forall i$
      Update $(\zeta_{\boldsymbol{b}}, \nu_{\boldsymbol{b}})$ as in Eq. (5)
      Update $(\xi^i_{b_i, \tilde{s}^t_i}, \varrho^{(i)}_{b_i, \tilde{s}^t_i})$ as in Eq. (6) $\forall i$
      $\bar{\varrho} \leftarrow \max_{i, b_i, s_i} \varrho^{(i)}_{b_i, s_i}$
      $\underline{d} \leftarrow \min_{i, s_i, s'_i} \|\boldsymbol{\xi}^{(i)}_{b_i, s_i} - \boldsymbol{\xi}^{(i)}_{b_i, s'_i}\|^2_2$
    **end while**
  **end for**
  **return** $(\boldsymbol{\zeta}_{\boldsymbol{b}})_{\boldsymbol{b}}, \nu, (\boldsymbol{\xi}^{(i)}_{b_i, s_i})_{i, b_i, s_i}, \varrho$

---

## 7 ESTIMATION OF THE COST DIFFERENCES

The second phase aims at obtaining high-confidence bounds for the cost differences $C_i(b_i, b'_i)$ for $i \in \mathcal{N}$, $b_i, b'_i \in B_i$. For each agent $i \in \mathcal{N}$, the algorithm explores each pair $b_i, b'_i \in B_i$ and executes a binary search (BS) routine in order to estimate $C_i(b_i, b'_i)$, leveraging the knowledge of the uncorrelated scoring rules $\gamma^{b_i}_i, \gamma^{b'_i}_i \in \Gamma_i$. In particular, playing convex combinations of the two scoring rules for $N_2$ rounds, the algorithm finds two scoring rules $\gamma_i$ and $\gamma'_i$ such that (i) $\|\gamma_i - \gamma'_i\|_\infty \leq M/2^{N_2}$, (ii) $\gamma_i$ incentivizes $b_i$ over $b'_i$, and (iii) $\gamma'_i$ incentivizes $b'_i$ over $b_i$. Then, the algorithm estimates for $N_3$ rounds the expected payments received by agent $i$ under scoring rules $\gamma_i$ and $\gamma'_i$ and uses such estimates, together with the bound on $\|\gamma_i - \gamma'_i\|_\infty$, to obtain the estimator $\Lambda_i(b_i, b'_i)$ and the confidence bound $\chi_i[b_i, b'_i]$. However, it might happen that the BS routine ends before finding such scoring rules, thus requiring a recursive execution of the algorithm. Due to space constraints, the study of those cases, as well as a more thorough description of ESTIMATECOSTS, are deferred to Appendix B. To formalize the theoretical guarantees of ESTIMATECOSTS, we need the following clean event.

**Definition 7.1** (Clean event for cost estimation). Let $\chi = \max_{i, b_i, b'_i} \chi_i[b_i, b'_i]$. The clean event $\mathcal{E}_c$ holds if at the end of the execution of ESTIMATECOSTS:

$$\Lambda_i(b_i, b'_i) - \chi \leq C_i(b_i, b'_i) \leq \Lambda_i(b_i, b'_i) + \chi \quad \forall i \in \mathcal{N}, \forall b_i, b'_i \in B_i,$$

Then, it is possible to provide a lower bound on the probability with which $\mathcal{E}_c$ is verified.

**Lemma 7.1.** *The clean event $\mathcal{E}_c$ holds with probability at least $1 - \frac{\delta}{2}$.*

Furthermore, to conclude this section, we show how the number of rounds used by ESTIMATECOSTS varies as a function of $N_2$ and $N_3$.

**Lemma 7.2.** *Let $\mathcal{T}_d$ be the set of rounds devoted to the execution of* ESTIMATECOSTS. *Then it holds that $|\mathcal{T}_d| \leq nk^3 l^2 (N_2 + N_3)$.*

# 8 COMMIT PHASE

In the commit phase, the algorithm exploits the estimations of the probability distributions and the pairwise cost differences. Here, the learner selects a sequence of mechanisms $(\boldsymbol{\mu}^t, \boldsymbol{\gamma}^t, \boldsymbol{\pi}^t) \in \mathcal{C}$ that pursues a twofold objective: the minimization of the regret $R^T$ and the satisfaction of the IC constraints. To find a regret minimizing mechanism, we first obtain a mechanism $(\boldsymbol{\mu}^t, \tilde{\boldsymbol{\gamma}}^t, \boldsymbol{\pi}^t)$ from an optimal solution to LP$(\boldsymbol{\zeta}, \Lambda, \varepsilon)$ through the function GETMECH-ANISM as specified by Theorem 3.1, where $\varepsilon = 2M|\mathcal{S}|m\nu + \chi$. The parameters are chosen to guarantee that, assuming clean events $\mathcal{E}_c$ and

---

**Algorithm 3** COMMIT

**Require:** $\boldsymbol{\zeta}, \nu, \boldsymbol{\xi}, \varrho, \Lambda, \chi, \Gamma, \rho$
   ▷ *Construction of IC mechanism*
  $\varepsilon \leftarrow 2M|\mathcal{S}|m\nu + \chi$
  $(\tilde{\boldsymbol{x}}^t, \tilde{\boldsymbol{y}}^t, \tilde{\boldsymbol{\mu}}^t, \tilde{\boldsymbol{z}}^t) \leftarrow$ Opt. solution to LP$(\boldsymbol{\zeta}, \Lambda, \varepsilon)$
  $(\boldsymbol{\mu}^t, \tilde{\boldsymbol{\gamma}}^t, \boldsymbol{\pi}^t) \leftarrow$ GETMECHANISM$(\tilde{\boldsymbol{x}}^t, \tilde{\boldsymbol{y}}^t, \tilde{\boldsymbol{\mu}}^t)$
  Define $\boldsymbol{\gamma}_i^t$ as in Eq. (8) $\forall i \in \mathcal{N}$.
   ▷ *Commit rounds*
  **while** $t \le T$ **do**
    Commit to mechanism $(\boldsymbol{\mu}^t, \boldsymbol{\gamma}^t, \boldsymbol{\pi}^t)$
  **end while**

---

$\mathcal{E}_p$ hold, the optimal mechanism $(\boldsymbol{\mu}^\star, \boldsymbol{\gamma}^\star, \boldsymbol{\pi}^\star)$ is in the feasibility set. Then, since the objective function of LP$(\boldsymbol{\zeta}, \Lambda, \varepsilon)$ and the one of LP$(\mathbb{P}, C, 0)$ have close values, it follows that –assuming that all the agents behave truthfully– mechanism $(\boldsymbol{\mu}^t, \tilde{\boldsymbol{\gamma}}^t, \boldsymbol{\pi}^t)$ guarantees a vanishing per-round regret w.r.t. $(\boldsymbol{\mu}^\star, \boldsymbol{\gamma}^\star, \boldsymbol{\pi}^\star)$. However, the following lemma shows that $(\boldsymbol{\mu}^t, \tilde{\boldsymbol{\gamma}}^t, \boldsymbol{\pi}^t)$ is not IC and hence the agents are not incentivized to behave truthfully.

**Lemma 8.1.** *Assume clean events $\mathcal{E}_c$ and $\mathcal{E}_p$ hold. Let $(\boldsymbol{\mu}, \boldsymbol{\gamma}, \boldsymbol{\pi})$ be an optimal solution to LP$(\boldsymbol{\zeta}, \Lambda, \varepsilon)$, where $\varepsilon = 2M|\mathcal{S}|m\nu + \chi$. Then, letting $\lambda = 2M|\mathcal{S}|m(k+1)(\nu+\chi)$, it holds that:*

$$F_i(\boldsymbol{\mu}, \boldsymbol{\gamma}) - F_i^{\phi, \varphi}(\boldsymbol{\mu}, \boldsymbol{\gamma}) \ge \sum_{\boldsymbol{b} \in \mathcal{B}} \tilde{\mu}[\boldsymbol{b}] C_i(b_i, \phi(b_i)) - \lambda \qquad \forall i \in \mathcal{N}, \forall (\phi, \varphi) \in \Phi_i.$$

We recall that, in light of the discussion carried out in Section 4, committing to a non-IC mechanism yields an unpredictable response of the agents which can induce a constant per-round regret in the worst case. Thus, we provide a modification of $(\boldsymbol{\mu}^t, \tilde{\boldsymbol{\gamma}}^t, \boldsymbol{\pi}^t)$ that makes the mechanism IC, while maintaining vanishing per-round regret w.r.t. the optimal mechanism. To do so, we exploit the posterior estimates $\boldsymbol{\xi}$ obtained during the estimation phase, as well as the scoring rules $\Gamma$ described in Assumption 1. In particular, let $\hat{\ell} = \min_{i, b_i, s_i, s_i'} ||\boldsymbol{\xi}_{b_i, s_i}^{(i)} - \boldsymbol{\xi}_{b_i, s_i'}^{(i)}||_2^2$, $\bar{\ell} = \hat{\ell} + 4m\varrho$ and $\underline{\ell} = \hat{\ell} - 4m\varrho$. We define two coefficients $\alpha := (\rho\underline{\ell})/(\rho\bar{\ell} + 65\lambda)$ and $\beta := (45 + \bar{\ell})/(18\rho + 45 + \bar{\ell})$. Moreover, for each agent $i$ we define the uncorrelated scoring rules $(\hat{\boldsymbol{\gamma}}_i^{b_i})_{b_i \in B_i}$ such that

$$\hat{\gamma}_i^{b_i}[s_i, \theta] = \xi_{b_i, s_i}^{(i)}[\theta] + H_i - \frac{1}{2}||\boldsymbol{\xi}_{b_i, s_i}^{(i)}||_2^2, \quad \forall s_i \in S_i, \forall \theta \in \Theta, \tag{7}$$

where $H_i = \max_{i, b_i, s_i} \frac{1}{2}||\boldsymbol{\xi}_{b_i, s_i}^{(i)}||_2^2$, and the correlated scoring rule $\boldsymbol{\gamma}_i^t$ such that:

$$\gamma_i^t[\boldsymbol{b}, \boldsymbol{s}, \theta] = \alpha \tilde{\gamma}_i^t[\boldsymbol{b}, \boldsymbol{s}, \theta] + (1-\alpha) \left[ \beta \gamma_i^{b_i}[s_i, \theta] + (1-\beta) \hat{\gamma}_i^{b_i}[s_i, \theta] \right] \quad \forall \boldsymbol{b} \in \mathcal{B} \, \forall \boldsymbol{s} \in \mathcal{S}, \forall \theta \in \Theta. \tag{8}$$

The correlated scoring rule $\boldsymbol{\gamma}_i^t$ is a convex combination of the correlated scoring rule $\tilde{\boldsymbol{\gamma}}_i^t$ and two uncorrelated scoring rules $\gamma_i^{b_i}$ and $\hat{\gamma}_i^{b_i}$. The latter two compensate the violation of the IC constraints of $\tilde{\gamma}_i^t$, as shown in Lemma 8.1. In particular, scoring rules $\hat{\gamma}_i^{b_i}$ are designed to *strictly* incentivize agents' truthful reporting. [12]. We conclude proving that $(\boldsymbol{\mu}^t, \boldsymbol{\gamma}^t, \boldsymbol{\pi}^t)$ is indeed an IC mechanism.

**Lemma 8.2.** *Assume clean events $\mathcal{E}_c$, $\mathcal{E}_p$ hold. If mechanism $(\boldsymbol{\mu}^t, \boldsymbol{\gamma}^t, \boldsymbol{\pi}^t)$ is chosen according to Algorithm 3, then it is IC,* i.e., *if satisfies Equation (1b) of optimization problem (1).*

Lemma 8.2 provides the formal guarantees on the incentive compatibility of $(\boldsymbol{\mu}^t, \boldsymbol{\gamma}^t, \boldsymbol{\pi}^t)$. By noticing that the parameter $\alpha$ is chosen so to balance correctly the regret minimizing scoring rule $\tilde{\boldsymbol{\gamma}}^t$ and the other two uncorrelated scoring rules, we can recover sublinear regret during the commit phase. We refer the reader to the proof of Theorem 5.1 for the technical details on this aspect.

---

[12]We remark that also scoring rules in $\Gamma$ incentivize such truthful reporting (see Eq. (4)), but not in a strict way. Thus, scoring rules $\hat{\gamma}_i^{b_i}$ is necessary to compensate the IC violations of $\tilde{\boldsymbol{\gamma}}^t$.

## ACKNOWLEDGMENTS

This paper is supported by the FAIR (Future Artificial Intelligence Research) project, funded by the NextGenerationEU program within the PNRR-PE-AI scheme (M4C2, Investment 1.3, Line on Artificial Intelligence), and by the EU Horizon project ELIAS (European Lighthouse of AI for Sustainability, No. 101120237).

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

## A    RELATED WORKS

The study of information acquisition has been mostly confined to economics. However, in recent years, computational problems related to information acquisition have received increasing attention. Several works have focused on optimization problems related to the computation of scoring rules (Chen & Yu, 2021; Papireddygari & Waggoner, 2022; Li et al., 2022; Neyman et al., 2021). The closest to our work is (Chen et al., 2023), which studies an online learning problem in which a principal acquires information from a single agent. Chen et al. (2023) provide a $\widetilde{\mathcal{O}}(T^{2/3})$ regret bound. To the best of our knowledge, ours is the first computational work that considers scoring rules with multiple agents. In the online multi-agent information acquisition problem, (Cacciamani et al., 2023) is the closest work. Cacciamani et al. (2023) study an online mechanism design problem *without money* in which the agents' utilities depend on the principal's action and the principal tries to incentivize the agents to report the information playing agent-favorable actions. The information acquisition problem is also related to the principal-agent problem, a.k.a. contract design, in which a principal designs an outcome-dependent payment scheme to incentivize an agent to play a hidden action (Babaioff et al., 2012; Guruganesh et al., 2021; Alon et al., 2021; Castiglioni et al., 2022b;a; Dütting et al., 2019; Dutting et al., 2021). A recent work that is closely related to ours is Zhu et al. (2023), which studies the repeated interaction between a principal and an agent, providing sublinear regret bounds. Finally, our online information acquisition problem is related to Bayesian persuasion in online settings (Castiglioni et al., 2021; 2020; 2023; Wu et al., 2022; Bernasconi et al., 2023). The works closest to ours consider online problems in which the agents do not know the prior over the states of nature. Zu et al. (2021) studies an online persuasion problem in which the sender and the receiver do not know the prior. Bernasconi et al. (2022) extend the analysis to sequential games.

## B    MORE DETAILS ON COSTS ESTIMATION

---

**Algorithm 4** ESTIMATECOSTS

---

**Require:** $N_2, N_3, \Gamma, \delta$
   **for** $i \in \mathcal{N}$ **do**
      **for** $b_i, b_i' \in B_i \times B_i'$ **do**
         $\gamma_i \leftarrow \gamma_i^{b_i} \in \Gamma_i$
         $\gamma_i' \leftarrow \gamma_i^{b_i'} \in \Gamma_i$
         $\Lambda_i(b_i, b_i'), \chi_i[b_i, b_i'] \leftarrow \mathrm{BS}(i, b_i, b_i', \gamma_i, \gamma_i', N_2, N_3, \delta)$
      **end for**
   **end for**
   **return** $(\Lambda_i)_{i \in \mathcal{N}}, \max_{i, b_i, b_i'} \chi_i[b_i, b_i']$

---

In this section we exhaustively describe the algorithm for cost estimation. The pseudocode for the main algorithm is presented in Algorithm 4. For each agent $i \in \mathcal{N}$, the algorithm iteratively explores each pair $b_i, b_i' \in B_i$ to obtain an estimate $\Lambda_i(b_i, b_i')$ of the cost difference $C_i(b_i, b_i')$ as well as an high confidence bound $\chi_i[b_i, b_i']$ such that

$$\Lambda_i(b_i, b_i') - \chi_i[b_i, b_i'] \leq C_i(b_i, b_i') \leq \Lambda_i(b_i, b_i') + \chi_i[b_i, b_i'].$$

To this extent, we leverage the knowledge of scoring rules $\Gamma_i$ described in Assumption 1 and use them as starting points of a binary search routine that is described below. As input to the Algorithm, we provide the number of rounds $N_2, N_3$ that regulate the amount of rounds used by the binary search routine, the scoring rules $\Gamma$ described in Assumtpion 1 and the desired confidence level $\delta$.

### B.1    BINARY SEARCH

The pseudocode for BS is presented in Algorithm 5. For clarity, we fix an agent $i \in \mathcal{N}$ and two actions $b_i, b_i' \in B_i$ and describe the execution of the BS algorithm for estimation of cost difference $C_i(b_i, b_i')$. Intuitively, the binary search routine between the two actions $b_i, b_i' \in B_i$ is structured in three distinct phases. The first phase is a *search phase* that implements a search over the space of scoring rules to obtain two uncorrelated scoring rules $\gamma_i \, \gamma_i' : S_i \times \Theta \to [0, M]$ with appropriately

---

**Algorithm 5** Binary Search (BS)

---

**Require:** $i, b_i, b_i', \boldsymbol{\gamma}_i, \boldsymbol{\gamma}_i', N_2, N_3, \delta$
  $\boldsymbol{\pi}[\boldsymbol{s}, a] = 1/d \quad \forall \boldsymbol{s} \in \mathcal{S}, \forall a \in \mathcal{A}$
  ▷ *Search phase*
  **for** $t \in [N_2]$ **do**
    $\bar{\boldsymbol{\gamma}}_i \leftarrow \frac{1}{2}\boldsymbol{\gamma}_i + \frac{1}{2}\boldsymbol{\gamma}_i'$
    Commit to $(\bar{\boldsymbol{\gamma}}, \boldsymbol{\pi})$ and observe $\tilde{b}_i^t$
    **if** $\tilde{b}_i^t = b_i$ **then**
      $\boldsymbol{\gamma}_i \leftarrow \bar{\boldsymbol{\gamma}}_i$
    **else if** $\tilde{b}_i^t = b_i'$ **then**
      $\boldsymbol{\gamma}_i' \leftarrow \bar{\boldsymbol{\gamma}}_i$
    **else**
      ▷ *Split phase*
      $x_1, y_1 \leftarrow \text{BS}(i, b_i, \tilde{b}_i^t, \boldsymbol{\gamma}, \bar{\boldsymbol{\gamma}}, N_2, N_3, \mathcal{V}, \Lambda_i, \boldsymbol{\chi}_i, \delta)$
      $x_2, y_2 \leftarrow \text{BS}(i, \tilde{b}_i^t, b_i', \bar{\boldsymbol{\gamma}}, \boldsymbol{\gamma}', N_2, N_3, \mathcal{V}, \Lambda_i, \boldsymbol{\chi}_i, \delta)$
      **return** $x_1 + x_2, y_1 + y_2$
    **end if**
  **end for**
  ▷ *Payment estimation phase*
  Commit for $N_3$ rounds to $(\boldsymbol{\gamma}, \boldsymbol{\pi})$ and observe $\gamma[\tilde{s}_i^t, \theta^t]$ for $t \in [N_3]$
  Commit for $N_3$ rounds to $(\boldsymbol{\gamma}', \boldsymbol{\pi})$ and observe $\gamma'[\tilde{s}_i^t, \theta^t]$ for $t \in [N_3]$
  $x_1, y_1 \leftarrow$ compute as in Equation (14)
  **return** $x_1, y_1$

---

bounded distance between them such that $\boldsymbol{\gamma}_i$ incentivizes[13] action $b_i$ and $\boldsymbol{\gamma}_i'$ incentivizes action $b_i'$. The second phase, *i.e.,* the *payment estimation phase*, estimates the expected payment received by agent $i$ when the principal commits to the two scoring rules $\boldsymbol{\gamma}_i$ and $\boldsymbol{\gamma}_i'$ and uses such estimates to recover an high-confidence interval for the cost difference $C_i(b_i, b_i')$. The third phase is called *split phase* and is needed to manage all those cases in which the scoring rules found during the search phase are not adequate to recover meaningful upper and lower bounds for the cost difference.

The algorithm takes as input the agent $i$, the actions $b_i, b_i'$, two uncorrelated scoring rules $\boldsymbol{\gamma}_i, \boldsymbol{\gamma}_i' :$ $S_i \times \Theta \to [0, M]$ that incentivize action $b_i$ and $b_i'$, respectively, and the desired confidence level $\delta$. To formalize the characteristics of scoring rules $\boldsymbol{\gamma}_i, \boldsymbol{\gamma}_i'$, we define the maximum expected payment received by agent $i$ when the principal commits to an uncorrelated scoring rule $\boldsymbol{\gamma}_i$ and agent $i$ plays action $b_i \in B_i$, using a best-response policy for signal reporting:

$$F_i^{\circ}(\boldsymbol{\gamma}_i | b_i) = \max_{\varphi: S_i \to S_i} \left\{ \sum_{s_i \in S_i} \sum_{\theta \in \Theta} \mathbb{P}^{(i)}(s_i, \theta | b_i) \, \boldsymbol{\gamma}_i[\varphi(s_i), \theta] \right\}.$$

Then, if $\boldsymbol{\gamma}_i$ incentivizes action $b_i$ and $\boldsymbol{\gamma}_i'$ incentivizes action $b_i'$, by noticing that $C_i(b_i, b_i') = -C_i(b_i', b_i)$, we get that

$$F_i^{\circ}(\boldsymbol{\gamma}_i | b_i) - F_i^{\circ}(\boldsymbol{\gamma}_i | b_i') \geq C_i(b_i, b_i') \tag{9a}$$

$$F_i^{\circ}(\boldsymbol{\gamma}_i' | b_i) - F_i^{\circ}(\boldsymbol{\gamma}_i' | b_i') \leq C_i(b_i, b_i'). \tag{9b}$$

Equation (9) provides a rather natural way of finding upper and lower bounds to $C_i(b_i, b_i')$. We will leverage this idea to obtain an estimator for $C_i(b_i, b_i')$. In the following we distinctly analyze each of the three phases of the algorithm.

**Search phase.** The search phase for agent $i$ between two actions $b_i, b_i' \in B_i$ aims at finding two uncorrelated scoring rules $\boldsymbol{\gamma}_i, \boldsymbol{\gamma}_i'$ that satisfy Equations (9a) and (9b), while also guaranteeing that $||\boldsymbol{\gamma}_i - \boldsymbol{\gamma}_i'||_\infty \leq \eta$ for some $\eta > 0$. To do so, for $N_2$ rounds we interpolate between the two scoring rules and iteratively update them using the feedback received from the online interaction. In particular, let $\mathcal{T}_{d,s}^{(i)}(b_i, b_i')$ be the set of rounds used by the search phase, when we want to estimate

---

[13]Recall that an uncorrelated scoring rule $\boldsymbol{\gamma}_i$ incentivizes an action $b_i$ if $b_i^{\circ}(\boldsymbol{\gamma}_i) = b_i$.

the cost difference $C_i(b_i, b'_i)$. Then, at each $t \in \mathcal{T}^{(i)}_{d,s}(b_i, b'_i)$, the principal commits to scoring rule $\bar{\gamma}_i = \frac{1}{2}\gamma_i + \frac{1}{2}\gamma'_i$ and observes the action $\tilde{b}^t_i = b^\circ_i(\bar{\gamma}_i)$ played by agent $i$. If $\tilde{b}^t_i = b_i$, the scoring rules $\bar{\gamma}_i$ incentivizes action $b_i$ and thus, we proceed by updating $\gamma_i = \bar{\gamma}_i$. Similarly, if $\tilde{b}^t_i = b'_i$, then the scoring rule $\bar{\gamma}_i$ incentivizes action $b'_i$ and we update $\gamma'_i = \bar{\gamma}_i$. If, instead $\tilde{b}^t_i \neq b_i, b'_i$, we enter the so-called *split phase*, which, for clarity, we describe next. If we never enter the split phase, the search phase ends after $N_2$ rounds (*i.e.,* when $|T^{(i)}_{d,s}(b_i, b'_i)| = N_2$). At this point, scoring rules $\gamma_i$, $\gamma'_i$ satisfy Equations (9a) and (9b), since the feedback received from the environment guarantees that $b^\circ_i(\gamma_i) = b_i$ and $b^\circ_i(\gamma'_i) = b'_i$. Furthermore, we can show the following bound on the distance between the two scoring rules.

**Lemma B.1.** *For any agent $i \in \mathcal{N}$ and actions $b_i, b'_i \in B_i$, assume the search phase is completed without entering into the split phase. Let $\gamma_i, \gamma'_i$ be the scoring rules obtained at the end of the search phase. Then the following holds:*

$$||\gamma_i - \gamma'_i||_\infty \leq \frac{M}{2^{N_2}}.$$

*Proof.* Fix agent $i$ and actions $b_i, b'_i \in B_i$. Let $d^t$ be the distance between scoring rules $\gamma_i$ and $\gamma'_i$ at iteration $t \in T^{(i)}_{d,s}(b_i, b'_i)$. With an abuse of notation, we denote as $d^0$ the distance between the two scoring rules at the beginning of the search phase. Trivially, it holds $d^0 \leq M$. Assume without loss of generality that $\tilde{b}^t_i = b'_i$. Then, we have that

$$d^{t+1} = ||\gamma_i - \bar{\gamma}_i||_\infty = ||\gamma_i - \frac{1}{2}\gamma_i - \frac{1}{2}\gamma'_i||_\infty = \frac{1}{2}||\gamma_i - \gamma'_i||_\infty = \frac{1}{2}d^t. \tag{10}$$

Thus, by recalling that if we do not enter the split phase, then $|T^{(i)}_{d,s}(b_i, b'_i)| = N_2$, we can iteratively apply Equation (10) to bound $d^{N_2}$ and obtain the result. $\square$

After the completion of the search phase, the algorithm enters into the payment estimation phase, in which we leverage knowledge of the scoring rules $\gamma_i, \gamma'_i$ to recover an estimate of $C_i(b_i, b'_i)$.

**Payment estimation phase.** The payment estimation phase starts after the completion of the search phase. In this phase we aim at finding estimates of the expected payments $F^\circ_i(\gamma_i|b_i)$ and $F^\circ_i(\gamma'_i|b'_i)$ and then use those to obtain an high-confidence bound for $C_i(b_i, b'_i)$, leveraging both the properties of $\gamma_i$ and $\gamma'_i$ mentioned above and Equation (9). In particular, notice that when the principal commits to uncorrelated scoring rule $\gamma_i$, the payment $\gamma_i[\tilde{s}^t_i, \theta^t]$ is an unbiased estimate of $F^\circ_i(\gamma_i|b_i)$ since:

$$\mathbb{E}[\gamma_i[\tilde{s}^t_i, \theta^t]] = \sum_{s_i \in S_i} \sum_{\theta \in \Theta} \mathbb{P}^{(i)}(s_i, \theta | b^\circ(\gamma_i)) \, \gamma_i[\varphi^\circ_i(s_i|\gamma_i), \theta]$$

$$= \max_{\varphi: S_i \to S_i} \left\{ \sum_{s_i \in S_i} \sum_{\theta \in \Theta} \mathbb{P}^{(i)}(s_i, \theta | b_i) \, \gamma_i[\varphi^\circ_i(s_i|\gamma_i), \theta] \right\}$$

$$= F^\circ_i(\gamma_i|b_i).$$

Similarly, when the principal commits to scoring rule $\gamma'_i$ it holds that $\mathbb{E}[\gamma'_i[\tilde{s}^t_i, \theta^t]] = F^\circ_i(\gamma'_i|b'_i)$. Thus, in order to obtain high confidence bounds for the expected payments $F^\circ_i(\gamma_i|b_i)$ and $F^\circ_i(\gamma'_i|b'_i)$, we commit to each one of the two scoring rules for $N_3$ rounds and obtain the following estimates,

$$\hat{F}^\circ_i(\gamma_i|b_i) = \frac{1}{N_3} \sum_{t \in \mathcal{T}^{(i)}_{d,p}(\gamma_i)} \gamma_i[\tilde{s}^t_i, \theta^t] \qquad \hat{F}^\circ_i(\gamma'_i|b'_i) = \frac{1}{N_3} \sum_{t \in \mathcal{T}^{(i)}_{d,p}(\gamma'_i)} \gamma'_i[\tilde{s}^t_i, \theta^t],$$

where $\mathcal{T}^{(i)}_{d,p}(\gamma_i)$ denotes the set of round in which the principal committed to scoring rule $\gamma_i$ during the payment estimation phase. Moreover, using standard concentration arguments, we can provide the following high-confidence bounds for the estimated $\hat{F}^\circ_i$:

**Lemma B.2.** *For each $\gamma_i : S_i \times \Theta \to [0, M]$, for each $\delta \in (0, 1)$ and for each $K > 0$, the following holds:*

$$\mathbb{P}\left(|\hat{F}^\circ_i(\gamma_i|b_i) - F^\circ_i(\gamma_i|b_i)| \leq M\sqrt{\frac{\ln(2K/\delta)}{2N_3}}\right) \geq 1 - \frac{\delta}{K},$$

*where $b_i = b_i^\circ(\boldsymbol{\gamma}_i)$.*

*Proof.* The result follows directly from Hoeffding's bound, noticing that for all $t \in \mathcal{T}_{d,p}^{(i)}(\boldsymbol{\gamma}_i)$, $0 \leq \gamma_i[\tilde{s}_i^t, \theta^t] \leq M$, for all $t \in \mathcal{T}_{d,p}^{(i)}(\boldsymbol{\gamma}_i')$, $0 \leq \gamma_i'[\tilde{s}_i^t, \theta^t] \leq M$ and that

$$\mathbb{E}\left[\hat{F}_i^\circ(\boldsymbol{\gamma}_i|b_i)\right] = F_i^\circ(\boldsymbol{\gamma}_i|b_i), \qquad \mathbb{E}\left[\hat{F}_i^\circ(\boldsymbol{\gamma}_i'|b_i')\right] = F_i^\circ(\boldsymbol{\gamma}_i'|b_i').$$

$\square$

Starting from Equation (9), the last elements needed in order to obtain an high-confidence interval for the cost difference $C_i(b_i, b_i')$ are estimates for the expected payments $F_i^\circ(\boldsymbol{\gamma}_i|b_i')$ and $F_i^\circ(\boldsymbol{\gamma}_i'|b_i)$. We observe that, differently than $F_i^\circ(\boldsymbol{\gamma}_i|b_i)$ and $F_i^\circ(\boldsymbol{\gamma}_i'|b_i')$, it is not possible to estimate $F_i^\circ(\boldsymbol{\gamma}_i|b_i')$ and $F_i^\circ(\boldsymbol{\gamma}_i'|b_i)$ directly using the feedback collected from the online interaction, since every time the principal commits to scoring rule $\boldsymbol{\gamma}_i$, agent $i$ would respond by playing action $b_i$ rather than action $b_i'$ (the same holds for scoring rule $\boldsymbol{\gamma}'$ and action $b_i$). Nonetheless, it is possible to leverage the bound on the distance between $\boldsymbol{\gamma}_i$ and $\boldsymbol{\gamma}_i'$ shown in Lemma B.1 to address this issue. Formally,

**Lemma B.3.** *Let $\boldsymbol{\gamma}_i, \boldsymbol{\gamma}_i' : S_i \times \Theta \to [0, M]$ be such that $||\boldsymbol{\gamma}_i - \boldsymbol{\gamma}_i'||_\infty \leq \eta$. Then, for each $b_i \in B_i$, the following holds*

$$|F_i^\circ(\boldsymbol{\gamma}|b_i) - F_i^\circ(\boldsymbol{\gamma}_i'|b_i)| \leq \eta$$

*Proof.* Fix $b_i \in B_i$. Let $\bar{\varphi}, \bar{\varphi}'$ be such that

$$\bar{\varphi} \in arg \max_{\varphi: S_i \to S_i} \left\{ \sum_{s_i \in S_i} \sum_{\theta \in \Theta} \mathbb{P}^{(i)}(s_i, \theta|b_i) \gamma_i[\varphi(s_i), \theta] \right\}$$

$$\bar{\varphi}' \in arg \max_{\varphi: S_i \to S_i} \left\{ \sum_{s_i \in S_i} \sum_{\theta \in \Theta} \mathbb{P}^{(i)}(s_i, \theta|b_i) \gamma_i'[\varphi(s_i), \theta] \right\}.$$

Then, we have that

$$\begin{aligned}
F_i^\circ(\boldsymbol{\gamma}|b_i) &= \sum_{\theta \in \Theta} \mathbb{P}^{(i)}(s_i, \theta|b_i) \gamma_i[\bar{\varphi}(s_i), \theta] \\
&\geq \sum_{\theta \in \Theta} \mathbb{P}^{(i)}(s_i, \theta|b_i) \gamma_i[\bar{\varphi}'(s_i), \theta] \\
&\geq \sum_{\theta \in \Theta} \mathbb{P}^{(i)}(s_i, \theta|b_i) \gamma_i'[\bar{\varphi}'(s_i), \theta] - \eta \\
&= F_i^\circ(\boldsymbol{\gamma}'|b_i) - \eta,
\end{aligned}$$

and, similarly,

$$\begin{aligned}
F_i^\circ(\boldsymbol{\gamma}'|b_i) &= \sum_{\theta \in \Theta} \mathbb{P}^{(i)}(s_i, \theta|b_i) \gamma_i'[\bar{\varphi}'(s_i), \theta] \\
&\geq \sum_{\theta \in \Theta} \mathbb{P}^{(i)}(s_i, \theta|b_i) \gamma_i[\bar{\varphi}(s_i), \theta] \\
&\geq \sum_{\theta \in \Theta} \mathbb{P}^{(i)}(s_i, \theta|b_i) \gamma_i[\bar{\varphi}'(s_i), \theta] - \eta \\
&= F_i^\circ(\boldsymbol{\gamma}|b_i) - \eta.
\end{aligned}$$

Combining the two inequalities, we get

$$F_i(\boldsymbol{\gamma}'|b_i) - \eta \leq F_i(\boldsymbol{\gamma}|b_i) \leq F_i(\boldsymbol{\gamma}'|b_i) + \eta,$$

which gives the result. $\square$

In the following, for each agent $i \in \mathcal{N}$, we define as $\mathcal{D}_i \subseteq B_i \times B_i$ the set that contains all the couples $(b_i, b_i')$ for which the execution of the binary search reached the cost estimation phase. Furthermore, for each $i \in \mathcal{N}$ and $b_i, b_i' \in \mathcal{D}_i$, let $\boldsymbol{\gamma}_{i,b_i}$ and $\boldsymbol{\gamma}_{i,b_i'}$ be the scoring rules found at the end of the search phase, incentivizing actions $b_i$ and $b_i'$, respectively. To conclude the analysis of the payment estimation phase and to show how to leverage the results of Lemma B.1, Lemma B.2 and Lemma B.3 to recover upper and lower confidence bounds for the expected payment $C_i(b_i, b_i')$, we introduce the auxiliary clean event $\bar{\mathcal{E}}_d$.

**Definition B.1** (Auxiliary clean event for cost estimation). The clean event $\bar{\mathcal{E}}_d$ holds if

$$|\hat{F}_i^\circ(\boldsymbol{\gamma}_{i,b_i}|b_i) - F_i^\circ(\boldsymbol{\gamma}_{i,b_i}|b_i)| \leq M\sqrt{\frac{\ln(4nk^2/\delta)}{2N_3}}$$
$$|\hat{F}_i^\circ(\boldsymbol{\gamma}_{i,b_i'}|b_i') - F_i^\circ(\boldsymbol{\gamma}_{i,b_i'}|b_i')| \leq M\sqrt{\frac{\ln(4nk^2/\delta)}{2N_3}} \qquad \forall i \in \mathcal{N}, \, \forall (b_i, b_i') \in \mathcal{D}_i.$$

**Lemma B.4.** *The clean event $\bar{\mathcal{E}}_d$ holds with probability at least $1 - \frac{\delta}{2}$.*

*Proof.* By Lemma B.2, we have that

$$\mathbb{P}\left(|\hat{F}_i^\circ(\boldsymbol{\gamma}_{i,b_i}|b_i) - F_i^\circ(\boldsymbol{\gamma}_{i,b_i}|b_i)| \leq M\sqrt{\frac{\ln(4nk^2/\delta)}{2N_3}}\right) \geq 1 - \frac{\delta}{2nk^2}$$
$$\mathbb{P}\left(|\hat{F}_i^\circ(\boldsymbol{\gamma}_{i,b_i'}|b_i') - F_i^\circ(\boldsymbol{\gamma}_{i,b_i'}|b_i')| \leq M\sqrt{\frac{\ln(4nk^2/\delta)}{2N_3}}\right) \geq 1 - \frac{\delta}{2nk^2} \qquad \forall i \in \mathcal{N}, \, \forall (b_i, b_i') \in \mathcal{D}_i.$$

The result follows from a union bound, noticing that

$$\sum_{i \in \mathcal{N}} \sum_{(b_i, b_i') \in \mathcal{D}_i} \frac{\delta}{2nk^2} \leq \frac{\delta}{2}.$$

$\square$

The introduction of the auxiliary clean event allows us to define, for all $i \in \mathcal{N}$ and for all pairs of actions $(b_i, b_i') \in \mathcal{D}_i$ for which the binary search algorithms reached the cost estimation phase, the following estimators and confidence bounds:

$$\Lambda_i(b_i, b_i') = \hat{F}_i^\circ(\boldsymbol{\gamma}_{i,b_i}|b_i) - \hat{F}_i^\circ(\boldsymbol{\gamma}_{i,b_i'}|b_i'), \tag{14a}$$

$$\chi_i[b_i, b_i'] = 2M\sqrt{\frac{\ln(4nk^2/\delta)}{2N_3}} + \frac{M}{2^{N_2}}. \tag{14b}$$

We formally prove that, if event $\bar{\mathcal{E}}_d$ holds, then the above definitions yield, indeed, a valid high-confidence region for the pairwise cost differences.

**Lemma B.5.** *Assume clean event $\bar{\mathcal{E}}_d$ holds. Then the following holds:*

$$|\Lambda_i(b_i, b_i') - C_i(b_i, b_i')| \leq \chi_i[b_i, b_i'] \qquad \forall i \in \mathcal{N}, \, \forall (b_i, b_i') \in \mathcal{D}_i.$$

*Proof.* Fix $i \in \mathcal{N}$ and $(b_i, b_i') \in \mathcal{D}$. By Equation (9a), we have that

$$C_i(b_i, b_i') \leq F_i^\circ(\boldsymbol{\gamma}_{i,b_i}|b_i) - F_i^\circ(\boldsymbol{\gamma}_{i,b_i}|b_i') \tag{15a}$$

$$\leq F_i^\circ(\boldsymbol{\gamma}_{i,b_i}|b_i) - F_i^\circ(\boldsymbol{\gamma}_{i,b_i'}|b_i') + \frac{M}{2^{N_2}} \tag{15b}$$

$$\leq \hat{F}_i^\circ(\boldsymbol{\gamma}_{i,b_i}|b_i) - \hat{F}_i^\circ(\boldsymbol{\gamma}_{i,b_i'}|b_i') + \frac{M}{2^{N_2}} + 2M\sqrt{\frac{\ln(4nk^2/\delta)}{2N_3}}, \tag{15c}$$

where the first equation follows from Lemma B.3 and Lemma B.1, while the last equation follows from the definition of $\bar{\mathcal{E}}_d$. Similarly, from Equation (9b), we get

$$C_i(b_i, b'_i) \geq F_i^\circ(\gamma_{i,b'_i}|b_i) - F_i^\circ(\gamma_{i,b'_i}|b'_i) \tag{16a}$$

$$\geq F_i^\circ(\gamma_{i,b_i}|b_i) - F_i^\circ(\gamma_{i,b'_i}|b'_i) - \frac{M}{2^{N_2}} \tag{16b}$$

$$\geq \hat{F}_i^\circ(\gamma_{i,b_i}|b_i) - \hat{F}_i^\circ(\gamma_{i,b'_i}|b'_i) - \frac{M}{2^{N_2}} - 2M\sqrt{\frac{\ln(4nk^2/\delta)}{2N_3}}, \tag{16c}$$

where, also in this case, the first equation follows from Lemma B.3 and Lemma B.1, while the last equation follows from the definition of $\bar{\mathcal{E}}_d$. The result follows by substituting the definitions of $\Lambda_i(b_i, b'_i)$ and $\chi_i[b_i, b'_i]$. $\qquad\square$

**Split phase.** The analysis of the payment estimation phase suggested that whenever we are able to successfully complete the search phase, then we can easily recover high-confidence regions for the cost differences. However, unfortunately, this is not always the case, since it might happen that during the search phase for an agent $i$ and between two actions $b_i, b'_i \in B_i$, the agent responds with an action $\tilde{b}_i^t$ which is different than $b_i$ and $b'_i$, thus causing the search phase to end. In such a case, the algorithm enters the split phase. The intuition upon which the design of the split phase is based is that the cost difference $C_i(b_i, b'_i)$ can be expressed in terms of the cost differences $C_i(b_i, \tilde{b}_i^t)$ and $C_i(\tilde{b}_i^t, b'_i)$ as follows:

$$C_i(b_i, b'_i) = C_i(b_i, \tilde{b}_i^t) + C_i(\tilde{b}_i^t, b'_i).$$

Then, this shows that it is possible to recursively obtain an high-confidence interval for $C_i(b_i, b'_i)$ starting from high-confidence intervals for $C_i(b_i, \tilde{b}_i^t)$ and $C_i(\tilde{b}_i^t, b'_i)$. In particular,

$$\Lambda_i(b_i, \tilde{b}_i^t) + \Lambda_i(\tilde{b}_i^t, b'_i) - \chi_i[b_i, \tilde{b}_i^t] - \chi_i[\tilde{b}_i^t, b'_i] \leq C_i(b_i, b'_i) \leq \Lambda_i(b_i, \tilde{b}_i^t) + \Lambda_i(\tilde{b}_i^t, b'_i) + \chi_i[b_i, \tilde{b}_i^t] + \chi_i[\tilde{b}_i^t, b'_i],$$

which allows us to define the estimator and confidence bound for $C_i(b_i, b'_i)$ as

$$\Lambda_i(b_i, b'_i) = \Lambda_i(b_i, \tilde{b}_i^t) + \Lambda_i(\tilde{b}_i^t, b'_i)$$
$$\chi_i[b_i, b'_i] = \chi_i[b_i, \tilde{b}_i^t] + \chi_i[\tilde{b}_i^t, b'_i].$$

Hence, when the algorithm enters into the split phase, it instantiates two distinct bynary search routines, one for estimating the cost difference $C_i(b_i, \tilde{b}_i^t)$ using as starting points the scoring rules $\gamma_i$ and $\bar{\gamma}_i$ and the other for estimating $C_i(\tilde{b}_i^t, b'_i)$ starting from scoring rules $\bar{\gamma}_i$ and $\gamma'_i$.

We proceed by bounding the number of times the algorithm enters the split phase, thus allowing us to bound the number of rounds used by ESTIMATECOSTS, as well as the high-confidence bound $\chi = \max_{i,b_i,b'_i} \chi_i[b_i, b'_i]$. To address this question, we consider the binary search procedures instantiated by the main loop of the algorithm. For the sake of presentation, let us fix an agent $i$ and two actions $b_i, b'_i \in B_i$ and consider a binary search between the uncorrelated scoring rules $\gamma_i^b, \gamma_i^{b'_i} \in \Gamma_i$. Then, the maximum number of times the algorithm enters the split phase during the process of estimating the cost difference $C_i(b_i, b'_i)$ can be characterized by investigating all the possible convex combinations of the two scoring rules $\gamma_i^{b_i}, \gamma_i^{b'_i}$. In particular, we introduce *best-response (BR) regions, i.e.,* sets $\mathcal{G}_i(\tilde{b}_i, \varphi|b_i, b'_i) \subseteq [0,1]$ such that for each $\alpha \in \mathcal{G}_i(\tilde{b}_i, \varphi|b_i, b'_i)$, the best response of agent $i$ to scoring rule $\gamma_i^{\alpha,b_i,b'_i} := \alpha\gamma_i^{b_i} + (1-\alpha)\gamma_i^{b'_i}$ does not change. Formally,

$$\mathcal{G}_i(\tilde{b}_i, \varphi|b_i, b'_i) := \left\{ \alpha \in [0,1] \mid b_i^\circ(\gamma_i^{\alpha,b_i,b'_i}) = \tilde{b}_i, \; \varphi_i^\circ(s_i|\gamma_i^{\alpha,b_i,b'_i}) = \varphi(s_i) \quad \forall s_i \in S_i \right\}.$$

Trivially, $[0,1] = \cup_{\tilde{b}_i,\varphi} \mathcal{G}_i(\tilde{b}_i, \varphi|b_i, b'_i)$ and $\mathcal{G}_i(\tilde{b}_i, \varphi|b_i, b'_i) \cap \mathcal{G}_i(\tilde{b}'_i, \varphi'|b_i, b'_i) = \varnothing$ for all $\tilde{b}_i, \tilde{b}'_i \in B_i$ and for all $\varphi, \varphi' : S_i \to S_i$, thus, the set

$$\mathcal{G} = \{\mathcal{G}_i(\tilde{b}_i, \varphi|b_i, b'_i) \mid \tilde{b}_i \in B_i, \; \varphi : S_i \to S_i, \; \mathcal{G}_i(\tilde{b}_i, \varphi|b_i, b'_i) \neq \varnothing\}$$

provides a partition of $[0,1]$. In principle, $[0,1]$ can be partitioned in many BR regions (since the number of different signal reporting policies $\varphi : S_i \to S_i$ is $l^l$). As we show next, there exist in fact a limited number of non-empty BR regions. Before proving such result, we introduce an accessory Lemma.

**Lemma B.6.** *Let $\mathcal{H} \subset \mathbb{R}$ be a closed, convex subset of $\mathbb{R}$ and $\{\mathcal{G}^1, ..., \mathcal{G}^n\}$ be a set of partitions of $\mathcal{H}$, where $\mathcal{G}^i = \{G_1^i, ..., G_n^i\}$ with $G_j^i$ convex for each $i, j \in [n]$. Let $\mathcal{P} = \{P_1, ..., P_m\}$ be a partition of $\mathcal{H}$ such that for $j \in [n]$, $P_j = \cap_{i \in [n]} G_{f(i,j)}^i$ for some $f(i, j) \in [n]$. Then, it holds that $m \leq n^2 - n + 1$.*

*Proof.* Let $\mathcal{P}^{(1)} = \mathcal{G}^1$ and $m(1) = n$. Let now $\mathcal{P}^{(k)} = \{P_1^{(k)}, ..., P_{m(k)}^{(k)}\}$, for $m(k) \in \mathbb{N}_{>0}$ be a partition of $\mathcal{H}$ such that for each $j \in [m(k)]$, $P_j^{(k)}$ is convex and $P_j^{(k)} = \cap_{i \in [k]} G_{f^{(k)}(i,j)}^i$, with $f^{(k)}(i, j) \in [n]$. In other words, $\mathcal{P}^{(k)}$ is a set of non-empty and disjoint intervals $P_j^{(k)} \subseteq \mathcal{H}$ such that $\cup_{j \in m(k)} P_j^{(k)} = \mathcal{H}$ in which each element $P_j^{(k)}$ is obtained as the intersection of one element from each $\{\mathcal{G}^1, ...\mathcal{G}^k\}$. Similarly, define $\mathcal{P}^{(k-1)} = \{P_1^{(k-1)}, ..., P_{m(k-1)}^{(k-1)}\}$, for $m(k-1) \in \mathbb{N}_{>0}$ be a partition of $\mathcal{H}$ such that for each $j \in [m(k-1)]$, $P_j^{(k-1)} = \cap_{i \in [k-1]} G_{f^{(k-1)}(i,j)}^i$ for $f^{(k-1)}(i, j) \in [n]$. Notice that, by definition of $\mathcal{P}^{(k-1)}$, in holds that $\forall j \in [m(k)]$, $P_j^{(k)} = G_a^k \cap P_b^{(k-1)}$ for some $a \in [n]$, $b \in [m(k-1)]$. Furthermore, it is easy to see that by convexity of every $P_j^{(k-1)}$ and $G_j^k$, $m(k)$ can be at most $n + m(k-1) - 1$. Thus, we can conclude the following chain of inequalities:

$$m \leq m(n) = n + m(n-1) - 1 \leq 2n + m(n-2) - 2 \leq ... \leq (n-1)n + m(1) - (n-1) = n^2 - n + 1.$$

This concludes the proof. $\square$

We proceed to bound the maximum number of non-empty BR regions.

**Lemma B.7.** *For any agent $i$ and actions $b_i, b_i' \in B_i$, there exist at most $kl^2$ non-empty BR regions, i.e., $|\mathcal{G}| \leq kl^2$. Furthermore, for each $\tilde{b}_i \in B_i$ and $\varphi : S_i \to S_i$, the set $\mathcal{G}_i(\tilde{b}_i, \varphi | b_i, b_i')$ is convex.*

*Proof.* Fix an agent $i$ and two actions $b_i, b_i' \in B_i$. We first prove the convexity of the BR regions and then we proceed to bound their number.

**Convexity.** To prove that the sets $\mathcal{G}_i(\tilde{b}_i, \varphi | b_i, b_i')$ are convex for each $\tilde{b}_i \in B_i$ and $\varphi : S_i \to S_i$, let us explicit the constraints defining such sets. In particular, imposing that $\tilde{b}_i$ and $\varphi$ are a BR for agent $i$ is equivalent to imposing the following constraints for each $\tilde{b}_i \in B_i$ and for all $\varphi : S_i \to S_i$:

$$\sum_{s_i \in S_i} \sum_{\theta \in \Theta} \left[ \mathbb{P}^{(i)}\left(s_i, \theta | \tilde{b}_i\right) \gamma_i^{\alpha b_i, b_i'}[\varphi(s_i), \theta] - \mathbb{P}^{(i)}\left(s_i, \theta | \tilde{b}_i'\right) \gamma_i^{\alpha b_i, b_i'}[\varphi'(s_i), \theta] \right] \geq 0.$$

Clearly, since $\gamma_i^{\alpha, b_i, b_i'}$ is a linear expression in $\alpha$, all the constraints are linear in $\alpha$, thus implying the convexity of the set.

**Bound on the number of non-empty BR regions.** To bound the number of non-empty BR regions, we introduce, for two signals $s_i, s_i' \in S_i$ and an action $\tilde{b}_i \in B_i$, the set $G_i(\tilde{b}_i, s_i, s_i' | b_i, b_i') \subseteq [0, 1]$, containing all those $\alpha \in [0, 1]$ for which agent $i$'s BR to scoring rule $\gamma_i^{\alpha, b_i, b_i'}$ after having played action $\tilde{b}_i \in B_i$ and having observed signal $s_i \in S_i$, is to report signal $s_i'$. Formally,

$$G_i(\tilde{b}_i, s_i, s_i' | b_i, b_i') := \left\{ \alpha \in [0, 1] \mid \sum_{\theta \in \Theta} \mathbb{P}^{(i)}\left(s_i, \theta | \tilde{b}_i\right) \left[ \gamma_i^{\alpha, b_i, b_i'}[s_i', \theta] - \gamma_i^{\alpha, b_i, b_i'}[\tilde{s}_i, \theta] \right] \geq 0 \ \forall \tilde{s}_i \in S_i \right\}.$$

Furthermore, for each $\tilde{b}_i \in B_i$ and $\varphi : S_i \to S_i$, we let, with a slight abuse of notation, $G_i(\tilde{b}_i, \varphi | b_i, b_i')$ be the subset of $[0, 1]$ such that, after having selected action $\tilde{b}_i$, the best-response policy for agent $i$ to scoring rule $\gamma_i^{\alpha, b_i, b_i'}$ is to report signals according to $\varphi$. Formally,

$$G_i(\tilde{b}_i, \varphi | b_i, b_i') := \left\{ \alpha \in [0, 1] \mid \sum_{\theta \in \Theta} \mathbb{P}^{(i)}\left(s_i, \theta | \tilde{b}_i\right) \left[ \gamma_i^{\alpha, b_i, b_i'}[\varphi(s_i), \theta] - \gamma_i^{\alpha, b_i, b_i'}[\tilde{s}_i, \theta] \right] \geq 0 \ \forall s_i, \tilde{s}_i \in S_i \right\}.$$

By definition, each $G_i(\tilde{b}_i, \varphi|b_i, b_i')$ can be obtained as the intersection of suitably chosen BR regions $G_i(\tilde{b}_i, s_i, s_i'|b_i, b_i')$, i.e.,

$$G_i(\tilde{b}_i, \varphi|b_i, b_i') = \bigcap_{s_i \in S_i} G_i(\tilde{b}_i, s_i, \varphi(s_i)|b_i, b_i') \qquad \forall \tilde{b}_i \in B_i, \forall \varphi : S_i \to S_i.$$

Moreover, we highlight that, by definition, for each $\tilde{b}_i \in B_i$ and $s_i \in S_i$, the set $\{G_i(\tilde{b}_i, s_i, s_i'|b_i, b_i') \mid s_i' \in S_i\}$ provides a partition of $[0, 1]$. Then, by Lemma B.6 we can conclude that $|G_i(\tilde{b}_i, \varphi|b_i, b_i')| \leq l^2 - l + 1$. Notice that, in general it holds that for each $\tilde{b}_i \in B_i$ and $\varphi : S_i \to S_i$, $G_i(\tilde{b}_i, \varphi|b_i, b_i') \supseteq \mathcal{G}_i(\tilde{b}_i, \varphi|b_i, b_i')$, since by definition of $G_i(\tilde{b}_i, \varphi|b_i, b_i')$, for each $\alpha$ belonging to the set, the signal reporting policy $\varphi$ is a best-response for agent $i$ to $\gamma_i^{\alpha, b_i, b_i'}$ after the agent played action $\tilde{b}_i$, but nothing guarantees that action $\tilde{b}_i$ is a best-response itself. To conclude the proof it is enough to notice that, in the worst case, for each $\tilde{b}_i \in B_i$ and $\varphi : S_i \to S_i$, $G_i(\tilde{b}_i, \varphi|b_i, b_i') = \mathcal{G}_i(\tilde{b}_i, \varphi|b_i, b_i')$ and since the number of non-empty $G_i(\tilde{b}_i, \varphi|b_i, b_i')$ is at most $l^2 - l + 1$, we can conclude that $|\mathcal{G}| \leq k(l^2 - l + 1) \leq kl^2$. This concludes the proof. $\square$

The result of Lemma B.7 allows us to bound the maximum possible confidence bound as follows.

**Lemma B.8.** *At the end of the execution of* ESTIMATECOSTS *it holds that*

$$\max_{i \in \mathcal{N}} \max_{b_i, b_i' \in B_i} \chi_i[b_i, b_i'] \leq 2kl^2 M \sqrt{\frac{\ln(4nk^2/\delta)}{2N_3}} + \frac{kl^2 M}{2^{N_2}}.$$

*Proof.* By Lemma B.7, since for each $i \in \mathcal{N}$ and for each $b_i, b_i' \in B_i$, $[0, 1]$ can be partitioned in at most $kl^2$ convex BR regions, the condition for entering the split phase will be triggered at most $\lfloor \frac{kl^2}{2} \rfloor$ times during the routine for the estimation of $C_i(b_i, b_i')$, triggering the beginning of a new search phase at most $kl^2$ times. The result follows since the confidence bound $\chi_i[b_i, b_i']$ is obtained as the sum of all confidence bounds obtained at the end of the corresponding payment estimation phases, hence:

$$\chi_i[b_i, b_i'] \leq 2kl^2 M \sqrt{\frac{\ln(4nk^2/\delta)}{2N_3}} + \frac{kl^2 M}{2^{N_2}}.$$

This concludes the proof. $\square$

### B.2 FORMAL GUARANTEES OF ESTIMATECOSTS

We conclude this Section by formally proving main properties of Algorithm 4.

**Lemma 7.1.** *The clean event $\mathcal{E}_c$ holds with probability at least $1 - \frac{\delta}{2}$.*

*Proof.* Assume auxiliary clean event $\bar{\mathcal{E}}_d$ holds. By Lemma B.5, for all $i \in \mathcal{N}$ and for all $(b_i, b_i') \in \mathcal{D}_i$,

$$|\Lambda_i(b_i, b_i') - C_i(b_i, b_i')| \leq \chi_i[b_i, b_i'] \leq \chi.$$

Then, let us fix $i \in \mathcal{N}$ and consider a couple $(b_i, b_i') \notin \mathcal{D}_i$. Let $\mathcal{D}_i(b_i, b_i')$ be the set that contains all the pairs of actions $(\hat{b}_i, \hat{b}_i')$ for which the payment estimation phase has been reached during the procedure for estimating $C_i(b_i, b_i')$. Then, the algorithm guarantees that

$$C_i(b_i, b_i') = \sum_{(\hat{b}_i, \hat{b}_i') \in \mathcal{D}_i(b_i, b_i')} C_i(\hat{b}_i, \hat{b}_i').$$

The result follows from Lemma B.5 noticing that $\mathcal{D}_i(b_i, b_i') \subseteq \mathcal{D}_i$, $\Lambda_i(b_i, b_i) = \sum_{(\hat{b}_i, \hat{b}_i') \in \mathcal{D}_i(b_i, b_i')} \Lambda_i(\hat{b}_i, \hat{b}_i')$, and $\chi_i[b_i, b_i'] = \sum_{(\hat{b}_i, \hat{b}_i') \in \mathcal{D}_i(b_i, b_i')} \chi_i[\hat{b}_i, \hat{b}_i']$. $\square$

**Lemma 7.2.** *Let $\mathcal{T}_d$ be the set of rounds devoted to the execution of* ESTIMATECOSTS. *Then it holds that $|\mathcal{T}_d| \leq nk^3 l^2(N_2 + N_3)$.*

*Proof.* By Lemma B.7, since for each $i \in \mathcal{N}$ and for each $b_i, b_i' \in B_i$, $\mathcal{H}_i(b_i, b_i')$ can be partitioned in at most $kl^2$ BR regions, the agent will change her action at most $\lfloor \frac{kl^2}{2} \rfloor$ times during the routine for the estimation of $C_i(b_i, b_i')$, triggering the beginning of a new search phase at most $kl^2$ times. This implies that

$$|\mathcal{T}_d| \leq \sum_{i \in \mathcal{N}} \sum_{b_i, b_i' \in B_i} kl^2(N_2 + N_3) \leq nk^3l^2(N_2 + N_3).$$

This concludes the proof. □

## C  PROOFS OMITTED FROM SECTION 3

**Theorem 3.1.** *Let* $(\boldsymbol{x}^\star, \boldsymbol{y}^\star, \boldsymbol{\mu}^\star, \boldsymbol{z}^\star)$ *be an optimal solution to LP* $(\mathbb{P}, C, 0)$*, where* $C = (C_1, ..., C_n)$*. Then, let* $\boldsymbol{\gamma}^\star = (\gamma_1^\star, ..., \gamma_n^\star)$ *and* $\boldsymbol{\pi}^\star$ *be such that*

$$\gamma_i^\star[\boldsymbol{b}, \boldsymbol{s}, \theta] = \begin{cases} \frac{x_i^\star[\boldsymbol{b}, \boldsymbol{s}, \theta]}{\mu^\star[\boldsymbol{b}]} & \text{if } \mu[\boldsymbol{b}] \neq 0 \\ 0 & \text{otherwise} \end{cases} \qquad \pi^\star[\boldsymbol{b}, \boldsymbol{s}, a] = \begin{cases} \frac{y^\star[\boldsymbol{b}, \boldsymbol{s}, a]}{\mu^\star[\boldsymbol{b}]} & \text{if } \mu[\boldsymbol{b}] \neq 0 \\ \frac{1}{d} & \text{otherwise.} \end{cases}$$

*Then* $(\boldsymbol{\mu}^\star, \boldsymbol{\gamma}^\star, \boldsymbol{\pi}^\star)$ *is an optimal solution to Problem (1).*

*Proof.* We start the proof of this theorem by showing that $(\boldsymbol{\mu}^\star, \boldsymbol{\gamma}^\star, \boldsymbol{\pi}^\star)$ is feasible for optimization problem (1). Then, we prove that $(\boldsymbol{\mu}^\star, \boldsymbol{\gamma}^\star, \boldsymbol{\pi}^\star)$ is also optimal.

**Feasibility.**   First, notice that

$$0 \leq \gamma_i^\star[\boldsymbol{b}, \boldsymbol{s}, \theta] \geq M \quad \forall i \in \mathcal{N}, \forall \boldsymbol{b} \in \mathcal{B}, \forall \boldsymbol{s} \in \mathcal{S}, \forall \theta \in \Theta,$$

thus guaranteeing that $\boldsymbol{\gamma}^\star$ is a valid scoring rule. Furthermore,

$$\sum_{a \in \mathcal{A}} \pi^\star[\boldsymbol{b}, \boldsymbol{s}, a] = \begin{cases} \sum_{a \in \mathcal{A}} \frac{y^\star[\boldsymbol{b}, \boldsymbol{s}, a]}{\mu^\star[\boldsymbol{b}]} & \text{if } \mu^\star[\boldsymbol{b}] \neq 0 \\ \sum_{a \in \mathcal{A}} \frac{1}{d} \text{otherwise} \end{cases} = 1 \quad \forall \boldsymbol{b} \in \mathcal{B}, \forall \boldsymbol{s} \in \mathcal{S},$$

which implies that $\boldsymbol{\pi}^\star$ is a valid action policy.

For what concerns incentive compatibility, for each $i \in \mathcal{N}$ let us notice that:

$$F_i(\boldsymbol{\mu}^\star, \boldsymbol{\gamma}^\star) = \sum_{\substack{\boldsymbol{b} \in \mathcal{B} \\ \boldsymbol{s} \in \mathcal{S} \\ \theta \in \Theta}} \mu^\star[\boldsymbol{b}] \gamma_i^\star[\boldsymbol{b}, \boldsymbol{s}, \theta] \mathbb{P}(\boldsymbol{s}, \theta | \boldsymbol{b}) \tag{17a}$$

$$= \sum_{\substack{\boldsymbol{b} \in \mathcal{B} \\ \boldsymbol{s} \in \mathcal{S} \\ \theta \in \Theta}} x_i^\star[\boldsymbol{b}, \boldsymbol{s}, \theta] \mathbb{P}(\boldsymbol{s}, \theta | \boldsymbol{b}) \tag{17b}$$

$$= \sum_{b \in B_i} \sum_{s \in S_i} \sum_{\substack{\boldsymbol{b}_{-i} \in \mathcal{B}_{-i} \\ \boldsymbol{s}_{-i} \in \mathcal{S}_{-i} \\ \theta \in \Theta}} x_i[(b, \boldsymbol{b}_{-i}), (s', \boldsymbol{s}_{-i}), \theta] \mathbb{P}((s, \boldsymbol{s}_{-i}), \theta | (b', \boldsymbol{b}_{-i})) \tag{17c}$$

$$= \sum_{b \in B_i} \sum_{s \in S_i} f_i(\boldsymbol{x}_i | b, s, \mathbb{P}). \tag{17d}$$

Similarly, for each $i \in \mathcal{N}$ and for each $(\phi, \varphi) \in \Phi_i$:

$$F_i^{\phi,\varphi}(\boldsymbol{\mu}^\star, \boldsymbol{\gamma}^\star) = \sum_{\substack{\boldsymbol{b} \in \mathcal{B} \\ \boldsymbol{s} \in \mathcal{S} \\ \theta \in \Theta}} \mu^\star[\boldsymbol{b}]\gamma_i^\star[\boldsymbol{b}, (\varphi(b_i, s_i), \boldsymbol{s}_{-i}), \theta]\mathbb{P}\left(\boldsymbol{s}, \theta | (b_i, \boldsymbol{b}_{-i})\right) \tag{18a}$$

$$= \sum_{\substack{\boldsymbol{b} \in \mathcal{B} \\ \boldsymbol{s} \in \mathcal{S} \\ \theta \in \Theta}} x_i^\star[\boldsymbol{b}, (\varphi(b_i, s_i), \boldsymbol{s}_{-i}), \theta]\mathbb{P}\left(\boldsymbol{s}, \theta | (\phi(b_i), \boldsymbol{b}_{-i})\right) \tag{18b}$$

$$= \sum_{b \in B_i} \sum_{s \in S_i} \sum_{\substack{\boldsymbol{b}_{-i} \in \mathcal{B}_{-i} \\ \boldsymbol{s}_{-i} \in \mathcal{S}_{-i} \\ \theta \in \Theta}} x_i^\star[\boldsymbol{b}, (\varphi(b, s), \boldsymbol{s}_{-i}), \theta]\mathbb{P}\left(\boldsymbol{s}, \theta | (\phi(b), \boldsymbol{b}_{-i})\right) \tag{18c}$$

$$= \sum_{b \in B_i} \sum_{s \in S_i} f_i(\boldsymbol{x}_i | b, \phi(b), s, \varphi(b, s), \mathbb{P}) \tag{18d}$$

$$\leq \sum_{b \in B_i} \sum_{s \in S_i} z_i^\star[b, \phi(b), s], \tag{18e}$$

where the last inequality follows from feasibility of $(\boldsymbol{x}^\star, \boldsymbol{y}^\star, \boldsymbol{\mu}^\star, \boldsymbol{z}^\star)$. Combining Equations (17d) and (18e), we get:

$$F_i(\boldsymbol{\mu}^\star, \boldsymbol{\gamma}^\star) - F_i^{\phi,\varphi}(\boldsymbol{\mu}^\star, \boldsymbol{\gamma}^\star) \geq \sum_{b \in B_i} \sum_{s \in S_i} f_i(\boldsymbol{x}_i | b, s, \mathbb{P}) - z_i^\star[b, \phi(b), s] \tag{19a}$$

$$\geq \sum_{b \in B_i} \sum_{\boldsymbol{b}_{-i} \in \mathcal{B}_i} \mu^\star[(b, \boldsymbol{b}_{-i})]\left(c_i(b) - c_i(\phi(b))\right) \tag{19b}$$

$$= \sum_{\boldsymbol{b} \in \mathcal{B}} \mu[\boldsymbol{b}]\left(c_i(b_i) - c_i(\phi(b_i))\right), \tag{19c}$$

where the second inequality follows from feasibility of $(\boldsymbol{x}^\star, \boldsymbol{y}^\star, \boldsymbol{\mu}^\star, \boldsymbol{z}^\star)$. This proves the incentive compatibility of $(\boldsymbol{\mu}^\star, \boldsymbol{\gamma}^\star, \boldsymbol{\pi}^\star)$.

**Optimality.** To prove that $(\boldsymbol{\mu}^\star, \boldsymbol{\gamma}^\star, \boldsymbol{\pi}^\star)$ is optimal, assume, by contradiction that there exists an IC mechanism $(\boldsymbol{\mu}, \boldsymbol{\gamma}, \boldsymbol{\pi})$ such that $U(\boldsymbol{\mu}, \boldsymbol{\gamma}, \boldsymbol{\pi}) > U(\boldsymbol{\mu}^\star, \boldsymbol{\gamma}^\star, \boldsymbol{\pi}^\star)$. Let $(\boldsymbol{x}, \boldsymbol{y}, \boldsymbol{z})$ be such that:

$$x_i[\boldsymbol{b}, \boldsymbol{s}, \theta] = \mu[\boldsymbol{b}]\gamma_i[\boldsymbol{b}, \boldsymbol{s}, \theta] \quad \forall i \in \mathcal{N}, \forall \boldsymbol{b} \in \mathcal{B}, \forall \boldsymbol{s} \in \mathcal{S}, \forall \theta \in \Theta$$

$$y[\boldsymbol{b}, \boldsymbol{s}, a] = \mu[\boldsymbol{b}]\pi[\boldsymbol{s}, a] \quad \forall \boldsymbol{b} \in \mathcal{B}, \forall \boldsymbol{s} \in \mathcal{S}, \forall a \in \mathcal{A}$$

$$z_i[b, b', s] = \max_{s' \in S_i}\left\{f_i(\boldsymbol{x}_i | b, b', s, s', \mathbb{P})\right\} \quad \forall i \in \mathcal{N}, \forall b, b' \in B_i, \forall s \in S_i.$$

Let us notice that $(\boldsymbol{x}, \boldsymbol{y}, \boldsymbol{\mu}, \boldsymbol{z})$ satisfy, by definition, constraints (2c), (2d) and (2e) of $\text{LP}(\mathbb{P}, C, 0)$. To prove that they satisfy also constraints (2b), for each $i \in \mathcal{N}$ and $b, b' \in B_i$, let $(\phi, \varphi) \in \Phi_i$ be such that:

$$\phi(b'') = \begin{cases} b' & \text{if } b'' = b \\ b'' & \text{otherwise} \end{cases} \quad \text{and} \quad \varphi(b'', s) = s \quad \forall b'' \in B_i \setminus \{b\}, \forall s \in S_i.$$

Then, by incentive compatibility of $(\boldsymbol{\mu}, \boldsymbol{\gamma}, \boldsymbol{\pi})$:

$$\sum_{\boldsymbol{b} \in \mathcal{B}} \mu[\boldsymbol{b}] \left( c_i(b) - c_i(\phi(b)) \right) = \sum_{\boldsymbol{b}_{-i} \in \mathcal{B}_{-i}} \mu[(b, \boldsymbol{b}_{-i})] \left( c_i(b) - c_i(b') \right) \tag{21a}$$

$$\leq F_i(\boldsymbol{\mu}, \boldsymbol{\gamma}) - F_i^{\phi, \varphi}(\boldsymbol{\mu}, \boldsymbol{\gamma}) \tag{21b}$$

$$= \sum_{\substack{\boldsymbol{b} \in \mathcal{B} \\ \boldsymbol{s} \in \mathcal{S} \\ \theta \in \Theta}} \mu[\boldsymbol{b}] \left( \gamma_i[\boldsymbol{b}, \boldsymbol{s}, \theta] \mathbb{P}\left( \boldsymbol{s}, \theta | \boldsymbol{b} \right) - \gamma_i[\boldsymbol{b}, \varphi(b_i, s_i), \theta] \mathbb{P}\left( \boldsymbol{s}, \theta | (b_i, \phi(b_i)) \right) \right) \tag{21c}$$

$$= \sum_{\substack{\boldsymbol{b} \in \mathcal{B} \\ \boldsymbol{s} \in \mathcal{S} \\ \theta \in \Theta}} x_i[\boldsymbol{b}, \boldsymbol{s}, \theta] \mathbb{P}\left( \boldsymbol{s}, \theta | \boldsymbol{b} \right) - x_i[\boldsymbol{b}, \varphi(b_i, s_i), \theta] \mathbb{P}\left( \boldsymbol{s}, \theta | (b_i, \phi(b_i)) \right) \tag{21d}$$

$$= \sum_{s \in S_i} \left[ f_i(\boldsymbol{x} | b, s, \mathbb{P}) - f_i(\boldsymbol{x}_i | b, s, b', \varphi(b, s)) \right] \tag{21e}$$

$$\leq \sum_{s \in S_i} \left[ f_i(\boldsymbol{x} | b, s, \mathbb{P}) - z_i[b, b', s] \right]. \tag{21f}$$

Thus we can conclude that $(\boldsymbol{x}, \boldsymbol{y}, \boldsymbol{\mu}, \boldsymbol{z})$ is feasible for $LP(\mathbb{P}, C, 0)$. Moreover:

$$\sum_{\substack{\boldsymbol{b} \in \mathcal{B} \\ \boldsymbol{s} \in \mathcal{S} \\ a \in \mathcal{A}}} \left[ y[\boldsymbol{b}, \boldsymbol{s}, a] \left( \sum_{\theta \in \Theta} \mathbb{P}\left( \boldsymbol{s}, \theta | \boldsymbol{b} \right) u(a, \theta) \right) \right] - \sum_{i \in \mathcal{N}} \sum_{\substack{\boldsymbol{b} \in \mathcal{B} \\ \boldsymbol{s} \in \mathcal{S} \\ \theta \in \Theta}} x_i[\boldsymbol{b}, \boldsymbol{s}, \theta] \mathbb{P}\left( \boldsymbol{s}, \theta | \boldsymbol{b} \right) \tag{22a}$$

$$= \sum_{\substack{\boldsymbol{b} \in \mathcal{B} \\ \boldsymbol{s} \in \mathcal{S} \\ a \in \mathcal{A}}} \left[ \mu[\boldsymbol{b}] \pi[\boldsymbol{b}, \boldsymbol{s}, a] \left( \sum_{\theta \in \Theta} \mathbb{P}\left( \boldsymbol{s}, \theta | \boldsymbol{b} \right) u(a, \theta) \right) \right] - \sum_{i \in \mathcal{N}} \sum_{\substack{\boldsymbol{b} \in \mathcal{B} \\ \boldsymbol{s} \in \mathcal{S} \\ \theta \in \Theta}} \mu[\boldsymbol{b}] \gamma_i[\boldsymbol{b}, \boldsymbol{s}, \theta] \mathbb{P}\left( \boldsymbol{s}, \theta | \boldsymbol{b} \right) \tag{22b}$$

$$= U(\boldsymbol{\mu}, \boldsymbol{\gamma}, \boldsymbol{b}) \tag{22c}$$

$$> U(\boldsymbol{\mu}^\star, \boldsymbol{\gamma}^\star, \boldsymbol{b}^\star) \tag{22d}$$

$$= \sum_{\substack{\boldsymbol{b} \in \mathcal{B} \\ \boldsymbol{s} \in \mathcal{S} \\ a \in \mathcal{A}}} \left[ \mu^\star[\boldsymbol{b}] \pi^\star[\boldsymbol{b}, \boldsymbol{s}, a] \left( \sum_{\theta \in \Theta} \mathbb{P}\left( \boldsymbol{s}, \theta | \boldsymbol{b} \right) u(a, \theta) \right) \right] - \sum_{i \in \mathcal{N}} \sum_{\substack{\boldsymbol{b} \in \mathcal{B} \\ \boldsymbol{s} \in \mathcal{S} \\ \theta \in \Theta}} \mu^\star[\boldsymbol{b}] \gamma_i^\star[\boldsymbol{b}, \boldsymbol{s}, \theta] \mathbb{P}\left( \boldsymbol{s}, \theta | \boldsymbol{b} \right) \tag{22e}$$

$$= \sum_{\substack{\boldsymbol{b} \in \mathcal{B} \\ \boldsymbol{s} \in \mathcal{S} \\ a \in \mathcal{A}}} \left[ y^\star[\boldsymbol{b}, \boldsymbol{s}, a] \left( \sum_{\theta \in \Theta} \mathbb{P}\left( \boldsymbol{s}, \theta | \boldsymbol{b} \right) u(a, \theta) \right) \right] - \sum_{i \in \mathcal{N}} \sum_{\substack{\boldsymbol{b} \in \mathcal{B} \\ \boldsymbol{s} \in \mathcal{S} \\ \theta \in \Theta}} x_i^\star[\boldsymbol{b}, \boldsymbol{s}, \theta] \mathbb{P}\left( \boldsymbol{s}, \theta | \boldsymbol{b} \right), \tag{22f}$$

which contradicts the assumption on the optimality of $(\boldsymbol{x}^\star, \boldsymbol{y}^\star, \boldsymbol{\mu}^\star, \boldsymbol{z}^\star)$. This concludes the proof. $\square$

## D  PROOFS OMITTED FROM SECTION 4

Before proving the formal results of Section 4, we introduce a useful preliminary result.

**Lemma D.1.** *For any uncorrelated mechanism* $(\boldsymbol{\gamma}, \boldsymbol{\pi}) \in \mathcal{U}$, *there exists an equivalent uncorrelated mechanism* $(\boldsymbol{\gamma}', \boldsymbol{\pi}') \in \mathcal{U}$ *which is truthful, i.e., such that for any* $i \in \mathcal{N}$ *and* $s_i \in S_i$, $\varphi_i^\circ(s_i | \boldsymbol{\gamma}_i') = s_i$ *and such that*

$$U^\circ(\boldsymbol{\gamma}, \boldsymbol{\pi}) = U^\circ(\boldsymbol{\gamma}', \boldsymbol{\pi}').$$

*Proof.* Let $(\boldsymbol{\gamma}', \boldsymbol{\pi}')$ be such that

$$\gamma_i'[s_i, \theta] = \gamma_i[\varphi_i^\circ(s_i | \boldsymbol{\gamma}_i)] \quad \forall i \in \mathcal{N}, \, \forall s_i \in S_i, \, \forall \theta \in \Theta,$$

and

$$\pi'[\boldsymbol{s}, a] = \pi[\varphi^\circ(\boldsymbol{s} | \boldsymbol{\gamma}), a] \quad \forall \boldsymbol{s} \in \mathcal{S}, \, \forall a \in \mathcal{A}.$$

First, we show that for any agent $i \in \mathcal{N}$, $(b_i^\circ(\boldsymbol{\gamma}_i), \varphi_i^H)$ is a best response to $\boldsymbol{\gamma}'$, where $\varphi_i^H$ is the honest signal reporting function such that $\varphi_i^H(s_i) = s_i$ for all $s_i \in S_i$. Fix $i \in \mathcal{N}$. Assume, by contradiction, that there exists $b_i' \in B_i$ and $\varphi_i : S_i \to S_i$ such that

$$\sum_{s_i \in S_i} \sum_{\theta \in \Theta} \mathbb{P}^{(i)}\left(s_i, \theta | b_i'\right) \gamma_i'[\varphi_i(s_i), \theta] - c_i(b_i') > \sum_{s_i \in S_i} \sum_{\theta \in \Theta} \mathbb{P}^{(i)}\left(s_i, \theta | b_i^\circ(\boldsymbol{\gamma}_i)\right) \gamma_i'[s_i, \theta] - c_i(b_i^\circ(\boldsymbol{\gamma}_i)),$$

and let $\varphi_i' : S_i \to S_i$ such that $\varphi_i'(s_i) = \varphi^\circ(\varphi(s_i)|\boldsymbol{\gamma}_i)$. Then, we have that

$$\sum_{s_i \in S_i} \sum_{\theta \in \Theta} \mathbb{P}^{(i)}\left(s_i, \theta | b_i^\circ(\boldsymbol{\gamma}_i)\right) \gamma_i[\varphi_i^\circ(s_i|\boldsymbol{\gamma}_i), \theta] - c_i(b_i^\circ(\boldsymbol{\gamma}_i))$$

$$= \sum_{s_i \in S_i} \sum_{\theta \in \Theta} \mathbb{P}^{(i)}\left(s_i, \theta | b_i^\circ(\boldsymbol{\gamma}_i)\right) \gamma_i'[s_i, \theta] - c_i(b_i^\circ(\boldsymbol{\gamma}_i))$$

$$< \sum_{s_i \in S_i} \sum_{\theta \in \Theta} \mathbb{P}^{(i)}\left(s_i, \theta | b_i'\right) \gamma_i'[\varphi_i(s_i), \theta] - c_i(b_i')$$

$$= \sum_{s_i \in S_i} \sum_{\theta \in \Theta} \mathbb{P}^{(i)}\left(s_i, \theta | b_i'\right) \gamma_i[\varphi_i'(s_i), \theta] - c_i(b_i'),$$

which contradicts the assumption that $b_i^\circ(\boldsymbol{\gamma}_i)$, $\varphi_i^\circ(\cdot|\boldsymbol{\gamma}_i)$ is a best response of agent $i$ to scoring rule $\boldsymbol{\gamma}_i$. To conclude the proof, we show that $U^\circ(\boldsymbol{\gamma}, \boldsymbol{\pi}) = U^\circ(\boldsymbol{\gamma}', \boldsymbol{\pi}')$. Notice that

$$U^\circ(\boldsymbol{\gamma}, \boldsymbol{\pi}) = \sum_{\boldsymbol{s} \in \mathcal{S}} \sum_{\theta \in \Theta} \mathbb{P}\left(\boldsymbol{s}, \theta | \boldsymbol{b}^\circ(\boldsymbol{\gamma})\right) \left[ \left( \sum_{a \in \mathcal{A}} \pi[\varphi^\circ(\boldsymbol{s}|\boldsymbol{\gamma}), a] u(a, \theta) \right) - \sum_{i \in \mathcal{N}} \gamma_i[\varphi_i^\circ(s_i|\boldsymbol{\gamma}_i), \theta] \right] \quad \text{(24a)}$$

$$= \sum_{\boldsymbol{s} \in \mathcal{S}} \sum_{\theta \in \Theta} \mathbb{P}\left(\boldsymbol{s}, \theta | \boldsymbol{b}^\circ(\boldsymbol{\gamma})\right) \left[ \left( \sum_{a \in \mathcal{A}} \pi'[\boldsymbol{s}, a] u(a, \theta) \right) - \sum_{i \in \mathcal{N}} \gamma_i'[s_i, \theta] \right] \quad \text{(24b)}$$

$$= U^\circ(\boldsymbol{\gamma}', \boldsymbol{\pi}'). \quad \text{(24c)}$$

This concludes the proof. □

Now we are ready to prove Theorem 4.1.

**Theorem 4.1.** *There exists a game in which no uncorrelated mechanism is optimal.*

*Proof.* Consider a game in which $\mathcal{N} = \{1, 2\}$, $B_i = \{b_\triangleright, b_\triangleleft\}$ and $S_i = \{s_\triangleright, s_\triangleleft\}$, $\forall i \in \mathcal{N}$. The action costs are specified such that $c_1(b_\triangleright) = c_2(b_\triangleright) = c_2(b_\triangleleft) = 0$, $c_1(b_\triangleleft) = K \leq 1/24$. The set of states of nature is $\Theta = \{\theta_1, \theta_2\}$. The joint probability distributions are defined as in the following tables:

| $\mathbb{P}\left(\theta, \boldsymbol{s} | b_\triangleright b_\triangleright\right)$ | $s_\triangleright s_\triangleright$ | $s_\triangleright s_\triangleleft$ | $s_\triangleleft s_\triangleright$ | $s_\triangleleft s_\triangleleft$ |
|---|---|---|---|---|
| $\theta_1$ | 1/8 | 1/8 | 1/8 | 1/8 |
| $\theta_2$ | 1/8 | 1/8 | 1/8 | 1/8 |

| $\mathbb{P}\left(\theta, \boldsymbol{s} | b_\triangleright b_\triangleleft\right)$ | $s_\triangleright s_\triangleright$ | $s_\triangleright s_\triangleleft$ | $s_\triangleleft s_\triangleright$ | $s_\triangleleft s_\triangleleft$ |
|---|---|---|---|---|
| $\theta_1$ | 1/8 | 1/8 | 1/8 | 1/8 |
| $\theta_2$ | 1/8 | 1/8 | 1/8 | 1/8 |

| $\mathbb{P}\left(\theta, \boldsymbol{s} | b_\triangleleft b_\triangleright\right)$ | $s_\triangleright s_\triangleright$ | $s_\triangleright s_\triangleleft$ | $s_\triangleleft s_\triangleright$ | $s_\triangleleft s_\triangleleft$ |
|---|---|---|---|---|
| $\theta_1$ | 1/6 | 1/6 | 1/12 | 1/12 |
| $\theta_2$ | 1/12 | 1/12 | 1/6 | 1/6 |

| $\mathbb{P}\left(\theta, \boldsymbol{s} | b_\triangleleft b_\triangleleft\right)$ | $s_\triangleright s_\triangleright$ | $s_\triangleright s_\triangleleft$ | $s_\triangleleft s_\triangleright$ | $s_\triangleleft s_\triangleleft$ |
|---|---|---|---|---|
| $\theta_1$ | 1/3 | 0 | 0 | 1/6 |
| $\theta_2$ | 1/6 | 0 | 0 | 1/3 |

**Optimal Uncorrelated Mechanism.** As a first step, we characterize the set of optimal uncorrelated mechanisms for this game. Since the cost of both actions for agent 2 is 0, we can trivially set $\gamma_2[s, \theta] = 0$ for each $s \in S_2$ and $\theta \in \Theta$ to minimize payments without affecting the incentive compatibility of the mechanism. As a direct implication of Lemma D.1, there exists an optimal uncorrelated mechanism which incentivizes truthful signal reporting, thus, in order to find an optimal uncorrelated mechanism, we can enumerate among the optimal uncorrelated mechanisms incentivizing each action profile while incentivizing each agent to report truthfully her information. Furthermore, it is easy to see that, when players report their information truthfully, signal profiles $(b_\triangleright b_\triangleright)$ and $(b_\triangleright b_\triangleleft)$ induce the

same posteriors over the states of nature. The same holds for signal profile $(b_\lhd b_\rhd)$ and $(b_\lhd b_\lhd)$. Thus, we have that:

$$\max_{\boldsymbol{\pi}} \sum_{\substack{a \in \mathcal{A} \\ \boldsymbol{s} \in \mathcal{S} \\ \theta \in \Theta}} \pi[\boldsymbol{s}, a] \mathbb{P}\left(\boldsymbol{s}, \theta | b_\rhd b_\rhd\right) u(a, \theta) = \max_{\boldsymbol{\pi}} \sum_{\substack{a \in \mathcal{A} \\ \boldsymbol{s} \in \mathcal{S} \\ \theta \in \Theta}} \pi[\boldsymbol{s}, a] \mathbb{P}\left(\boldsymbol{s}, \theta | b_\rhd b_\lhd\right) u(a, \theta) = 1/2 \qquad (25)$$

$$\max_{\boldsymbol{\pi}} \sum_{\substack{a \in \mathcal{A} \\ \boldsymbol{s} \in \mathcal{S} \\ \theta \in \Theta}} \pi[\boldsymbol{s}, a] \mathbb{P}\left(\boldsymbol{s}, \theta | b_\lhd b_\rhd\right) u(a, \theta) = \max_{\boldsymbol{\pi}} \sum_{\substack{a \in \mathcal{A} \\ \boldsymbol{s} \in \mathcal{S} \\ \theta \in \Theta}} \pi[\boldsymbol{s}, a] \mathbb{P}\left(\boldsymbol{s}, \theta | b_\lhd b_\lhd\right) u(a, \theta) = 2/3. \qquad (26)$$

Moreover, fixing the action profile $\boldsymbol{b} \in \mathcal{B}$ that we want to incentivize, for any scoring rule $\boldsymbol{\gamma}_1 \in [0, M]^{B_i \times S_i \times \Theta}$, consider the incentive compatibility constraints (recall that we can trivially ignore the incentive compatibility constraints of agent 2 by setting $\boldsymbol{\gamma}_2 = \mathbf{0}$ as discussed above):

$$\sum_{s \in S_1} \sum_{\theta \in \Theta} \gamma_1[s, \theta] \mathbb{P}^{(1)}\left(s, \theta | b_1\right) - \gamma_1[\varphi(b_1, s), \theta] \mathbb{P}^{(1)}\left(s, \theta | \phi(b_1)\right) \geq c_1(b_1) - c_1(\phi(b_1)) \quad \forall \phi, \varphi \in \Phi_1,$$

while the expected utility of the principal when the agents behave honestly is

$$U^\circ(\boldsymbol{b}) = \max_{\boldsymbol{\pi}} \left\{ \sum_{\boldsymbol{s} \in \mathcal{S}} \sum_{\theta \in \Theta} \sum_{a \in \mathcal{A}} \pi[\boldsymbol{s}, a] \mathbb{P}\left(\boldsymbol{s}, \theta | \boldsymbol{b}\right) u(a, \theta) \right\} - \sum_{s \in S_1} \sum_{\theta \in \Theta} \mathbb{P}^{(1)}\left(s, \theta | b_1\right) \gamma_1[s, \theta].$$

Noticing that, for any two action profiles $\boldsymbol{b}, \boldsymbol{b}' \in \mathcal{B}$, if $b_1 = b_1'$ then the incentive compatibility does not change and that, by equations (25) and (26), $U(b_\rhd b_\rhd) = U(b_\rhd b_\lhd)$ and $U(b_\lhd b_\rhd) = U(b_\lhd b_\rhd)$, it follows that the problem can be reduced to a single-agent problem in which we have to decide whether to incentivize action $b_\rhd$ or $b_\lhd$ to agent 1. Hence, let us fix, without loss of generality, $b_\lhd$ as the action played by 2.

Since action $b_\rhd$ has a cost $c_1(b_\rhd) = 0$ for agent 1, it can be trivially incentivized by setting $\boldsymbol{\gamma}_1 = \mathbf{0}$, guaranteeing a utility for action profile $(b_\rhd, b_\lhd)$ of $U(b_\rhd b_\lhd) = 1/2$.

Consider, instead, action profile $b_\lhd b_\lhd$. Given a scoring rule $\boldsymbol{\gamma}_1$, the expected payment received by agent 1 is:

$$F_1(b_\lhd b_\lhd, \boldsymbol{\gamma}) = \sum_{s \in S_1} \sum_{\theta \in \Theta} \gamma_1[s, \theta] \mathbb{P}^{(1)}\left(s, \theta | b_\lhd\right)$$
$$= \frac{1}{3} \left[\gamma_1[s_\rhd, \theta_1] + \gamma_1[s_\lhd, \theta_2]\right] + \frac{1}{6} \left[\gamma_1[s_\lhd, \theta_1] + \gamma_1[s_\rhd, \theta_2]\right].$$

Similarly, consider the deviation $(\phi, \varphi) \in \Phi_1$ such that $\phi(b_\lhd) = b_\rhd$ and $\varphi(b_\lhd, s) = s$ for any $s \in S_1$. Then:

$$F_1^{\phi, \varphi}(b_\lhd b_\lhd, \boldsymbol{\gamma}) = \sum_{s \in S_1} \sum_{\theta \in \Theta} \gamma_1[s, \theta] \mathbb{P}^{(1)}\left(s, \theta | b_\rhd\right)$$
$$= \frac{1}{4} \left[\gamma_1[s_\rhd, \theta_1] + \gamma_1[s_\lhd, \theta_2] + \gamma_1[s_\lhd, \theta_1] + \gamma_1[s_\rhd, \theta_2]\right].$$

The associated incentive compatibility constraint is, therefore:

$$\frac{1}{12} \left[\gamma_1[s_\rhd, \theta_1] + \gamma_1[s_\lhd, \theta_2]\right] - \frac{1}{12} \left[\gamma_1[s_\rhd, \theta_2] + \gamma_1[s_\lhd, \theta_1]\right] \geq K.$$

Repeating the same procedure for all possible deviation functions, we obtain the following incentive compatibility constraints:

$$\frac{1}{12} \left[\gamma_1[s_\rhd, \theta_1] + \gamma_1[s_\lhd, \theta_2]\right] - \frac{1}{12} \left[\gamma_1[s_\rhd, \theta_2] + \gamma_1[s_\lhd, \theta_1]\right] \geq K$$
$$\frac{1}{3}\gamma_1[s_\lhd, \theta_2] + \frac{1}{6}\gamma_1[s_\lhd, \theta_1] - \frac{1}{6}\gamma_1[s_\rhd, \theta_1] - \frac{1}{3}\gamma_1[s_\rhd, \theta_2] \geq K$$
$$\frac{1}{3}\gamma_1[s_\rhd, \theta_2] + \frac{1}{6}\gamma_1[s_\rhd, \theta_1] - \frac{1}{6}\gamma_1[s_\lhd, \theta_1] - \frac{1}{3}\gamma_1[s_\lhd, \theta_2] \geq K$$
$$\frac{1}{6} \left[\gamma_1[s_\rhd, \theta_1] + \gamma_1[s_\lhd, \theta_2]\right] - \frac{1}{6} \left[\gamma_1[s_\lhd, \theta_1] + \gamma_1[s_\rhd, \theta_2]\right] \geq 0.$$

Thus, it is easy to check that in order to minimize the expected payment, while satisfying the above constraints it is enough to select a scoring rule such that

$$\gamma_1[s_\triangleright, \theta_1] = \gamma_1[s_\triangleleft, \theta_2] = 6K$$
$$\gamma_1[s_\triangleright, \theta_2] = 0$$
$$\gamma_1[s_\triangleleft, \theta_1] = 0.$$

Since $K \leq 1/24$ by assumption, we have that the optimal utility achievable by using an uncorrelated mechanism is $2/3 - 4K$.

**Optimal Correlated Mechanism.** Consider the correlated mechanism $(\boldsymbol{\mu}, \boldsymbol{\gamma}, \boldsymbol{\pi})$ defined such that $\mu[b_\triangleleft b_\triangleleft] = \varepsilon$ and $\mu[b_\triangleleft b_\triangleright] = 1 - \varepsilon$ for some $\varepsilon = 12K/5 = 1/10$. Similarly to before, since all actions for agent 2 have cost 0, we can be incentive compatible by setting $\boldsymbol{\gamma}_2 = \mathbf{0}$. The scoring rule for agent 1 is defined as:

$$\gamma_1[\boldsymbol{b}, \boldsymbol{s}, \theta] = 0 \qquad \forall \boldsymbol{b} \in \{b_\triangleright b_\triangleright, b_\triangleright b_\triangleleft, b_\triangleleft b_\triangleright\}, \, \forall \boldsymbol{s} \in \mathcal{S}, \, \forall \theta \in \Theta$$
$$\gamma_1[b_\triangleleft b_\triangleleft, \boldsymbol{s}, \theta_1] = 0 \qquad \forall \boldsymbol{s} \in \mathcal{S} \setminus \{s_\triangleright s_\triangleright\}$$
$$\gamma_1[b_\triangleleft b_\triangleleft, \boldsymbol{s}, \theta_2] = 0 \qquad \forall \boldsymbol{s} \in \mathcal{S} \setminus \{s_\triangleleft s_\triangleleft\}$$
$$\gamma_1[b_\triangleleft b_\triangleleft, s_\triangleright s_\triangleright, \theta_1] = \gamma_1[b_\triangleleft b_\triangleleft, s_\triangleleft s_\triangleleft, \theta_2] = 1.$$

Let $\boldsymbol{\pi}$ be such that $\boldsymbol{\pi} \in arg\max_{\boldsymbol{\pi}'}\{U(\boldsymbol{\mu}, \boldsymbol{\gamma}, \boldsymbol{\pi}')\}$ Simple calculations show that $U(\boldsymbol{\mu}, \boldsymbol{\gamma}, \boldsymbol{\pi}) = 2/3 - 2\varepsilon/3$. It is possible to show that the above mechanism is incentive compatible. In particular, the expected payment received by agent 1 is:

$$F_1(\boldsymbol{\mu}, \boldsymbol{\gamma}) = \frac{2}{3}\varepsilon.$$

Consider any two deviation functions $(\phi, \varphi) \in \Phi_i$ such that $\phi(b_\triangleleft) = b_\triangleright$. The expected payment received by agent 1 when she deviates according to $(\phi, \varphi)$ is:

$$F_i^{\phi, \varphi}(\boldsymbol{\mu}, \boldsymbol{\gamma}) = \varepsilon/4 \quad \forall \varphi.$$

Similarly, for any two deviation functions $(\phi, \varphi) \in \Phi_i$ such that $\phi(b_\triangleleft) = b_\triangleleft$, the expected payment is

$$F_1^{\phi, \varphi}(\boldsymbol{\mu}, \boldsymbol{\gamma}) = \begin{cases} \varepsilon/3 & \text{if } \varphi(b_\triangleleft, s_\triangleright) = s_\triangleright \wedge \varphi(b_\triangleleft, s_\triangleleft) = s_\triangleright \\ 2\varepsilon/3 & \text{if } \varphi(b_\triangleleft, s_\triangleright) = s_\triangleright \wedge \varphi(b_\triangleleft, s_\triangleleft) = s_\triangleleft \\ 0 & \text{if } \varphi(b_\triangleleft, s_\triangleright) = s_\triangleleft \wedge \varphi(b_\triangleleft, s_\triangleleft) = s_\triangleright \\ \varepsilon/3 & \text{otherwise.} \end{cases}$$

Thus, the incentive compatibility constraints that are yielded are:

$$\frac{5\varepsilon}{12} \geq K$$
$$\frac{\varepsilon}{3} \geq 0,$$

which are trivially satisfied since, by assumption, $\varepsilon = 12K/5 > 0$. Hence, the expected utility of the principal is

$$U(\boldsymbol{\mu}, \boldsymbol{\gamma}, \boldsymbol{\pi}) = \frac{2}{3} - \frac{2}{3}\varepsilon = \frac{2}{3} - \frac{2}{3}\frac{12}{5}K > \frac{2}{3} - 4K.$$

Any optimal correlated mechanism $(\boldsymbol{\mu}^\star, \boldsymbol{\gamma}^\star, \boldsymbol{\pi}^\star)$ achieves a utility $U(\boldsymbol{\mu}^\star, \boldsymbol{\gamma}^\star, \boldsymbol{\pi}^\star) \geq U(\boldsymbol{\mu}, \boldsymbol{\gamma}, \boldsymbol{\pi}) > 2/3 - 4K$, hence we can conclude that in this game there does not exist an uncorrelated mechanism which is optimal for optimization problem (1). This concludes the proof

$\square$

**Theorem 4.2.** *Assume there for each $i \in \mathcal{N}$, $\theta \in \Theta$ and $b_i \in B_i$, there exists a probability distribution $\psi_i(\cdot|b_i) \in \Delta(S_i)$ such that $\forall \boldsymbol{b} \in \mathcal{B}$, $\forall \boldsymbol{s} \in \mathcal{S}$ and $\forall \theta \in \Theta$, $\mathbb{P}(\boldsymbol{s}, \theta|\boldsymbol{b}) = p[\theta] \prod_{i \in \mathcal{N}} \psi_i(s_i|b_i, \theta)$. Then, there exists a mechanism $(\boldsymbol{\gamma}, \boldsymbol{\pi}) \in \mathcal{U}$ that is optimal among correlated mechanisms.*

*Proof.* Let $(\boldsymbol{\mu}^{\star}, \boldsymbol{\gamma}^{\star}, \boldsymbol{\pi}^{\star}) \in \mathcal{C}$ be an optimal correlated mechanism. For each $\boldsymbol{b} \in \mathcal{B}$, let $G(\boldsymbol{b})$ be such that

$$G(\boldsymbol{b}) = \sum_{\boldsymbol{s} \in \mathcal{S}} \sum_{\theta \in \Theta} \mathbb{P}\left(\boldsymbol{s}, \theta | \boldsymbol{b}\right) \left[ \sum_{a \in \mathcal{A}} \pi^{\star}[\boldsymbol{b}, \boldsymbol{s}, a] u(a, \theta) - \sum_{i \in \mathcal{N}} \gamma_i^{\star}[\boldsymbol{b}, \boldsymbol{s}, \theta] \right].$$

It is easy to see that it holds $U(\boldsymbol{\mu}^{\star}, \boldsymbol{\gamma}^{\star}, \boldsymbol{\pi}^{\star}) = \sum_{\boldsymbol{b} \in \mathcal{B}} \mu^{\star}[\boldsymbol{b}] G(\boldsymbol{b})$. Let $\bar{\boldsymbol{b}} \in arg \max_{\boldsymbol{b} \in \mathcal{B}: \mu^{\star}[\boldsymbol{b}] > 0} \{G(\boldsymbol{b})\}$ be the action profile yielding optimal utility to the principal and define the uncorrelated scoring rule $\boldsymbol{\gamma} = (\boldsymbol{\gamma}_1, ..., \boldsymbol{\gamma}_n)$ be such that

$$\gamma_i[s_i, \theta] = \frac{1}{\sum_{\boldsymbol{b}_{-i} \in \mathcal{B}_{-i}} \mu^{\star}[(\bar{b}_i, \boldsymbol{b}_{-i})]} \sum_{\substack{\boldsymbol{b}_{-i} \in \mathcal{B}_{-i} \\ \boldsymbol{s}_{-i} \in \mathcal{S}_{-i}}} \mu^{\star}[(\bar{b}_i, \boldsymbol{b}_{-i})] \prod_{j \in \mathcal{N} \setminus \{i\}} \psi_j(s_j | b_j, \theta) \gamma_i^{\star}[(\bar{b}_i, \boldsymbol{b}_{-i}), (s_i, \boldsymbol{s}_{-i}), \theta].$$

Furthermore, let $\boldsymbol{\pi}$ be such that $\boldsymbol{\pi}[\boldsymbol{s}, a] = \boldsymbol{\pi}^{\star}[\bar{\boldsymbol{b}}, \boldsymbol{s}, a]$. It is possible to show that for each $i \in \mathcal{N}$, $(\hat{b}_i, \varphi_i^H)$ is a best response for agent $i$ to uncorrelated mechanism $(\boldsymbol{\gamma}, \boldsymbol{\pi})$, where $\varphi_i^H : S_i \to S_i$ is the honest signal reporting policy, *i.e.*, such that $\varphi_i^H(s_i) = s_i$ for all $s_i \in S_i$. In particular, notice that the expected payment received by agent $i$ when the principal commits to mechanism $(\boldsymbol{\mu}^{\star}, \boldsymbol{\gamma}^{\star}, \boldsymbol{\pi}^{\star})$ can be written as

$$F_i(\boldsymbol{\mu}^{\star}, \boldsymbol{\gamma}^{\star}) = \sum_{\substack{b_i \in B_i \\ s_i \in S_i \\ \theta \in \Theta}} \mathbb{P}^{(i)}\left(s_i, \theta | b_i\right) \left( \sum_{\substack{\boldsymbol{b}_{-i} \in \mathcal{B}_{-i} \\ \boldsymbol{s}_{-i} \in \mathcal{S}_{-i}}} \mu^{\star}[(b_i, \boldsymbol{b}_{-i})] \prod_{j \in \mathcal{N} \setminus \{i\}} \psi_j(s_j | b_j, \theta) \gamma_i^{\star}[(b_i, \boldsymbol{b}_{-i}), (s_i, \boldsymbol{s}_{-i}), \theta] \right)$$

and similarly, for each $(\phi, \varphi) \in \Phi_i$,

$$F_i^{\phi, \varphi}(\boldsymbol{\mu}^{\star}, \boldsymbol{\gamma}^{\star}) =$$

$$\sum_{\substack{b_i \in B_i \\ s_i \in S_i \\ \theta \in \Theta}} \mathbb{P}^{(i)}\left(s_i, \theta | \phi(b_i)\right) \left( \sum_{\substack{\boldsymbol{b}_{-i} \in \mathcal{B}_{-i} \\ \boldsymbol{s}_{-i} \in \mathcal{S}_{-i}}} \mu^{\star}[(b_i, \boldsymbol{b}_{-i})] \prod_{j \in \mathcal{N} \setminus \{i\}} \psi_j(s_j | b_j, \theta) \gamma_i^{\star}[(b_i, \boldsymbol{b}_{-i}), (\varphi(b_i, s_i), \boldsymbol{s}_{-i}), \theta] \right).$$

For any $b_i' \in B_i$ and $\varphi_i' : S_i \to S_i$, let $(\phi, \varphi) \in \Phi_i$ be the deviation functions such that $\phi(b_i) = b_i$ for all $b_i \in B_i \setminus \{\bar{b}_i\}$, $\phi(\bar{b}_i) = b_i'$, $\varphi(b_i, s_i) = s_i$ for all $b_i \in B_i \setminus \{\bar{b}_i\}$ and $s_i \in S_i$, and $\varphi(\bar{b}_i, s_i) = \varphi_i'(s_i)$ for all $s_i \in S_i$. Then, by substituting the definition of $\boldsymbol{\gamma}_i$ into the expressions for $F_i$ and $F_i^{\phi, \varphi}$, we can write the following:

$$F_i(\boldsymbol{\mu}^{\star}, \boldsymbol{\gamma}^{\star}) - F_i^{\phi, \varphi}(\boldsymbol{\mu}^{\star}, \boldsymbol{\gamma}^{\star})$$

$$= \left( \sum_{\boldsymbol{b}_{-i} \in \mathcal{B}_{-i}} \mu^{\star}[(\bar{b}_i, \boldsymbol{b}_{-i})] \right) \sum_{s_i \in S_i} \sum_{\theta \in \Theta} \left[ \mathbb{P}^{(i)}\left(s_i, \theta | \bar{b}_i\right) \gamma_i[s_i, \theta] - \mathbb{P}^{(i)}\left(s_i, \theta | b_i'\right) \gamma_i[\varphi_i'(s_i), \theta] \right] \quad (27a)$$

$$\geq \sum_{\boldsymbol{b}_{-i} \in \mathcal{B}_{-i}} \mu^{\star}[(\bar{b}_i, \boldsymbol{b}_{-i})] C_i(\bar{b}_i, b_i'), \quad (27b)$$

where 27b follows from incentive compatibility of $(\boldsymbol{\mu}^{\star}, \boldsymbol{\pi}^{\star}, \boldsymbol{\gamma}^{\star})$. By dividing for $\sum_{\boldsymbol{b}_{-i} \in \mathcal{B}_{-i}} \mu^{\star}[(\bar{b}_i, \boldsymbol{b}_{-i})]$, we get

$$\sum_{s_i \in S_i} \sum_{\theta \in \Theta} \left[ \mathbb{P}^{(i)}\left(s_i, \theta | \bar{b}_i\right) \gamma_i[s_i, \theta] - \mathbb{P}^{(i)}\left(s_i, \theta | b_i'\right) \gamma_i[\varphi_i'(s_i), \theta] \right] \geq C_i(b_i, b_i'),$$

which proves that $(\bar{b}_i, \varphi_i^H)$ is a best response of agent $i$ to mechanism $(\boldsymbol{\gamma}, \boldsymbol{\pi}) \in \mathcal{U}$. The result follows by noting that $U^{\circ}(\boldsymbol{\gamma}, \boldsymbol{\pi}) = G(\bar{\boldsymbol{b}}) \geq U(\boldsymbol{\mu}^{\star}, \boldsymbol{\gamma}^{\star}, \boldsymbol{\pi}^{\star})$. $\qquad \square$

## E  PROOFS OMITTED FROM SECTION 6

**Lemma 6.1.** *The clean event $\mathcal{E}_p$ holds with probability at least $1 - \frac{\delta}{2}$.*

*Proof.* By Hoeffding bound, we have that

$$\mathbb{P}\left(\left|\zeta_{\boldsymbol{b}}[\boldsymbol{s}, \theta] - \mathbb{P}\left(\boldsymbol{s}, \theta | \boldsymbol{b}\right)\right| \leq \nu\right) \geq 1 - \frac{\delta}{K},$$

$\forall \boldsymbol{b} \in \mathcal{B}, \forall \boldsymbol{s} \in \mathcal{S}, \forall \theta \in \Theta, \forall t \in \cup_{\boldsymbol{b}} \mathcal{T}_p(\boldsymbol{b})$, and similarly,

$$\mathbb{P}\left(\left|\sigma_{b_i,s_i}^{(i)}[\theta] - \mathbb{P}^{(i)}\left(\theta | b_i, s_i\right)\right| \leq \varrho\right) \geq 1 - \frac{\delta}{K},$$

$\forall i \in \mathcal{N}, \forall b_i \in B_i, \forall s_i \in S_i \forall \theta \in \Theta, \forall t \in \cup_{\boldsymbol{b}} \mathcal{T}_p(\boldsymbol{b})$.

Moreover, let $\boldsymbol{b}$ be such that $|\mathcal{T}_p(\boldsymbol{b})| \geq \kappa$. Notice that for any $i \in \mathcal{N}$ and $s_i \in S_i$,

$$|\mathcal{T}_p^{(i)}(b_i, s_i)| = \sum_{\substack{\boldsymbol{b}' \in \mathcal{B}: \\ b_i' = b_i}} \sum_{t \in \mathcal{T}_p(\boldsymbol{b}')} \mathbb{1}[\tilde{s}_i^t = s_i].$$

Therefore, $|\mathcal{T}_p^{(i)}(b_i, s_i)|$ can be seen as the sum of a sequence of Bernoulli random variables with parameter $\mathbb{P}^{(i)}(s_i | b_i)$. Additionally,

$$\mathbb{E}\left[|\mathcal{T}_p^{(i)}(b_i, s_i)|\right] = \sum_{\substack{\boldsymbol{b}' \in \mathcal{B}: \\ b_i' = b_i}} \sum_{t \in \mathcal{T}_p(\boldsymbol{b}')} \mathbb{P}^{(i)}(s_i | b_i) \geq \iota \sum_{\substack{\boldsymbol{b}' \in \mathcal{B}: \\ b_i' = b_i}} |\mathcal{T}_p(\boldsymbol{b}')| \geq \iota |\mathcal{T}_p(\boldsymbol{b})|. \qquad (28)$$

By applying Chernoff bound together with Equation(28), we get that $\forall \boldsymbol{b} \in \mathcal{B}, \forall i \in \mathcal{N}, \forall s_i \in S_i$, $\forall t \in \cup_{\boldsymbol{b}} \mathcal{T}_p(\boldsymbol{b})$,

$$\mathbb{P}\left(|\mathcal{T}_p^{(i)}(b_i, s_i)| \geq \frac{1}{2}\iota|\mathcal{T}_p(\boldsymbol{b})|\right) \geq 1 - \exp\left(-\frac{\iota^2 |\mathcal{T}_p(\boldsymbol{b})|}{2}\right)$$

$$\geq 1 - \exp\left(-\frac{\iota^2}{2}\kappa\right)$$

$$\geq 1 - \exp\left(-\ln(2K/\delta)\frac{289m^2}{4\ell^2}\right)$$

$$\geq 1 - \frac{\delta}{K}$$

The Lemma follows by applying a union bound with $K = 6|\mathcal{B}|T|\mathcal{S}|nm$.

$\square$

**Lemma 6.2.** *Assume the clean event $\mathcal{E}_p$ holds. Then, at the end of the execution of* ESTIMATEPROB, *for each $\boldsymbol{b} \in \mathcal{B}$ it holds that $|\mathcal{T}_p(\boldsymbol{b})| \leq \max\{N_1, \kappa\}$.*

*Proof.* The prove this Lemma, we show that for any $\boldsymbol{b} \in \mathcal{B}$, if $|\mathcal{T}_p(\boldsymbol{b})| = \max\{N_i, \kappa\}$, then the condition $|\mathcal{T}_p(\boldsymbol{b})| < N_1 \vee \bar{\varrho} > \frac{d}{13m}$ is not satisfied, and hence the while loop in Algorithm 2 terminates. To prove this, we show that if $|\mathcal{T}(\boldsymbol{b})| = \max\{N_i, \kappa\}$, then $\bar{\varrho} \leq \frac{d}{13m}$.

Fix $\boldsymbol{b} \in \mathcal{B}$. Since the clean event $\mathcal{E}_p$ holds, and since, by assumption, $|\mathcal{T}_p(\boldsymbol{b})| \geq \kappa$ we have that

$$|\mathcal{T}_p^{(i)}(b_i, s_i)| \geq \frac{1}{2}\iota|\mathcal{T}_p(\boldsymbol{b})| \geq \frac{1}{2}\iota\kappa \quad \forall i \in \mathcal{N}, \forall s_i \in S_i. \qquad (29)$$

Furthermore, let $(i, s_i, s_i') \in arg\min_{i \in \mathcal{N}, s_i, s_i' \in S_i} ||\boldsymbol{\sigma}_{b_i,s_i}^{(i)} - \boldsymbol{\sigma}_{b_i,s_i'}^{(i)}||_2^2$. By definition, we have that $\underline{d} = ||\boldsymbol{\sigma}_{b_i,s_i}^{(i)} - \boldsymbol{\sigma}_{b_i,s_i'}^{(i)}||_2^2$. By definition of $\bar{\varrho}$ and by assumption that the clean event holds, for all $\theta \in \Theta$ we have that

$$|\mathbb{P}\left(\theta | b_i, s_i\right) - \mathbb{P}\left(\theta | b_i, s_i'\right)| \leq |\sigma_{b_i,s_i}^{(i)}[\theta] - \sigma_{b_i,s_i'}^{(i)}[\theta]| + 2\bar{\varrho},$$

and thus

$$\sum_{\theta \in \Theta} |\mathbb{P}\left(\theta | b_i, s_i\right) - \mathbb{P}\left(\theta | b_i, s_i'\right)|^2 \leq ||\boldsymbol{\sigma}_{b_i,s_i}^{(i)} - \boldsymbol{\sigma}_{b_i,s_i'}^{(i)}||_2^2 + 4m\bar{\varrho}^2 = \underline{d} + 4m\bar{\varrho}^2.$$

Rearranging, we get the following chain of inequalities:

$$\underline{d} \geq \sum_{\theta \in \Theta} |\mathbb{P}(\theta|b_i, s_i) - \mathbb{P}(\theta|b_i, s_i')|^2 - 4m\bar{\varrho}^2 \tag{30a}$$

$$\geq \ell - 4m\bar{\varrho} + 13m\bar{\varrho} - 13m\bar{\varrho} \tag{30b}$$

$$= \ell - 17m\bar{\varrho} + 13m\bar{\varrho} \tag{30c}$$

$$= \ell - 17m\sqrt{\frac{\ln(2K/\delta)}{2|\mathcal{T}_p^{(i)}(b_i, s_i)|}} + 13m\bar{\varrho} \tag{30d}$$

$$\geq \ell - 17m\sqrt{\frac{\ln(2K/\delta)}{\iota\kappa}} + 13m\bar{\varrho} \tag{30e}$$

$$= \ell - 17m\sqrt{\frac{2\iota\ell^2}{289m^2}} + 13m\bar{\varrho} \tag{30f}$$

$$\geq \ell\left(1 - \sqrt{2\iota}\right) + 13m\bar{\varrho} \tag{30g}$$

$$\geq 13m\bar{\varrho}, \tag{30h}$$

where Equation (30b) follows from the definition of $\ell$ and from the fact that $\bar{\varrho} \leq 1$ whenever $|\mathcal{T}_p(\boldsymbol{b})| \geq \kappa$ and Equation (30d) follows by definition of $\bar{\varrho}$. Rearranging, we get $\bar{\varrho} \leq \frac{d}{13m}$, which concludes the proof. $\qquad\square$

## F  PROOFS OMITTED FROM SECTION 8

**Lemma F.1.** *Let $(\boldsymbol{x}^\star, \boldsymbol{y}^\star, \boldsymbol{\mu}^\star, \boldsymbol{z}^\star)$ be an optimal solution to $LP(\mathbb{P}, C, 0)$. Then, if clean events $\mathcal{E}_p$ and $\mathcal{E}_d$ hold, there exists $\boldsymbol{z}'$ such that $(\boldsymbol{x}^\star, \boldsymbol{y}^\star, \boldsymbol{\mu}^\star, \boldsymbol{z}')$ is a feasible solution for $LP(\boldsymbol{\zeta}, \Lambda, 2M|\mathcal{S}|m\nu + \chi)$.*

*Proof.* Let $\boldsymbol{z}'$ be such that $z_i'[b_i, b_i', s_i] = z_i^\star[b_i, b_i', s_i] + M|\mathcal{S}_{-i}|m\nu$, for all $i \in \mathcal{N}$, $b_i, b_i' \in B_i$ and $s_i \in S_i$. Trivially, $(\boldsymbol{x}^\star, \boldsymbol{y}^\star, \boldsymbol{\mu}^\star, \boldsymbol{z}')$ satisfy constraints (2d) and (2e) of $LP(\boldsymbol{\zeta}, \Lambda, 2M|\mathcal{S}|m\nu + \chi)$. Consider now constraint (2c). By feasibility of $(\boldsymbol{x}^\star, \boldsymbol{y}^\star, \boldsymbol{\mu}^\star, \boldsymbol{z}^\star)$, we have that

$$z_i^\star[b_i, b_i', s_i] \geq f_i(\boldsymbol{x}_i^\star|b_i, b_i', s_i, s_i', \mathbb{P}) \tag{31a}$$

$$= \sum_{\substack{\boldsymbol{b}_{-i} \in \mathcal{B}_i \\ \boldsymbol{s}_{-i} \in \mathcal{S}_{-i} \\ \theta \in \Theta}} x_i^\star[(b_i, \boldsymbol{b}_{-i}), (s_i', \boldsymbol{s}_{-i}), \theta]\mathbb{P}((s_i, \boldsymbol{s}_{-i}), \theta|(b_i', \boldsymbol{b}_{-i})) \tag{31b}$$

$$\geq \sum_{\substack{\boldsymbol{b}_{-i} \in \mathcal{B}_i \\ \boldsymbol{s}_{-i} \in \mathcal{S}_{-i} \\ \theta \in \Theta}} x_i^\star[(b_i, \boldsymbol{b}_{-i}), (s_i', \boldsymbol{s}_{-i}), \theta]\left(\zeta_{(b_i', \boldsymbol{b}_{-i})}[(s_i, \boldsymbol{s}_{-i}), \theta] - \nu\right) \tag{31c}$$

$$\geq f_i(\boldsymbol{x}_i^\star|b_i, b_i', s_i, s_i', \boldsymbol{\zeta}) - M|\mathcal{S}_{-i}|m\nu. \tag{31d}$$

Rearranging and substituting the definition of $\boldsymbol{z}'$, we get $z_i'[b_i, b_i', s_i] \geq f_i(\boldsymbol{x}_i^\star|b_i, b_i', s_i, s_i', \boldsymbol{\zeta})$, thus proving that constraint (2c) is satisfied.

Let us consider constraint (2b). By feasibility of $(\boldsymbol{x}^\star, \boldsymbol{y}^\star, \boldsymbol{\mu}^\star, \boldsymbol{z}^\star)$, for any $i \in \mathcal{N}$ and $b_i, b_i' \in B_i$, the following holds:

$$\sum_{\boldsymbol{b}_{-i} \in \mathcal{B}_{-i}} \mu^\star[(b_i, \boldsymbol{b}_{-i})] C_i(b_i, b_i') \leq \sum_{s \in S_i} [f_i(\boldsymbol{x}_i^\star | b_i, s_i, \mathbb{P}) - z_i^\star[b_i, b_i', s]] \tag{32a}$$

$$= \sum_{\substack{\boldsymbol{b}_{-i} \in \mathcal{B}_i \\ \boldsymbol{s} \in \mathcal{S} \\ \theta \in \Theta}} [x_i^\star[(b_i, \boldsymbol{b}_{-i}, \boldsymbol{s}, \theta)] \mathbb{P}(\boldsymbol{s}, \theta | (b_i, \boldsymbol{b}_{-i}))] - \sum_{s_i \in S_i} z_i^\star[b_i, b_i', s_i] \tag{32b}$$

$$\leq \sum_{\substack{\boldsymbol{b}_{-i} \in \mathcal{B}_i \\ \boldsymbol{s} \in \mathcal{S} \\ \theta \in \Theta}} [x_i^\star[(b_i, \boldsymbol{b}_{-i}, \boldsymbol{s}, \theta)] (\zeta_{(b_i, \boldsymbol{b}_{-i})}[\boldsymbol{s}, \theta] + \nu)] - \sum_{s_i \in S_i} z_i^\star[b_i, b_i', s_i] \tag{32c}$$

$$= M|\mathcal{S}|m\nu + \sum_{s_i \in S_i} [f_i(\boldsymbol{x}_i^\star | b_i, s_i, \boldsymbol{\zeta}) - z_i'[b_i, b_i', s_i] + M|\mathcal{S}_{-i}|m\nu] \tag{32d}$$

$$= 2M|\mathcal{S}|m\nu + \sum_{s_i \in S_i} [f_i(\boldsymbol{x}_i^\star | b_i, s_i, \boldsymbol{\zeta}) - z_i'[b_i, b_i', s_i]]. \tag{32e}$$

Furthermore, notice that

$$\sum_{\boldsymbol{b}_{-i} \in \mathcal{B}_{-i}} \mu^\star[(b_i, \boldsymbol{b}_{-i})] C_i(b_i, b_i') \geq \sum_{\boldsymbol{b}_{-i} \in \mathcal{B}_{-i}} \mu^\star[(b_i, \boldsymbol{b}_{-i})] (\Lambda_i(b_i, b_i') - \chi) \tag{33a}$$

$$\geq \sum_{\boldsymbol{b}_{-i} \in \mathcal{B}_{-i}} [\mu^\star[(b_i, \boldsymbol{b}_{-i})] \Lambda_i(b_i, b_i')] - \chi. \tag{33b}$$

Putting all together, we get

$$\sum_{s_i \in S_i} [f_i(\boldsymbol{x}_i^\star | b_i, s_i, \boldsymbol{\zeta}) - z_i'[b_i, b_i', s_i]] \geq \sum_{\boldsymbol{b}_{-i} \in \mathcal{B}_{-i}} [\mu^\star[(b_i, \boldsymbol{b}_{-i})] \Lambda_i(b_i, b_i')] - (\chi + 2M|\mathcal{S}|m\nu),$$

which gives the desired result. $\qquad\square$

**Lemma 8.1.** *Assume clean events $\mathcal{E}_c$ and $\mathcal{E}_p$ hold. Let $(\boldsymbol{\mu}, \boldsymbol{\gamma}, \boldsymbol{\pi})$ be an optimal solution to $\mathrm{LP}(\boldsymbol{\zeta}, \Lambda, \varepsilon)$, where $\varepsilon = 2M|\mathcal{S}|m\nu + \chi$. Then, letting $\lambda = 2M|\mathcal{S}|m(k+1)(\nu + \chi)$, it holds that:*

$$F_i(\boldsymbol{\mu}, \boldsymbol{\gamma}) - F_i^{\phi, \varphi}(\boldsymbol{\mu}, \boldsymbol{\gamma}) \geq \sum_{\boldsymbol{b} \in \mathcal{B}} \tilde{\mu}[\boldsymbol{b}] C_i(b_i, \phi(b_i)) - \lambda \qquad \forall i \in \mathcal{N}, \forall (\phi, \varphi) \in \Phi_i.$$

*Proof.* Let $(\tilde{\boldsymbol{x}}^t, \tilde{\boldsymbol{y}}^t, \tilde{\boldsymbol{\mu}}^t, \tilde{\boldsymbol{z}}^t)$ be the vectors from which mechanism $(\tilde{\boldsymbol{\mu}}^t, \tilde{\boldsymbol{\gamma}}^t, \tilde{\boldsymbol{\pi}}^t)$ is obtained, *i.e.,* an optimal solution to $\mathrm{LP}(\boldsymbol{\zeta}, \Lambda, \varepsilon)$. Fix an agent $i \in \mathcal{N}$ and a deviation $(\phi, \varphi) \in \Phi_i$. Notice that:

$$F_i(\tilde{\boldsymbol{\mu}}^t, \tilde{\boldsymbol{\gamma}}^t) = \sum_{\substack{\boldsymbol{b} \in \mathcal{B} \\ \boldsymbol{s} \in \mathcal{S} \\ \theta \in \Theta}} \tilde{\mu}^t[\boldsymbol{b}] \tilde{\gamma}_i^t[\boldsymbol{b}, \boldsymbol{s}, \theta] \mathbb{P}(\boldsymbol{s}, \theta | \boldsymbol{b}) \tag{34a}$$

$$\geq \sum_{\substack{\boldsymbol{b} \in \mathcal{B} \\ \boldsymbol{s} \in \mathcal{S} \\ \theta \in \Theta}} [\tilde{\mu}^t[\boldsymbol{b}] \tilde{\gamma}_i^t[\boldsymbol{b}, \boldsymbol{s}, \theta] \zeta_{\boldsymbol{b}}[\boldsymbol{s}, \theta]] - M|\mathcal{S}|m\nu \tag{34b}$$

$$= \sum_{\substack{\boldsymbol{b} \in \mathcal{B} \\ \boldsymbol{s} \in \mathcal{S} \\ \theta \in \Theta}} [\tilde{x}_i^t[\boldsymbol{b}, \boldsymbol{s}, \theta] \zeta_{\boldsymbol{b}}[\boldsymbol{s}, \theta]] - M|\mathcal{S}|m\nu \tag{34c}$$

$$= \sum_{b_i \in B_i} \sum_{s_i \in S_i} f_i(\tilde{\boldsymbol{x}}_i^t | b_i, s_i, \boldsymbol{\zeta}) - M|\mathcal{S}|m\nu. \tag{34d}$$

Similarly,

$$F_i^{\phi,\varphi}(\tilde{\boldsymbol{\mu}}^t, \tilde{\boldsymbol{\gamma}}^t) = \sum_{\substack{\boldsymbol{b}\in\mathcal{B}\\ \boldsymbol{s}\in\mathcal{S}\\ \theta\in\Theta}} \tilde{\mu}^t[\boldsymbol{b}]\tilde{\gamma}_i^t[\boldsymbol{b}, (\varphi(b_i, s_i), \boldsymbol{s}_{-i}), \theta]\mathbb{P}\left(\boldsymbol{s}, \theta|(\phi(b_i), \boldsymbol{b}_{-i})\right) \tag{35a}$$

$$\leq \sum_{\substack{\boldsymbol{b}\in\mathcal{B}\\ \boldsymbol{s}\in\mathcal{S}\\ \theta\in\Theta}} \left[\tilde{\mu}^t[\boldsymbol{b}]\tilde{\gamma}_i^t[\boldsymbol{b}, (\varphi(b_i, s_i), \boldsymbol{s}_{-i}), \theta]\zeta_{(\phi(b_i), \boldsymbol{b}_{-i})}[\boldsymbol{s}, \theta]\right] + M|\mathcal{S}|m\nu \tag{35b}$$

$$= \sum_{\substack{\boldsymbol{b}\in\mathcal{B}\\ \boldsymbol{s}\in\mathcal{S}\\ \theta\in\Theta}} \left[\tilde{x}_i^t[\boldsymbol{b}, (\varphi(b_i, s_i), \boldsymbol{s}_{-i}), \theta]\zeta_{(\phi(b_i), \boldsymbol{b}_{-i})}[\boldsymbol{s}, \theta]\right] + M|\mathcal{S}|m\nu \tag{35c}$$

$$= \sum_{b_i\in B_i} \sum_{s_i\in S_i} \left[f_i(\tilde{\boldsymbol{x}}_i^t|b_i, \phi(b_i), s_i, \varphi(b_i, s_i))\right] + M|\mathcal{S}|m\nu \tag{35d}$$

$$\leq \sum_{b_i\in B_i} \sum_{s_i\in S_i} \left[\sum_{s_i\in S_i} \tilde{z}_i^t[b_i, \phi(b_i), s_i]\right] + M|\mathcal{S}|m\nu. \tag{35e}$$

Thus, by feasibility of $(\tilde{\boldsymbol{x}}^t, \tilde{\boldsymbol{y}}^t, \tilde{\boldsymbol{\mu}}^t, \tilde{\boldsymbol{z}}^t)$, we have that:

$$F_i(\tilde{\boldsymbol{\mu}}^t, \tilde{\boldsymbol{\gamma}}^t) - F_i^{\phi,\varphi}(\tilde{\boldsymbol{\mu}}^t, \tilde{\boldsymbol{\gamma}}^t) \geq \sum_{b_i\in B_i} \left[\sum_{s_i\in S_i} f_i(\tilde{\boldsymbol{x}}_i^t|b_i, s_i, \boldsymbol{\zeta}) - \tilde{z}_i^t[b_i, \phi(b_i), s_i]\right] - 2M|\mathcal{S}|m\nu \tag{36a}$$

$$\geq \sum_{\boldsymbol{b}\in\mathcal{B}} \tilde{\mu}^t[\boldsymbol{b}]\Lambda_i(b_i, \phi(b_i)) - 2M|\mathcal{S}|m\,(k+1)\,\nu - k\chi \tag{36b}$$

$$\geq \sum_{\boldsymbol{b}\in\mathcal{B}} \tilde{\mu}^t[\boldsymbol{b}]C_i(b_i, \phi(b_i)) - 2M|\mathcal{S}|m\,(k+1)\,\nu - (k+1)\chi \tag{36c}$$

$$\geq \sum_{\boldsymbol{b}\in\mathcal{B}} \tilde{\mu}^t[\boldsymbol{b}]C_i(b_i, \phi(b_i)) - 2M|\mathcal{S}|m\,(k+1)\,(\nu+\chi). \tag{36d}$$

This concludes the proof. $\qquad\square$

**Lemma F.2.** *Assume clean events $\mathcal{E}_p$ and $\mathcal{E}_d$ hold. Then, at the end of the execution of* ESTI-MATEPROB, *it holds that*

$$\sum_{\theta\in\Theta} \left[\mathbb{P}^{(i)}\left(\theta|b_i, s_i\right) - \mathbb{P}^{(i)}\left(\theta|b_i, s_i'\right)\right]^2 \geq 9m\varrho \quad \forall i\in\mathcal{N},\, b_i\in B_i,\, s_i\in S_i.$$

*Proof.* Fix $i\in\mathcal{N}$, $b_i\in B_i$ and $s_i\in S_i$. Notice that

$$|\xi_{b_i, s_i}^{(i)}[\theta] - \xi_{b_i, s_i'}^{(i)}[\theta]| \leq |\mathbb{P}^{(i)}\left(\theta|b_i, s_i\right) - \mathbb{P}^{(i)}\left(\theta|b_i, s_i'\right)| + 2\varrho,$$

and thus

$$||\boldsymbol{\xi}_{b_i, s_i}^{(i)} - \boldsymbol{\xi}_{b_i, s_i'}^{(i)}||_2^2 \leq \sum_{\theta\in\Theta} \left[|\mathbb{P}^{(i)}\left(\theta|b_i, s_i\right) - \mathbb{P}^{(i)}\left(\theta|b_i, s_i'\right)| + 2\varrho\right]^2 \tag{37a}$$

$$\leq 4m\varrho^2 + \sum_{\theta\in\Theta} \left[\mathbb{P}^{(i)}\left(\theta|b_i, s_i\right) - \mathbb{P}^{(i)}\left(\theta|b_i, s_i'\right)\right]^2, \tag{37b}$$

where the second inequality follows from triangle inequality. Hence, rearranging, we get

$$\sum_{\theta\in\Theta} \left[\mathbb{P}^{(i)}\left(\theta|b_i, s_i\right) - \mathbb{P}^{(i)}\left(\theta|b_i, s_i'\right)\right]^2 \geq ||\boldsymbol{\xi}_{b_i, s_i}^{(i)} - \boldsymbol{\xi}_{b_i, s_i'}^{(i)}||_2^2 - 4m\varrho^2 \geq 13m\varrho - 4m\varrho^2 \geq 9m\varrho,$$

where the third inequality follows from $\varrho \leq 1$ since $\overline{\varrho} \leq \underline{d}/13m$. This concludes the proof. $\qquad\square$

**Lemma F.3.** *For each $i\in\mathcal{N}$ and $b_i\in B_i$, let $\hat{\gamma}_i^{b_i}$ be an uncorrelated scoring rule defined according to Equation (7). Then, assuming clean events $\mathcal{E}_p$ and $\mathcal{E}_d$ are verified, the following holds*

$$\sum_{\theta\in\Theta} \mathbb{P}^{(i)}\left(\theta|s_i, b_i\right)\left[\hat{\gamma}_i^{b_i}[s_i, \theta] - \hat{\gamma}_i^{b_i}[s_i', \theta]\right] \geq \frac{\ell}{18} \quad \forall i\in\mathcal{N}, \forall b_i\in B_i, \forall s_i, s_i'\in S_i.$$

*Proof.* Fix $i \in \mathcal{N}$, $b_i \in B_i$ and $s_i, s_i' \in S_i$. Then, we can state the following:

$$\sum_{\theta \in \Theta} \mathbb{P}^{(i)}(\theta|s_i, b_i) \hat{\gamma}_i^{b_i}[s_i, \theta] = \sum_{\theta \in \Theta} \mathbb{P}^{(i)}(\theta|s_i, b_i) \left[ \xi_{b_i,s_i}^{(i)}[\theta] + H_i - \frac{1}{2}||\boldsymbol{\xi}_{b_i,s_i}^{(i)}||_2^2 \right] \tag{38a}$$

$$= H_i - \frac{1}{2}||\boldsymbol{\xi}_{b_i,s_i}^{(i)}||_2^2 + \sum_{\theta \in \Theta} \mathbb{P}^{(i)}(\theta|s_i, b_i) \xi_{b_i,s_i}^{(i)}[\theta] \tag{38b}$$

$$\geq H_i - \frac{1}{2}||\boldsymbol{\xi}_{b_i,s_i}^{(i)}||_2^2 + \sum_{\theta \in \Theta} \left( \xi_{b_i,s_i}^{(i)}[\theta] - \varrho \right) \xi_{b_i,s_i}^{(i)}[\theta] \tag{38c}$$

$$= H_i + \frac{1}{2}||\boldsymbol{\xi}_{b_i,s_i}^{(i)}||_2^2 - \varrho \tag{38d}$$

Furthermore,

$$\sum_{\theta \in \Theta} \mathbb{P}^{(i)}(\theta|s_i, b_i) \hat{\gamma}_i^{b_i}[s_i', \theta] = \sum_{\theta \in \Theta} \mathbb{P}^{(i)}(\theta|s_i, b_i) \left[ \xi_{b_i,s_i'}^{(i)}[\theta] + H_i - \frac{1}{2}||\boldsymbol{\xi}_{b_i,s_i'}^{(i)}||_2^2 \right] \tag{39a}$$

$$= H_i - \frac{1}{2}||\boldsymbol{\xi}_{b_i,s_i'}^{(i)}||_2^2 + \sum_{\theta \in \Theta} \mathbb{P}^{(i)}(\theta|s_i, b_i) \xi_{b_i,s_i'}^{(i)}[\theta] \tag{39b}$$

$$\leq H_i - \frac{1}{2}||\boldsymbol{\xi}_{b_i,s_i'}^{(i)}||_2^2 + \sum_{\theta \in \Theta} \left( \xi_{b_i,s_i}^{(i)}[\theta] + \varrho \right) \xi_{b_i,s_i'}^{(i)}[\theta] \tag{39c}$$

$$= H_i - \frac{1}{2}||\boldsymbol{\xi}_{b_i,s_i'}^{(i)}||_2^2 + \varrho + \sum_{\theta \in \Theta} \xi_{b_i,s_i}^{(i)}[\theta] \xi_{b_i,s_i'}^{(i)}[\theta]. \tag{39d}$$

Combining the two results, we get

$$\sum_{\theta \in \Theta} \mathbb{P}^{(i)}(\theta|s_i, b_i) \left[ \hat{\gamma}_i^{b_i}[s_i, \theta] - \hat{\gamma}_i^{b_i}[s_i', \theta] \right] \tag{40a}$$

$$\geq +\frac{1}{2}||\boldsymbol{\xi}_{b_i,s_i}^{(i)}||_2^2 + \frac{1}{2}||\boldsymbol{\xi}_{b_i,s_i'}^{(i)}||_2^2 - \sum_{\theta \in \Theta} \xi_{b_i,s_i}^{(i)}[\theta] \xi_{b_i,s_i'}^{(i)}[\theta] - 2\varrho \tag{40b}$$

$$= \frac{1}{2} \sum_{\theta \in \Theta} \left[ \left( \xi_{b_i,s_i}^{(i)}[\theta] \right)^2 - 2\xi_{b_i,s_i}^{(i)}[\theta] \xi_{b_i,s_i'}^{(i)}[\theta] + \left( \xi_{b_i,s_i'}^{(i)}[\theta] \right)^2 \right] - 2\varrho \tag{40c}$$

$$= \frac{1}{2} \sum_{\theta \in \Theta} \left[ \xi_{b_i,s_i}^{(i)}[\theta] - \xi_{b_i,s_i'}^{(i)}[\theta] \right]^2 - 2\varrho \tag{40d}$$

$$\geq \frac{1}{2} \sum_{\theta \in \Theta} \left[ \mathbb{P}^{(i)}(\theta|b_i, s_i) - \mathbb{P}^{(i)}(\theta|b_i, s_i') \right]^2 - 2m\varrho^2 - 2\varrho \tag{40e}$$

$$\geq \frac{\ell}{2} - 4m\varrho \tag{40f}$$

$$\geq \frac{\ell}{2} - \frac{4}{9}\ell \tag{40g}$$

$$= \frac{\ell}{18} \tag{40h}$$

where Equation (40e) follows from the fact that, since clean event $\mathcal{E}_p$ holds, $||\boldsymbol{\xi}_{b_i,s_i}^{(i)} - \boldsymbol{\xi}_{b_i,s_i'}^{(i)}||_2^2 \geq \sum_{\theta \in \Theta} \left[ \mathbb{P}^{(i)}(\theta|b_i, s_i) - \mathbb{P}^{(i)}(\theta|b_i, s_i') \right]^2 - 4m\varrho^2$, Equation (40f) follows from the definition of $\ell$ and Equation (40g) follows from Lemma F.2. This concludes the proof. $\square$

**Lemma F.4.** *Assume the clean events $\mathcal{E}_p$ and $\mathcal{E}_d$ hold. Let $\boldsymbol{\gamma}' = (\boldsymbol{\gamma}_1', ..., \boldsymbol{\gamma}_n')$ be such that*

$$\boldsymbol{\gamma}_i'[\boldsymbol{b}, \boldsymbol{s}, \theta] = \beta \gamma_i^{b_i}[s_i, \theta] + (1 - \beta)\hat{\gamma}_i^{b_i}[s_i, \theta],$$

*where $\beta = (45 + \bar{\ell})/(18\rho + 45 + \bar{\ell})$. Then, the following holds*

$$F_i(\tilde{\boldsymbol{\mu}}^t, \boldsymbol{\gamma}') - F_i^{\phi,\varphi}(\tilde{\boldsymbol{\mu}}^t, \boldsymbol{\gamma}') \geq \sum_{\boldsymbol{b} \in \mathcal{B}} \tilde{\mu}^t[\boldsymbol{b}] C_i(b_i, \phi(b_i)) + \frac{\rho\ell}{65} \quad \forall i \in \mathcal{N}, \forall(\phi, \varphi) \in \Phi_i.$$

*Proof.* Fix an agent $i \in \mathcal{N}$ and a deviation policy $(\phi, \varphi) \in \Phi_i$. With a slight abuse of notation, let $\gamma_i'[b_i, s_i, \theta] = \beta \gamma_i^{b_i}[s_i, \theta] + (1 - \beta)\hat{\gamma}_i^{b_i}[s_i, \theta]$. Then, we can write the following:

$$
\begin{aligned}
& F_i(\tilde{\boldsymbol{\mu}}^t, \boldsymbol{\gamma}') - F_i^{\phi, \varphi}(\tilde{\boldsymbol{\mu}}^t, \boldsymbol{\gamma}') \\
& = \underbrace{\sum_{\substack{\boldsymbol{b} \in \mathcal{B}: \\ \phi(b_i) \neq b_i}} \tilde{\mu}^t[\boldsymbol{b}] \sum_{\substack{s_i \in S_i \\ \theta \in \Theta}} \left[ \mathbb{P}^{(i)}\left(s_i, \theta | b_i\right) \gamma_i'[b_i, s_i, \theta] - \mathbb{P}^{(i)}\left(s_i, \theta | \phi(b_i)\right) \gamma_i'[b_i, \varphi(b_i, s_i), \theta] \right]}_{\text{\textcircled{A}}}
\end{aligned}
$$

$$
+ \underbrace{\sum_{\substack{\boldsymbol{b} \in \mathcal{B}: \\ \phi(b_i) = b_i}} \tilde{\mu}^t[\boldsymbol{b}] \left[ \sum_{\substack{s_i \in S_i \\ \theta \in \Theta}} \mathbb{P}^{(i)}\left(s_i, \theta | b_i\right) \gamma_i'[b_i, s_i, \theta] - \mathbb{P}^{(i)}\left(s_i, \theta | b_i\right) \gamma_i'[b_i, \varphi(b_i, s_i), \theta] \right]}_{\text{\textcircled{B}}}. \tag{41a}
$$

Let us analyze the two terms separately.

**Term \textcircled{A}.**   First, notice that, by Assumption 1, it holds that

$$
\begin{aligned}
& \sum_{\substack{\boldsymbol{b} \in \mathcal{B}: \\ \phi(b_i) \neq b_i}} \tilde{\mu}^t[\boldsymbol{b}] \left[ \sum_{\substack{s_i \in S_i \\ \theta \in \Theta}} \mathbb{P}^{(i)}\left(s_i, \theta | b_i\right) \gamma_i^{b_i}[s_i, \theta] - \mathbb{P}^{(i)}\left(s_i, \theta | \phi(b_i)\right) \gamma_i^{b_i}[\varphi(b_i, s_i), \theta] \right] \\
& \geq \sum_{\substack{\boldsymbol{b} \in \mathcal{B}: \\ \phi(b_i) \neq b_i}} \tilde{\mu}^t[\boldsymbol{b}] C_i(b_i, \phi(b_i)) + \rho. \tag{42a}
\end{aligned}
$$

Furthermore, noticing that $C_i(b_i, \phi(b_i)) \leq 1$ and that, by definition, $\hat{\gamma}_i^{b_i}[s_i, \theta] \leq 3/2$ for all $i \in \mathcal{N}$, $b_i \in B_i$, $s_i \in S_i$ and $\theta \in \Theta$, we can write the following

$$
\begin{aligned}
& \sum_{\substack{\boldsymbol{b} \in \mathcal{B}: \\ \phi(b_i) \neq b_i}} \tilde{\mu}^t[\boldsymbol{b}] \left[ \sum_{\substack{s_i \in S_i \\ \theta \in \Theta}} \left[ \mathbb{P}^{(i)}\left(s_i, \theta | b_i\right) \hat{\gamma}_i^{b_i}[s_i, \theta] - \mathbb{P}^{(i)}\left(s_i, \theta | \phi(b_i)\right) \hat{\gamma}_i^{b_i}[\varphi(b_i, s_i), \theta] \right] - C_i(b_i, \phi(b_i)) \right] \\
& \geq \sum_{\substack{\boldsymbol{b} \in \mathcal{B}: \\ \phi(b_i) \neq b_i}} -\frac{5}{2} \tilde{\mu}^t[\boldsymbol{b}]. \tag{43a}
\end{aligned}
$$

Rearranging, we get

$$
\begin{aligned}
& \sum_{\substack{\boldsymbol{b} \in \mathcal{B}: \\ \phi(b_i) \neq b_i}} \tilde{\mu}^t[\boldsymbol{b}] \left[ \sum_{\substack{s_i \in S_i \\ \theta \in \Theta}} \mathbb{P}^{(i)}\left(s_i, \theta | b_i\right) \hat{\gamma}_i^{b_i}[s_i, \theta] - \mathbb{P}^{(i)}\left(s_i, \theta | \phi(b_i)\right) \hat{\gamma}_i^{b_i}[\varphi(b_i, s_i), \theta] \right] \\
& \geq \sum_{\substack{\boldsymbol{b} \in \mathcal{B}: \\ \phi(b_i) \neq b_i}} \tilde{\mu}^t[\boldsymbol{b}] \left[ C_i(b_i, \phi(b_i)) - \frac{5}{2} \right], \tag{44a}
\end{aligned}
$$

By linearity and by definition of $\boldsymbol{\gamma}'$, we get

$$
\text{\textcircled{A}} \geq \sum_{\substack{\boldsymbol{b} \in \mathcal{B}: \\ \phi(b_i) \neq b_i}} \tilde{\mu}^t[\boldsymbol{b}] \left[ C_i(b_i, \phi(b_i)) + \beta \rho - \frac{5}{2}(1 - \beta) \right] \tag{45a}
$$

$$
= \sum_{\substack{\boldsymbol{b} \in \mathcal{B}: \\ \phi(b_i) \neq b_i}} \tilde{\mu}^t[\boldsymbol{b}] \left[ C_i(b_i, \phi(b_i)) + \frac{\rho \bar{\ell}}{18\rho + 45 + \bar{\ell}} \right] \tag{45b}
$$

$$
\geq \sum_{\substack{\boldsymbol{b} \in \mathcal{B}: \\ \phi(b_i) \neq b_i}} \tilde{\mu}^t[\boldsymbol{b}] \left[ C_i(b_i, \phi(b_i)) + \frac{\rho \ell}{65} \right], \tag{45c}
$$

where Equation (45b) follows from the definition of $\beta$ and Equation (45c) follows from $\rho < 1$ (w.l.o.g.), $\bar{\ell} \leq 2$ (by the stopping condition of Algorithm 2) and $\ell \leq \bar{\ell}$ (since clean event $\mathcal{E}_p$ holds).

**Term Ⓑ.** By assumption 1, it holds that

$$
\sum_{\substack{\boldsymbol{b}\in\mathcal{B}:\\ \phi(b_i)=b_i}} \tilde{\mu}^t[\boldsymbol{b}] \left[ \sum_{\substack{s_i\in S_i\\ \theta\in\Theta}} \mathbb{P}^{(i)}\left(s_i,\theta|b_i\right)\gamma_i^{b_i}[s_i,\theta] - \mathbb{P}^{(i)}\left(s_i,\theta|b_i\right)\gamma_i^{b_i}[\varphi(b_i,s_i),\theta] \right]
$$

$$
= \sum_{\substack{\boldsymbol{b}\in\mathcal{B}:\\ \phi(b_i)=b_i}} \tilde{\mu}^t[\boldsymbol{b}] \sum_{s_i\in S_i} \mathbb{P}^{(i)}\left(s_i|b_i\right) \sum_{\theta\in\Theta} \left[ \mathbb{P}^{(i)}\left(\theta|b_i,s_i\right)\left[\gamma_i^{b_i}[s_i,\theta] - \gamma_i^{b_i}[\varphi(b_i,s_i),\theta]\right] \right] \tag{46a}
$$

$$
\geq \sum_{\substack{\boldsymbol{b}\in\mathcal{B}:\\ \phi(b_i)=b_i}} \tilde{\mu}^t[\boldsymbol{b}] \sum_{s_i\in S_i} \mathbb{P}^{(i)}\left(s_i|b_i\right) \sum_{\theta\in\Theta} \left[ \mathbb{P}^{(i)}\left(\theta|b_i,s_i\right)\left[\gamma_i^{b_i}[s_i,\theta] - \gamma_i^{b_i}[s_i,\theta]\right] \right] \tag{46b}
$$

$$
= 0. \tag{46c}
$$

Furthermore, by Lemma F.3,

$$
\sum_{\substack{\boldsymbol{b}\in\mathcal{B}:\\ \phi(b_i)=b_i}} \tilde{\mu}^t[\boldsymbol{b}] \left[ \sum_{\substack{s_i\in S_i\\ \theta\in\Theta}} \mathbb{P}^{(i)}\left(s_i,\theta|b_i\right)\hat{\gamma}_i^{b_i}[s_i,\theta] - \mathbb{P}^{(i)}\left(s_i,\theta|b_i\right)\hat{\gamma}_i^{b_i}[\varphi(b_i,s_i),\theta] \right]
$$

$$
= \sum_{\substack{\boldsymbol{b}\in\mathcal{B}:\\ \phi(b_i)=b_i}} \tilde{\mu}^t[\boldsymbol{b}] \sum_{s_i\in S_i} \mathbb{P}^{(i)}\left(s_i|b_i\right) \sum_{\theta\in\Theta} \left[ \mathbb{P}^{(i)}\left(\theta|b_i,s_i\right)\left[\hat{\gamma}_i^{b_i}s_i,\theta] - \hat{\gamma}_i^{b_i}[\varphi(b_i,s_i),\theta]\right] \right] \tag{47a}
$$

$$
\geq \sum_{\substack{\boldsymbol{b}\in\mathcal{B}:\\ \phi(b_i)=b_i}} \tilde{\mu}^t[\boldsymbol{b}] \sum_{s_i\in S_i} \mathbb{P}^{(i)}\left(s_i|b_i\right) \frac{\ell}{18} \tag{47b}
$$

$$
= \sum_{\substack{\boldsymbol{b}\in\mathcal{B}:\\ \phi(b_i)=b_i}} \tilde{\mu}^t[\boldsymbol{b}] \frac{\ell}{18} \tag{47c}
$$

$$
\tag{47d}
$$

Also in this case, by linearity and by definition of $\boldsymbol{\gamma}'$, we get

$$
Ⓑ \geq \sum_{\substack{\boldsymbol{b}\in\mathcal{B}:\\ \phi(b_i)=b_i}} \tilde{\mu}^t[\boldsymbol{b}] \left( C_i(b_i,\phi(b_i)) + (1-\beta)\frac{\ell}{18} \right) \tag{48a}
$$

$$
= \sum_{\substack{\boldsymbol{b}\in\mathcal{B}:\\ \phi(b_i)=b_i}} \tilde{\mu}^t[\boldsymbol{b}] \left[ C_i(b_i,\phi(b_i)) + \frac{\rho\ell}{18\rho+45+\bar{\ell}} \right], \tag{48b}
$$

$$
\geq \sum_{\substack{\boldsymbol{b}\in\mathcal{B}:\\ \phi(b_i)=b_i}} \tilde{\mu}^t[\boldsymbol{b}] \left[ C_i(b_i,\phi(b_i)) + \frac{\rho\ell}{65} \right] \tag{48c}
$$

where to obtain Equation (48b) we substituted the definition of $\beta$ and Equation (48c) follows from $\rho < 1$ (w.l.o.g.) and $\bar{\ell} \leq 2$ (by the stopping condition of Algorithm 2).

**Putting all together.** By substituting Equations (45c) and (48c) into Equation 41a, we get

$$
F_i(\tilde{\boldsymbol{\mu}}^t,\boldsymbol{\gamma}') - F_i^{\phi,\varphi}(\tilde{\boldsymbol{\mu}}^t,\boldsymbol{\gamma}') \geq \sum_{\boldsymbol{b}\in\mathcal{B}} \tilde{\mu}^t[\boldsymbol{b}]C_i(b_i,\phi(b_i)) + \frac{\rho\ell}{65}.
$$

This concludes the proof. $\qquad\square$

**Lemma 8.2.** *Assume clean events $\mathcal{E}_c$, $\mathcal{E}_p$ hold. If mechanism $(\boldsymbol{\mu}^t,\boldsymbol{\gamma}^t,\boldsymbol{\pi}^t)$ is chosen according to Algorithm 3, then it is IC, i.e., if satisfies Equation (1b) of optimization problem (1).*

*Proof.* Fix an agent $i \in \mathcal{N}$ and a deviation policy $\phi, \varphi \in \Phi_i$. By linearity, we have that

$$F_i(\tilde{\boldsymbol{\mu}}^t, \boldsymbol{\gamma}^t) - F_i^{\phi,\varphi}(\tilde{\boldsymbol{\mu}}^t, \boldsymbol{\gamma}^t) = \alpha \left[ F_i(\tilde{\boldsymbol{\mu}}^t, \tilde{\boldsymbol{\gamma}}^t) - F_i^{\phi,\varphi}(\tilde{\boldsymbol{\mu}}^t, \tilde{\boldsymbol{\gamma}}^t) \right] + (1 - \alpha) \left[ F_i(\tilde{\boldsymbol{\mu}}^t, \boldsymbol{\gamma}') - F_i^{\phi,\varphi}(\tilde{\boldsymbol{\mu}}^t, \boldsymbol{\gamma}') \right] \tag{49a}$$

$$\geq \sum_{\boldsymbol{b} \in \mathcal{B}} \tilde{\mu}^t[\boldsymbol{b}] C_i(b_i, \phi(b_i)) - \alpha\lambda + (1 - \alpha)\frac{\rho\ell}{65} \tag{49b}$$

$$= \sum_{\boldsymbol{b} \in \mathcal{B}} \tilde{\mu}^t[\boldsymbol{b}] C_i(b_i, \phi(b_i)) + \frac{\lambda\rho\ell - \lambda\rho\underline{\ell} + 8m\varrho\rho\ell/65}{\rho\bar{\ell} + 65\lambda} \tag{49c}$$

$$\geq \sum_{\boldsymbol{b} \in \mathcal{B}} \tilde{\mu}^t[\boldsymbol{b}] C_i(b_i, \phi(b_i)), \tag{49d}$$

where Equation (49b) follows from Lemmas 8.1 and F.4, Equation (49c) follows from the definition of $\alpha$ in Algorithm 3 and Equation (49d) follows from the fact that, since clean event $\mathcal{E}_p$ holds, $\ell \geq \underline{\ell}$. This concludes the proof. $\square$

## G PROOF OF THEOREM 5.1

Assume clean events $\mathcal{E}_c$ and $\mathcal{E}_p$ hold. Notice that by a union bound on the results of Lemma 6.1 and Lemma 7.1, this happens with probability at least $1 - \delta$. From the definition of regret we have that

$$R^T = \underbrace{\sum_{t \in T_c} \left[ U(\boldsymbol{\mu}^\star, \boldsymbol{\gamma}^\star, \boldsymbol{\pi}^\star) - U(\boldsymbol{\mu}^t, \boldsymbol{\gamma}^t, \boldsymbol{\pi}^t) \right]}_{R_c^T} + \underbrace{\sum_{t \in T_u} \left[ U(\boldsymbol{\mu}^\star, \boldsymbol{\gamma}^\star, \boldsymbol{\pi}^\star) - U^\circ(\boldsymbol{\gamma}^t, \boldsymbol{\pi}^t) \right]}_{R_u^T}.$$

Let us analyze the two terms separately.

**Term $R_u^T$.** Consider the rounds $T_u$ in which the principal committed to an uncorrelated mechanism. Since during the commit phase the principal uses only correlated mechanisms, we have that

$$T_u = \sum_{\boldsymbol{b} \in \mathcal{B}} \mathcal{T}_p(\boldsymbol{b}) + \mathcal{T}_d.$$

By Lemma 6.2 and Lemma 7.2, we have that

$$|T_u| \leq |\mathcal{B}| \max\{N_1, \kappa\} + 2nk^3 N_2 + nk^2 N_3 = |\mathcal{B}| \max\{T^{2/3}, \kappa\} + 2nk^3 \log(T) + nk^2 T^{2/3},$$

where $\kappa = \frac{289}{2} m^2 \ln(2K/\delta)\frac{1}{i^2\ell^2}$. Then, since for all $t$, by definition of $U$ we have that $U(\boldsymbol{\mu}^\star, \boldsymbol{\gamma}^\star, \boldsymbol{\pi}^\star) - U^\circ(\boldsymbol{\gamma}^t, \boldsymbol{\pi}^t) \leq nM + 1$, we can bound the term $R_u^T$ as

$$R_u^T \leq (nM + 1) \left( |\mathcal{B}| \max\{T^{2/3}, \kappa\} + nk^3 l^2 \log(T) + nk^3 l^2 T^{2/3} \right).$$

**Term $R_c^T$.** The only rounds in which the principal commits to correlated mechanisms are those of the commit phase. Let $t \in T_c$. By definition of $\boldsymbol{\gamma}^t$ and by linearity of $U$, we have that

$$R_c^T \leq \alpha \sum_{t \in T_c} \left[ U(\boldsymbol{\mu}^\star, \boldsymbol{\gamma}^\star, \boldsymbol{\pi}^\star) - U(\boldsymbol{\mu}^t, \tilde{\boldsymbol{\gamma}}^t, \boldsymbol{\pi}^t) \right] + (1 - \alpha)T(nM + 1). \tag{50}$$

Furthermore, notice that we can bound $1 - \alpha$ as

$$(1 - \alpha) = \frac{65\lambda}{\rho\bar{\ell} + 65\lambda} \leq \frac{65}{\rho\ell} 2M|\mathcal{S}|m(k + 1)(\nu + \chi).$$

Recalling that, by Lemma B.8

$$\chi \leq 2kl^2 M \sqrt{\frac{\ln(2nk^2/\delta)}{2N_3}} + \frac{kl^2 M}{2^{N_2}} = 2kl^2 M \sqrt{\frac{\ln(2nk^2/\delta)}{2T^{2/3}}} + \frac{kl^2 M}{T},$$

and by definition

$$\nu = \max_{\boldsymbol{b} \in \mathcal{B}} \left\{ \sqrt{\frac{\ln(12|\mathcal{B}|T|\mathcal{S}|nm/\delta)}{2|\mathcal{T}_p(\boldsymbol{b})|}} \right\} \leq \sqrt{\frac{\ln(12|\mathcal{B}|T|\mathcal{S}|nm/\delta)}{2T^{2/3}}},$$

we get

$$(1 - \alpha) \leq \frac{780}{\rho\ell} M^2 |\mathcal{S}| mk^2 l^2 \left( \sqrt{\frac{\ln(12|\mathcal{B}|T|\mathcal{S}|nm/\delta)}{2T^{2/3}}} + \frac{1}{T} \right). \tag{51}$$

Let us now consider the term $U(\boldsymbol{\mu}^\star, \boldsymbol{\gamma}^\star, \boldsymbol{\pi}^\star)$. Let $(\boldsymbol{x}^\star, \boldsymbol{y}^\star, \boldsymbol{\mu}^\star, \boldsymbol{z}^\star)$ be an optimal solution to LP$(\mathbb{P}, C, 0)$ and let $(\boldsymbol{x}^t, \boldsymbol{y}^t, \boldsymbol{\mu}^t, \boldsymbol{z}^t)$ be the optimal solution to LP$(\boldsymbol{\zeta}, \Lambda, \varepsilon)$ from which mechanism $(\boldsymbol{\mu}^t, \tilde{\boldsymbol{\gamma}}^t, \boldsymbol{\pi}^t)$ was obtained. Furthermore, we recall that the objective function of LP$(\boldsymbol{\zeta}, \Lambda_i, \varepsilon)$ is

$$\bar{U}(\boldsymbol{x}, \boldsymbol{y}, \boldsymbol{\mu}, \boldsymbol{z}) = \sum_{\boldsymbol{b} \in \mathcal{B}} \sum_{\boldsymbol{s} \in \mathcal{S}} \sum_{\theta \in \Theta} \left[ \zeta_{\boldsymbol{b}}[\boldsymbol{s}, \theta] \left[ \sum_{a \in \mathcal{A}} y[\boldsymbol{b}, \boldsymbol{s}, a] u(a, \theta) \right] - \sum_{i \in \mathcal{N}} x_i[\boldsymbol{b}, \boldsymbol{s}, \theta] \right].$$

Then, we can write the following

$$
\begin{aligned}
U(\boldsymbol{\mu}^\star, \boldsymbol{\gamma}^\star, \boldsymbol{\pi}^\star) &= \sum_{\boldsymbol{b} \in \mathcal{B}} \sum_{\boldsymbol{s} \in \mathcal{S}} \sum_{\theta \in \Theta} \left[ \mu^\star[\boldsymbol{b}] \mathbb{P}(\boldsymbol{s}, \theta | \boldsymbol{b}) \left[ \sum_{a \in \mathcal{A}} \pi^\star[\boldsymbol{b}, \boldsymbol{s}, a] u(a, \theta) \right] - \sum_{i \in \mathcal{N}} \gamma_i^\star[\boldsymbol{b}, \boldsymbol{s}, \theta] \right] \\
&\leq \sum_{\boldsymbol{b} \in \mathcal{B}} \sum_{\boldsymbol{s} \in \mathcal{S}} \sum_{\theta \in \Theta} \left[ \mu^\star[\boldsymbol{b}] \zeta_{\boldsymbol{b}}[\boldsymbol{s}, \theta] \left[ \sum_{a \in \mathcal{A}} \pi^\star[\boldsymbol{b}, \boldsymbol{s}, a] u(a, \theta) \right] - \sum_{i \in \mathcal{N}} \gamma_i^\star[\boldsymbol{b}, \boldsymbol{s}, \theta] \right] + 2M|\mathcal{S}|mn\nu \\
&= \sum_{\boldsymbol{b} \in \mathcal{B}} \sum_{\boldsymbol{s} \in \mathcal{S}} \sum_{\theta \in \Theta} \left[ \zeta_{\boldsymbol{b}}[\boldsymbol{s}, \theta] \left[ \sum_{a \in \mathcal{A}} y^\star[\boldsymbol{b}, \boldsymbol{s}, a] u(a, \theta) \right] - \sum_{i \in \mathcal{N}} x_i^\star[\boldsymbol{b}, \boldsymbol{s}, \theta] \right] + 2M|\mathcal{S}|mn\nu \\
&= \bar{U}(\boldsymbol{x}^\star, \boldsymbol{y}^\star, \boldsymbol{\mu}^\star, \boldsymbol{\pi}^\star) + 2M|\mathcal{S}|mn\nu,
\end{aligned}
$$

where the first inequality follows since, given that clean event holds, $|\mathbb{P}(\boldsymbol{s}, \theta | \boldsymbol{b}) - \zeta_{\boldsymbol{b}}[\boldsymbol{s}, \theta]| \leq \nu$. In a similar way, it holds that

$$
\begin{aligned}
U(\boldsymbol{\mu}^t, \tilde{\boldsymbol{\gamma}}^t, \boldsymbol{\pi}^t) &= \sum_{\boldsymbol{b} \in \mathcal{B}} \sum_{\boldsymbol{s} \in \mathcal{S}} \sum_{\theta \in \Theta} \left[ \mu^t[\boldsymbol{b}] \mathbb{P}(\boldsymbol{s}, \theta | \boldsymbol{b}) \left[ \sum_{a \in \mathcal{A}} \pi^t[\boldsymbol{b}, \boldsymbol{s}, a] u(a, \theta) \right] - \sum_{i \in \mathcal{N}} \tilde{\gamma}_i^t[\boldsymbol{b}, \boldsymbol{s}, \theta] \right] \\
&\geq \sum_{\boldsymbol{b} \in \mathcal{B}} \sum_{\boldsymbol{s} \in \mathcal{S}} \sum_{\theta \in \Theta} \left[ \mu^t[\boldsymbol{b}] \zeta_{\boldsymbol{b}}[\boldsymbol{s}, \theta] \left[ \sum_{a \in \mathcal{A}} \pi^t[\boldsymbol{b}, \boldsymbol{s}, a] u(a, \theta) \right] - \sum_{i \in \mathcal{N}} \tilde{\gamma}_i^t[\boldsymbol{b}, \boldsymbol{s}, \theta] \right] - 2M|\mathcal{S}|mn\nu \\
&= \sum_{\boldsymbol{b} \in \mathcal{B}} \sum_{\boldsymbol{s} \in \mathcal{S}} \sum_{\theta \in \Theta} \left[ \zeta_{\boldsymbol{b}}[\boldsymbol{s}, \theta] \left[ \sum_{a \in \mathcal{A}} y^t[\boldsymbol{b}, \boldsymbol{s}, a] u(a, \theta) \right] - \sum_{i \in \mathcal{N}} x_i^t[\boldsymbol{b}, \boldsymbol{s}, \theta] \right] - 2M|\mathcal{S}|mn\nu \\
&= \bar{U}(\boldsymbol{x}^t, \boldsymbol{y}^t, \boldsymbol{\mu}^t, \boldsymbol{\pi}^\star) - 2M|\mathcal{S}|mn\nu,
\end{aligned}
$$

Furthermore, we recall that, by Lemma F.1, $(\boldsymbol{x}^\star, \boldsymbol{y}^\star, \boldsymbol{\mu}^\star, \boldsymbol{z}^\star)$ is a feasible solution to LP$(\boldsymbol{\zeta}, \Lambda, \varepsilon)$, while, by definition, $(\boldsymbol{x}^t, \boldsymbol{y}^t, \boldsymbol{\mu}^t, \boldsymbol{z}^t)$ is optimal for LP$(\boldsymbol{\zeta}, \Lambda, \varepsilon)$. Thus, $\bar{U}(\boldsymbol{x}^t, \boldsymbol{y}^t, \boldsymbol{\mu}^t, \boldsymbol{\pi}^\star) \geq \bar{U}(\boldsymbol{x}^\star, \boldsymbol{y}^\star, \boldsymbol{\mu}^\star, \boldsymbol{\pi}^\star)$, which yields

$$
\begin{aligned}
\sum_{t \in T_c} \left[ U(\boldsymbol{\mu}^\star, \boldsymbol{\gamma}^\star, \boldsymbol{\pi}^\star) - U(\boldsymbol{\mu}^t, \tilde{\boldsymbol{\gamma}}^t, \boldsymbol{\pi}^t) \right] &\leq \sum_{t \in T_c} \left[ \bar{U}(\boldsymbol{x}^\star, \boldsymbol{y}^\star, \boldsymbol{\mu}^\star, \boldsymbol{\pi}^\star) - \bar{U}(\boldsymbol{x}^t, \boldsymbol{y}^t, \boldsymbol{\mu}^t, \boldsymbol{\pi}^\star) + 4M|\mathcal{S}|mn\nu \right] \\
&\leq \sum_{t \in T_c} 4M|\mathcal{S}|mn\nu \\
&\leq 4M|\mathcal{S}|mnT^{2/3}\sqrt{\ln(12|\mathcal{B}|T|\mathcal{S}|nm/\delta)}
\end{aligned}
$$

Combining the above results, we get

$$
\begin{aligned}
R_c^T &\leq \alpha \sum_{t \in T_c} \left[ U(\boldsymbol{\mu}^\star, \boldsymbol{\gamma}^\star, \boldsymbol{\pi}^\star) - U(\boldsymbol{\mu}^t, \tilde{\boldsymbol{\gamma}}^t, \boldsymbol{\pi}^t) \right] + (1 - \alpha)T(nM + 1) \\
&\leq \frac{1564}{\rho\ell} M^3 |\mathcal{S}| mnk^2 \left( T^{2/3}\sqrt{\ln(12|\mathcal{B}|T|\mathcal{S}|nm/\delta)} + 1 \right).
\end{aligned}
$$

**Putting all together.** Hence, the final regret bound is

$$R^T = R_c^T + R_u^T$$
$$\leq \frac{1567}{\rho\ell} M^3 |\mathcal{B}||\mathcal{S}|mnk^3l^2 \left(\sqrt{\ln(12|\mathcal{B}|T|\mathcal{S}|nm/\delta)} + 1\right) \max\{T^{2/3}, \kappa\} + (nM + 1)\log(T),$$

which gives the result.

