# OpenReview forum: "Online Information Acquisition: Hiring Multiple Agents"
_ICLR.cc/2024/Conference — ICLR 2024 poster_

### Official Review · Reviewer_CwwP · 2023-10-30

**Soundness:** 3 good
**Presentation:** 3 good
**Contribution:** 2 fair
**Rating:** 6
**Confidence:** 2

**Summary:**

The paper studies an online information acquisition problem. In the model, a principal interacts with a group of agents. The principal uses a mechanism to recommend actions for the agents to perform and to decide payments for the agents. The mechanism also elicits information from the agents about their observations of the state of nature. Besides the interactions, the principal also takes an action after the agents perform actions and report their observations. The paper first presents an algorithm based on linear programming to compute the optimal mechanism in the full information setting. It then studies a learning problem where the transition probabilities are unknown. A no-regret algorithm that gurantees sublinear regret is provided for the learning setting.

**Strengths:**

The problem studied is well-motivated. The paper presented a number of results and is mostly clear. The analysis looks solid and technically sound.

**Weaknesses:**

The algorithms presented look very standard even though the model is a more complicated one. The LP algorithm, for example, is based on the same formulation for standard principal-agent mechanism design problem -- by maximizing the principal's utility under the agents' truthful behavior and using IC constraints to enforce this behavior. While I appreciate the effort it takes to set up the constraints for this more complicated model, the insights the approach yields are somewhat limited. The same can be said about the learning algorithms.

Besides that, some specifications of the uncorrelated mechanisms are not well justified (see Questions). The model itself does not look scalable with respect to the number of agents because of the exponential growth of joint action profiles.

**Questions:**

Why uncorrelated mechanisms are not dependent on the agents' actions as are the correlated mechanisms? I think the mechanism is still uncorrelated if \gamma_i also depends on the action of agent i.

Similarly, why not keep the principal's action policy \pi dependent on the agents' actions? The principal's action does not seem to have any influence on the agents' payoffs, so keeping it dependent on the agents' actions will not introduce any actual externalities among the agents.

Some typos:

- In Equation (2b), in the term z_i[b_i, b_i', s], the "s" should be "s_i"?

- On page 5, there is a displayed equation that defines \mathcal{U}. In the part \gamma_i: S_i \times \Theta \in [0, M], the symbol "\in" should be "\to"?

---

> ### Author Response · Authors · 2023-11-14
> **Authors' response**
>
> We thank the reviewer for pointing out typos and for the questions. In the following, we address the comments made by the reviewer.
>
> **Re:** *"Why uncorrelated mechanisms are not dependent on the agents' actions as are the correlated mechanisms? I think the mechanism is still uncorrelated if $\gamma_i$ also depends on the action of agent $i$. [...] Similarly, why not keep the principal's action policy \pi dependent on the agents' actions? The principal's action does not seem to have any influence on the agents' payoffs, so keeping it dependent on the agents' actions will not introduce any actual externalities among the agents."*
>
> In order to keep the definition of uncorrelated mechanisms more clean and simple, we opted to specify such mechanisms as independent on the action of agent $i$. In particular, notice that each uncorrelated mechanism $(\boldsymbol{\gamma}, \boldsymbol{\pi})\in\mathcal U$ can be specified as an equivalent correlated mechanism $(\boldsymbol{\mu}^\prime, \boldsymbol{\gamma}^\prime, \boldsymbol{\pi}^\prime)\in\mathcal C$ in which the recommedation policy recommends deterministically the best-response (*i.e.,* such that $\mu^\prime[\boldsymbol{b}^\circ(\boldsymbol{\gamma)}] = 1$) and in which the payment function and action selection policy are specified such that
> $$
>     \gamma_i^\prime[\boldsymbol{b}, \boldsymbol{s}, \theta] = \begin{cases} \gamma_i[s_i, \theta] & \text{if } \boldsymbol{b} = \boldsymbol{b}^\circ(\boldsymbol{\gamma})  \\\\
>     0 & \text{otherwise} \end{cases} \quad\quad\forall i\in\mathcal N,\,\forall\boldsymbol{b}\in\mathcal B,\,\forall \boldsymbol{s}\in\mathcal{S},\,\forall\theta\in\Theta,
>  $$
>  and
>  $$
>     \pi^\prime[\boldsymbol{b}, \boldsymbol{s}, a] = \\begin{cases} \pi_i[\boldsymbol{s}, a] & \text{if } \boldsymbol{b} = \boldsymbol{b}^\circ(\boldsymbol{\gamma})  \\\\
>     \frac{1}{|\mathcal A|} & \text{otherwise} \\end{cases} \quad\quad\forall\boldsymbol{b}\in\mathcal B,\,\forall \boldsymbol{s}\in\mathcal{S},\,\forall a \in\mathcal A.
> $$
> Given these considerations, to lighten the notation, we opted to remove from the definition of the uncorrelated mechanism the deterministic recommendation policy $\boldsymbol{\mu}^\prime$ as well as the dependency of $\boldsymbol{\gamma}$ and $\boldsymbol{\pi}$ from the recommended action.

---

> > ### Comment · Reviewer_CwwP · 2023-11-20
> > **response to author feedback**
> >
> > Thank you for your clarification. I remain positive about your work.

---

### Official Review · Reviewer_cMnb · 2023-10-31

**Soundness:** 4 excellent
**Presentation:** 3 good
**Contribution:** 4 excellent
**Rating:** 8
**Confidence:** 4

**Summary:**

The authors study the problem of online acquiring information of the unobserved world state ($\theta$) from a group of strategic agents under the principal-agent framework.
The challenges of the online problems are in three folds:

1. The agents are strategic, meaning they could deviate from the principal's action recommendation and report fraud signals. Thus, the exploration phase (with uncorrelated scoring rule) is needed to ensure IC.
2. The cost differences $C_i(b_i, b_i')$ are unobservable by the principal, which prohibits standard RL methods (via estimating the agents' cost directly) and essentiates the binary searching method.
3. In the multiple-agent setup, the number of constraints can be growing dramatically without any reduction.

This paper formulates the multi-agent information acquisition problem, reduces the problem to linear programming that can be solved in polynomial time, and provide online learning guarantee.
In particular, the authors show there is a clear separation between the uncorrelated and the correlated mechanism.

**Strengths:**

For originality, the authors formulation of the information elicitation problem under the multi-agent setting. I appreciate the discussions of the computation issue and the separation between the correlated/uncorrelated scoring mechanism. Overall, the paper is well written, but a little bit redundant in terms of the notations. The paper achieves state-of-art learning guarantee for the multiple-agent setting. The algorithm design and the analysis look sound to me.

**Weaknesses:**

1. The authors should make it clear if agents can communicate with each other their signals/actions or not, as this can cause a huge difference. I understood that the agents cannot communicate with each other by forms of the deviation functions. Please correct me if I'm wrong.
2. There are some typos that could cause confusions, e.g., it should be $\sum_{s'\in\mathcal{S}:s_i'=s_i\mathbb{P}(s' \| b, \theta)}$ at the bottom of Page 2.
3. The authors may need to justify Assumption 1, perhaps by providing examples where the set of scoring rules known by the principal in advance can be effectively learned, e.g., by random searching, or constructed.

**Questions:**

1. Could the author provide a detailed comparison between the information acquisition framework and the Bayesian Correlated Equilibrium (BCE) (Bergemann, D. 2016)? It seems to me that if the agents are not allowed to communicate with each other, these IC concepts are closely related and similar challenges occur for the learning phase.
2. If the costs of the follower are directly observable, can the learning rate be improved?
3. Is the independency assumption $\mathbb{P}^{(i)}(s_i| b, \theta) = \mathbb{P}^{(i)}(s_i| b_i, \theta)$ necessary? If so, without the independency assumption, what could be added to the difficulty in terms of computation and statistical learning?

---

> ### Author Response · Authors · 2023-11-14
> **Authors' response**
>
> We thank the reviewer for the positive comments on our paper. In the following, we respond to specific questions raised by the reviewer
>
> **Re:** *"If the costs of the follower are directly observable, can the learning rate be improved?"*
>
> If the cost functions were observable by the principal, then we could avoid to execute the cost estimation phase of our algorithm. Nonetheless, this would not improve the regret bound in its dependency from $T$ as we would still need to run the estimation phase for the probability distributions.
>
>
> **Re:** *"Is the independency assumption $\mathbb{P}^{(i)}(s_i\vert \boldsymbol{b},\theta) = \mathbb{P}^{(i)}(s_i\vert b_i, \theta)$ necessary? If so, without the independency assumption, what could be added to the difficulty in terms of computation and statistical learning?"*
>
> The assumption is needed to guarantee the existence of uncorrelated mechanisms. Indeed, if the assumption does not hold, the posterior of an agent would depend on the actions of the others introducing externalities. This rules out the existence of uncorrelated mechanism.
> Hence, without this assumption, we would be forced to work with correlated mechanisms. In this case, at the beginning of the learning phase, it would be impossible for the principal to output non-trivial mechanisms which are IC and use the feedback received to learn the game parameters. Thus, we would be forced to use non-IC mechanisms in the intial phase. However, as discussed in Section 4, it is intractable (and also not clear how) to characterize equilibria unless the correlated mechanism is IC.
> To summarize, we believe that the problem without this assumption does not only lead to linear regret bounds but it is not even well-defined.
>
>
>
> **Re:** *"Could the author provide a detailed comparison between the information acquisition framework and the Bayesian Correlated Equilibrium (BCE) (Bergemann, D. 2016)? It seems to me that if the agents are not allowed to communicate with each other, these IC concepts are closely related and similar challenges occur for the learning phase."*
>
>
> According to the setting described in (Bergemann, 2016) a shared mediator sends to the players some signals which are sampled according to a joint probability distribution that depends on some state of nature observed by the mediator. Then, based on the signal received, the agents can decide which action to take.
> Our setting is different, since, while the signals received by the agents still depend on the state of nature, they are also dependent on some preliminary action $b$ taken by them. Furthermore, in our model, the principal assumes the role of changing the utility functions of the agents by means of payments, which fundamentally differs from the role of the mediator defined in (Bergemann, 2016). Indeed, the signals sent by the 'mediator' (as meant in (Bergemann, 2016)) are parameters of the game and cannot be changed by the principal.
> We thank the reviewer for the suggestion and we will include a comparison with this work in the revised version of the paper.

---

### Official Review · Reviewer_uose · 2023-11-01

**Soundness:** 3 good
**Presentation:** 2 fair
**Contribution:** 2 fair
**Rating:** 5
**Confidence:** 3

**Summary:**

The research paper delves into the dynamics of online information acquisition among multiple agents, providing a comprehensive analysis of how to design the mechanism that influences individuals' decisions. Besides, this paper designs a polynomial-time algorithm to find an optimal incentive compatible mechanism.

**Strengths:**

1. Algorithmic Design and Optimization in Multi-Agent Settings: This paper works on designing an efficient algorithm for the multi-agent information acquisition problem, addressing both the optimization and online learning dimensions of interactions between a principal and unknown agents. The proposed algorithm, which navigates through a quadratic optimization problem via linear relaxation, culminates in a polynomial-time solution to the original problem.

2. Addressing Uncertainty in Online Learning: The transition to online learning scenarios, characterized by the principal’s lack of knowledge regarding game parameters, is handled with a robust algorithmic approach, achieving a ($\tilde{O}(T^{2/3})$) regret. This aligns with state-of-the-art benchmarks in single-agent settings.

3. Ensuring Truthfulness and Optimality: They first discussed the relationship between the optimal design and the correlated and uncorrelated mechanism. They also introduce the novel definition of regret as the difference between the optimal (correlated + IC) and suboptimal (uncorrelated + IC, correlated + NonIC). The final phase of the algorithm, committed to achieving an approximately optimal strategy while upholding truthfulness under uncertainty. The authors leverage estimations from previous phases to find an approximately optimal and incentive-compatible mechanism, subsequently combining it with a strictly incentive-compatible scoring rule. This approach demonstrates a sophisticated understanding of the trade-offs and complexities involved in designing mechanisms that balance optimality and incentive compatibility.

**Weaknesses:**

1. This paper would benefit significantly from the inclusion of empirical demonstrations to substantiate the theoretical assertions made therein.

2. In terms of sample size efficiency, the paper presents an opportunity for enhancement through the integration of more sample-efficient online learning algorithms, such as Upper Confidence Bound (UCB) or Thompson Sampling. These methodologies hold potential for yielding a more favorable regret profile.

3. The articulation throughout the paper necessitates refinement. This is particularly pertinent in relation to the elucidation of the implications associated with the various theorems and lemmas presented, which requires additional clarity and precision.

**Questions:**

1. Is there any real examples of the optimal mechanism that are uncorrelated?

2. Is there any simulations, real data to demonstrate the effective of this ETC algorithm?

3. Typo: $\alpha ==$ to $\alpha=$.

---

> ### Author Response · Authors · 2023-11-14
> **Authors' response**
>
> We thank the reviewer for the comments and questions and for reporting typos which will be corrected in the final version of the paper. In the following we address each question specifically.
>
> **Re:** *"Is there any real examples of the optimal mechanism that are uncorrelated?"*
>
> Consider a particular instance of the case that we reported as a motivating example, in which a portfolio manager wants to learn the potential of a company to make an informed investment. The manager could hire multiple analysts to conduct separate researches on the same company, where each analyst spends effort to produce a report. If the information received by each analyst is independent from the information received by the others given their effort levels and the state of the company (this is true for instance when each analyst simply does **independent** researches without interacting with the other analysts), then an optimal mechanism for the portfolio manager is uncorrelated. This follows from Theorem 4.2. In general, the optimal mechanism is uncorrelated when the signal received by each agent is independent from the one received by the other agents.
>
> **Re:** *"Is there any simulations, real data to demonstrate the effective of this ETC algorithm?"*
>
> The great majority of recent works on the topic is more concerned with theoretical contributions rather than experimental evaluations (see *e.g.*, Cacciamani et al., 2023, Chen et al., 2023). Moreover, we are the first to propose an algorithm for this kind of multi-agent information acquisition setting, and we would have no previous algorithms to compare ours with. This is aligned with related works on the topic.
>
>
>
>
> **Re:** *"In terms of sample size efficiency, the paper presents an opportunity for enhancement through the integration of more sample-efficient online learning algorithms, such as Upper Confidence Bound (UCB) or Thompson Sampling. These methodologies hold potential for yielding a more favorable regret profile."*
>
> Unfortunately, standard methods like UCB and Thompson Sampling are not applicable to our setting. Indeed, if you look at our problem as a general continuous learning problem: i) the reward are not Lipschitz, and ii) the constraints on the feasibility set are unknown. Dealing with such a complex problem requires more complex techniques than UCB and TS. Nonetheless, we point out that one of the components of our algorithm (specifically the definition of the objective function during the commit phase) exploits upper confidence bounds (similar to UCB).

---

### Official Review · Reviewer_eEDk · 2023-11-01

**Soundness:** 3 good
**Presentation:** 2 fair
**Contribution:** 3 good
**Rating:** 8
**Confidence:** 3

**Summary:**

The paper is concerned with mechanism design where there is a principle who wants to know some state theta to take an action that maximizes utility. To estimate theta the principle uses reported signals from agents. The paper begins with the correlated mechanism setting and shows how to solve the problem using LP methods with some modifications. The paper then discusses uncorrelated mechanism showing that they are sub-optimal in general settings and optimal in some restricted settings. Finally, the online setting is considered, and the paper proposes an algorithm that follows the classical explore then commit paradigm in bandits.

**Strengths:**

-The problem seems well-motivated and the model captures a wide set of applications.

-I think the paper has interesting results such as characterization of optimality and suboptimality of uncorrelated mechanisms in section 4.

-I did not check the proofs carefully. But the technical details in the paper seem interesting.

**Weaknesses:**

A-The presentation of the paper can be improved. There seem to be some missing text, see the following:

          1-what is the auxiliary variables z_i in eq (2c) equal to? Further, Theorem 3.1 has a collection of values C_1, C_n, have they been specified?

          2-3rd line in section 2, why are some c’s (for the cost function) capitalized and others are not

          3-in the cumulative regret formula on page 6, why is T’_c not included is it because it is assumed to be empty, I found this sentence to be confusing “as discussed in Section 4, we used the fact that when the principal commits to a correlated mechanism which is not IC, then she can incur in a constant per-round regret in the worst case, since the behavior of the agents is unpredictable ”


B-In theorem 5.1, is it not reasonable to have a setting where $\ell$ and/or $\iota$ can equal zero? Would this not break the algorithm?

**Questions:**

Please see points A and B in the weaknesses above. Especially point B. Another question I have is the following:

-In mechanism design settings it is reasonable to consider agents engaging in collusion. I did not find comments in the paper about that. This is not necessarily a weakness, since one may just ignore the collusion issue in a problem.

---

> ### Author Response · Authors · 2023-11-14
> **Authors' rebuttal**
>
> We thank the reviewer for the comments and questions. In the following we address each question specifically.
>
> **Re:** *"what is the auxiliary variables z_i in eq (2c) equal to?"*
>
> The auxiliary variables $z_i \in \mathbb{R}^{|B_i|\times |B_i| \times |S_i|}_{\geq 0}$ are needed in order to provide upper bounds to the payment that agent $i$ can receive when she behaves untruthfully. In particular, for any two actions $b_i,b_i^\prime\in B_i$ and signal $s_i\in S_i$, the constraint (2c) guarantees that $z_i[b_i, b_i^\prime, s_i]$ is an upper bound to the payment that agent $i$ receives when she is recommended to play action $b_i$, she deviates by playing action $b_i^\prime$ and then she observes signal $s_i$. We apologize if this was not clear in the paper and we will clarify it in the final version.
>
> **Re:** *"Further, Theorem 3.1 has a collection of values C_1, C_n, have they been specified? [...] 3rd line in section 2, why are some c’s (for the cost function) capitalized and others are not"*
>
> The lowercase function $c_i: B_i\to [0,1]$ represents the cost function for agent $i$ that associates to each action in $B_i$ its cost. The collection of functions $C_1,...,C_n$, where $C_i\to B_i\times B_i\to [-1,1]$ for each $i\in\mathcal N$ are meant to represent pairwise cost differences, *i.e.*, $C_i(b_i, b_i^\prime) = c_i(b_i) - c_i(b_i^\prime)$ for each $i\in\mathcal N$ and for each $b_i,b_i^\prime\in B_i$. In the third line of section 2 there is a typo (it should have been $C_i(b_i,b_i^\prime) = c_i(b_i) - c_i(b_i^\prime)$). We apologize for any inconvenience that this may have caused and we will clarify these points in the final version of the paper.
>
> **Re:** *"in the cumulative regret formula on page 6, why is T’_c not included is it because it is assumed to be empty, I found this sentence to be confusing “as discussed in Section 4, we used the fact that when the principal commits to a correlated mechanism which is not IC, then she can incur in a constant per-round regret in the worst case, since the behavior of the agents is unpredictable”"*
>
> As the reviewer correctly pointed out and as discussed in Section 4, whenever we commit to a non-IC correlated mechanisms we can incur in constant per-round regret. For the sake of exposition, we assume that the principal's utility is $0$ when she commits to a non-IC correlated mechanism. Hence, the regret is the following:
> $$
> R^T = \sum_{t\in[T]} U(\boldsymbol{\mu}^\star, \boldsymbol{\gamma}^\star, \boldsymbol{\pi}^\star) - \sum_{t\in T_c} U(\boldsymbol{\mu}^t, \boldsymbol{\gamma}^t, \boldsymbol{\pi}^t) - \sum_{t\in T_u} U^\circ(\boldsymbol{\gamma}^t, \boldsymbol{\pi}^t) - \sum_{t\in T_c^\prime} 0,
> $$
> which gives exactly the definition of regret that we considered. Furthermore, notice that in light of the findings described in Section 4, we explicitly design our algorithm so to guarantee $T_c^\prime$ to be empty. To enhance clarity, we will explicitly state this aspect in the final version of the paper.
>
> **Re:** *"In theorem 5.1, is it not reasonable to have a setting where $\ell$ and/or $\iota$ can equal zero? Would this not break the algorithm?"*
>
> Settings in which $\ell$ and/or $\iota$ are equal to $0$ correspond to degenerate instances in which states of nature and/or signals are equivalent to the agents (since they are induced with equivalent probability). These degenerate instances can be reduced to non-degenerate ones as follows. If $\ell=0$, agents can avoid to distinguish between the two (or more) signals that induces the same posterior and replace them with a single signal. In a similar way, if $\iota=0$, we can simply remove from the instance the signal that is induced with probability $0$, since it is never observed.
>
>
> **Re:** *"In mechanism design settings it is reasonable to consider agents engaging in collusion. I did not find comments in the paper about that. This is not necessarily a weakness, since one may just ignore the collusion issue in a problem."*
>
> The basic scenarios considered in multi-agent principal-agent problems assumes that the agents are self-interested (see *e.g.*, Cacciamani et al., 2023). Assuming that they can collude, would require a different definition of the IC constraints, with the need to consider more complex correlated deviations of the agents. This may introduce non-trivial complexities (such as the non-existence of equilibria) as it happens in strong correlated equilibria [1]. Notheless, studying these kind of settings constitutes undoubtedly an interesting direction for future research.
> We thank the reviewer for the suggestion.
>
> [1] Ray, Indrajit. "Coalition-proof correlated equilibrium: A definition." Games and Economic Behavior 17.1 (1996): 56-79.

---

> > ### Comment · Reviewer_eEDk · 2023-11-22
> >
> > I thank the reviewers for their response. I have increased my score.

---

### Meta-Review · Area_Chair_arbP · 2023-12-04

**Metareview:**

This paper addresses the mechanism design problem where a principal aims to gather and aggregate information from multiple agents. The authors formulate the problem using linear programming and also address the online learning settings. The authors also discuss the separation between the uncorrelated and the correlated mechanism.

The reviewers overall share positive views of the paper: it addresses a well-motivated problem, the approach seems sound, and the writing is clear. While the technical approachs seem relatively standard, formulating the problem and addressing it presents a non-trivial contrituion, and we would recommend acceptance.

**Justification For Why Not Higher Score:**

While the paper is solid, overall the approaches seem relatively standard.

**Justification For Why Not Lower Score:**

The paper is addressing a well-motivated problem, the proposed approaches are reasonable, and the results are solid.

---

### Decision · Program_Chairs · 2024-01-16

Accept (poster)